# Optimal Rates for Generalization of Gradient Descent for Deep ReLU Classification

**Yuanfan Li**
School of Mathematical Sciences
Zhejiang University
22335070@zju.edu.cn

**Yunwen Lei**
Department of Mathematics
The University of Hong Kong
leiyw@hku.hk

**Zheng-Chu Guo**
School of Mathematical Sciences
Zhejiang University
guozc@zju.edu.cn

**Yiming Ying** *
School of mathematics and statistics
Univerity of Sydney
yiming.ying@sydney.edu.au

## Abstract

Recent advances have significantly improved our understanding of the generalization performance of gradient descent (GD) methods in deep neural networks. A natural and fundamental question is whether GD can achieve generalization rates comparable to the minimax optimal rates established in the kernel setting. Existing results either yield suboptimal rates of $O(1/\sqrt{n})$, or focus on networks with smooth activation functions, incurring exponential dependence on network depth $L$. In this work, we establish optimal generalization rates for GD with deep ReLU networks by carefully trading off optimization and generalization errors, achieving only polynomial dependence on depth. Specifically, under the assumption that the data are NTK separable from the margin $\gamma$, we prove an excess risk rate of $\widetilde{O}(L^6/(n\gamma^2))$, which aligns with the optimal SVM-type rate $\widetilde{O}(1/(n\gamma^2))$ up to depth-dependent factors. A key technical contribution is our novel control of activation patterns near a reference model, enabling a sharper Rademacher complexity bound for deep ReLU networks trained with gradient descent.

## 1 Introduction

Deep neural networks trained through first-order optimization methods have achieved remarkable empirical success in diverse domains [Krizhevsky et al., 2012]. Despite their widespread adoption, a rigorous theoretical understanding of their optimization dynamics and generalization behavior remains incomplete, particularly for ReLU networks. The inherent challenges arise from the non-smoothness and non-convexity of the loss landscape induced by ReLU activations and network architectures, which complicate the classical analysis. Intriguingly, empirical evidence demonstrates that over-parameterized models often achieve zero training error but still generalize well even in the absence of explicit regularization [Zhang et al., 2016]. This phenomenon has spurred significant theoretical research to understand its underlying mechanisms.

A prominent line of research uses the neural tangent kernel (NTK) framework to analyze neural network training [Jacot et al., 2018]. In the infinite-width limit, gradient descent (GD) dynamics can be characterized by functions in the NTK's reproducing kernel Hilbert space (RKHS), with convergence guarantees established for both shallow and deep networks [Du et al., 2019b,a]. These results demonstrate that GD converges to global minima within a local neighborhood of initialization,

---

*Corresponding author

39th Conference on Neural Information Processing Systems (NeurIPS 2025).

| | Activation | Width | Training error | Generalization error | Network |
|---|---|---|---|---|---|
| Ji and Telgarsky | ReLU | $\widetilde{\Omega}\left(\frac{1}{\gamma^8}\right)$ | $\widetilde{O}\left(\frac{1}{\gamma^2 T}\right)$ | $\widetilde{O}\left(\frac{1}{\gamma^2\sqrt{n}}\right)$ | Shallow |
| Lei et al. | ReLU | $\widetilde{\Omega}\left(\frac{1}{\gamma^8}\right)$ | $\widetilde{O}\left(\frac{1}{\gamma^2 T}\right)$ | $\widetilde{O}\left(\frac{1}{\gamma^2 n}\right)$ | Shallow |
| Chen et al. | ReLU | $\widetilde{\Omega}\left(\frac{L^{22}}{\gamma^8}\right)$ | $\widetilde{O}\left(\frac{1}{\gamma^2 T}\right)$ | $\widetilde{O}\left(\frac{e^{O(L)}}{\gamma}\sqrt{\frac{m}{n}}\right)$ | Deep |
| Taheri et al. | Smooth | $\widetilde{\Omega}\left(\frac{1}{\gamma^{6L+4}}\right)$ | $\widetilde{O}\left(\frac{1}{\gamma^2 T}\right)$ | $\widetilde{O}\left(\frac{e^{O(L)}}{\gamma^2 n}\right)$ | Deep |
| **Ours** | ReLU | $\widetilde{\Omega}\left(\frac{L^{16}}{\gamma^8}\right)$ | $\widetilde{O}\left(\frac{1}{\gamma^2 T}\right)$ | $\widetilde{O}\left(\frac{L^6}{\gamma^2 n}\right)$ | Deep |

Table 1: Comparison of learning neural networks with GD on NTK separable data with prior works. Here $m$ is the network width, $L$ is the network depth, $n$ is the sample size, $\gamma$ is the NTK-margin, $T$ is the number of iterations.

provided that the network width is sufficiently large. In particular, the appealing work [Arora et al., 2019a] showed that, if the network width $m = \widetilde{O}(\text{poly}(n, 1/\lambda_0, L))$, then the generalization error is of the order $L\sqrt{\frac{\mathbf{y}^\top(\mathbf{H}^\infty)^{-1}\mathbf{y}}{n}}$, where $\mathbf{H}^\infty$ denotes the NTK gram matrix over the training data and $\lambda_0 = \lambda_{\min}(\mathbf{H}^\infty)$. However, the assumption $\lambda_0 > 0$ is a strong assumption because it tends to zero if the size of the training data tends to infinity as shown by Su and Yang [2019]. Ji and Telgarsky [2020] achieved logarithmic width requirements for NTK-separable data with a margin $\gamma$. They derived the risk bound of order $1/(\gamma^2\sqrt{n})$ for two-layer ReLU networks using Rademacher complexity. The recent work Chen et al. [2021] extended their results from shallow to deep neural networks, the authors considered the NTK feature learning and proved the bound of $\widetilde{O}\left(\frac{e^{O(L)}}{\gamma}\sqrt{\frac{m}{n}}\right)$. Recently, Lei et al. [2026] derived the bound of $1/(\gamma^2 n)$ for two-layer ReLU networks. However, all the above bounds explicit suboptimal $\frac{1}{\sqrt{n}}$ dependence on the sample size $n$ or only focus on shallow networks.

Complementing the NTK framework, another line of research employs algorithmic stability to analyze neural networks. In particular, Liu et al. [2020] demonstrated that the Hessian spectral norm scales with width $m$ as $\widetilde{O}\left(\frac{1}{\sqrt{m}}\right)$, providing the theoretical basis for subsequent studies on generalization in overparameterized networks [Richards and Kuzborskij, 2021, Lei et al., 2022, Taheri and Thrampoulidis, 2024, Taheri et al., 2025]. The work [Taheri and Thrampoulidis, 2024] achieves the bound of $\widetilde{O}(1/n\gamma^2)$ for shallow neural networks, which is almost optimal, as illustrated by Shamir [2021], Schliserman and Koren [2023]. More recently, Taheri et al. [2025] extended this line of work to deep networks, obtaining a generalization bound of $\widetilde{O}(e^{O(L)}/(n\gamma^2))$. However, this approach often assume smooth activation functions and can not apply to the non-smooth ReLU networks. In summary, these works either consider smooth neural networks or develop generalization bounds with exponential dependency on $L$. These limitations motivate two fundamental questions:

*Can we develop optimal risk bounds of $1/(\gamma^2 n)$ for deep ReLU networks through refined Rademacher complexity analysis? Furthermore, is it possible to replace the exponential dependence on $L$ with $\text{poly}(L)$ by using neural networks with width as a polylogarithmic function of $n$?*

In this paper, we provide affirmative answers to both questions. Our main contributions are listed below.

1. We prove that gradient decent with step size $\eta$ and $T$ iterations achieves the convergence rate of $F_S(\overline{\mathbf{W}})/(\eta T)$, where $F_S(\overline{\mathbf{W}}) = 3\eta T \mathcal{L}_S(\overline{\mathbf{W}}) + \|\overline{\mathbf{W}} - \mathbf{W}(0)\|_F^2$, $\overline{\mathbf{W}}$ is a reference model, $\mathbf{W}(0)$ is the initialization point and $\mathcal{L}_S(\cdot)$ is the training error. This indicates that all iterates remain within a neighborhood of $\overline{\mathbf{W}}$. By refining the analysis of ReLU activation patterns, we reduce the overparameterization requirement by a factor of $L^6$ as compared to Chen et al. [2021] (see Remark 1 ).

2. We establish a population risk bound of $\widetilde{O}(L^4 F(\overline{\mathbf{W}})/n)$, where $F(\overline{\mathbf{W}})$ extends the empirical counterpart $F_S(\overline{\mathbf{W}})$ to the population setting. Our analysis introduces two key technical contributions. First, we derive sharper Rademacher complexity bounds for the hypothesis class encompassing all gradient descent iterates. A central innovation is our representation of the

complexity via products of sparse matrices, whose norms are tightly controlled using optimization-informed estimates (see Remark 2). Second, by leveraging the covering number techniques, we prove that ReLU networks remain $\widetilde{O}(L^2)$-Lipschitz continuous in a neighborhood around the initialization—a substantial improvement over previous exponential bounds [Xu and Zhu, 2024, Taheri et al., 2025] (see Remark 3).

3. For NTK separable data with a margin $\gamma$, we show that neural networks with $m = \widetilde{O}(L^{16}\log(n/\delta)(\log n)^8/\gamma^8)$ can achieve $\widetilde{O}(L^6/(\gamma^2 n))$ risk. This improves on Chen et al. [2021]'s $\widetilde{O}(e^{O(L)}\gamma^{-1}\sqrt{m/n})$ by simultaneously (a) removing the exponential depth dependence, (b) eliminating the $\sqrt{m}$ width factor and (c) achieving the optimal dependence on sample complexity (see Table 1 for a comparison with existing work).

## 2 Related Works

### 2.1 Optimization

The foundational work of Jacot et al. [2018] introduced the Neural Tangent Kernel (NTK) framework, demonstrating that in the infinite-width limit, neural networks behave as linear models with a fixed tangent kernel [Liu et al., 2020, Lee et al., 2019]. This lazy training regime [Chizat et al., 2019], where parameters remain close to initialization, enables gradient descent to converge to global optima near initialization [Du et al., 2019a, Arora et al., 2019a]. These analyses showed that the training dynamics can be governed by the NTK Gram matrix, which leads to substantial overparameterization $(m \gtrsim n^6/\lambda_0^4)$. This was later improved by Oymak and Soltanolkotabi [2020]. They showed that if the square-root of the number of the network parameters exceeds the size of the training data, randomly initialized gradient descent converges at a geometric rate to a nearby global optima. The work (Ji and Telgarsky [2020]) achieved polylogarithmic width requirements for logistic loss by leveraging the 1-homogeneity of two-layer ReLU networks. However, it should be mentioned that this special property does not hold for deep networks. The NTK framework was extended to deep architectures by Arora et al. [2019b] for CNNs and by Du et al. [2019b] for ResNets using the last-layer NTK. Xu and Zhu [2024] pointed out that such a characterization is loose, only capturing the contribution from the last layer. They further gave the uniform convergence of all layers as $m \to \infty$ and convergence guarantee for stochastic gradient descent (SGD) in streaming data setting. Allen-Zhu et al. [2019] showed that the optimization landscape is almost-convex and semi-smooth, based on which they proved that SGD can find global minima. Cao and Gu [2019] introduced the neural tangent random feature and showed the convergence of SGD under the overparameterized assumption $m \gtrsim n^7$.

### 2.2 Generalization

The NTK framework has yielded generalization bounds scaling as $\sqrt{\mathbf{y}^\top (\mathbf{H}^\infty)^{-1}\mathbf{y}/n}$ [Arora et al., 2019a, Cao and Gu, 2019]. This data-dependent complexity measure helps to distinguish between random labels and true labels. Li and Liang [2018] showed that SGD trained networks can achieve small test error on specific structured data. A very popular approach to studying the generalization of neural networks is via the uniform convergence, which analyzes generalization gaps in a hypothesis space using tools such as Rademacher complexity or covering numbers [Neyshabur et al., 2015, Bartlett et al., 2017, Golowich et al., 2018, Liu et al., 2024]. However, this could lead to vacuous generalization bound in some cases [Nagarajan and Kolter, 2019]. Moreover, these bounds typically exhibit exponential dependence on depth $L$, thus often leading to loose bounds [Chen et al., 2021]. This capacity-based method usually results in the generalization rate of the order $\widetilde{O}(1/\sqrt{n})$. Recent work has also exploited stability arguments for generalization guarantees [Richards and Kuzborskij, 2021, Lei et al., 2022, Taheri and Thrampoulidis, 2024, Deora et al., 2023, Taheri et al., 2025]. The main idea of algorithmic stability is to study how the perturbation of training samples would affect the output of an algorithm [Rogers and Wagner, 1978]. The connection to generalization bound was established in Bousquet and Elisseeff [2002]. Hardt et al. [2016] gave the stability analysis of SGD for Lipschitz, smooth and convex problems. Lei and Ying [2020] further studied SGD under much wilder assumptions. Liu et al. [2020] identified weak convexity of neural networks, enabling stability analyses with polynomial width requirements for quadratic loss [Richards and Kuzborskij, 2021, Lei et al., 2022]. Moreover, Taheri and Thrampoulidis [2024], Taheri et al. [2025] obtained generalization bounds of order $\widetilde{O}(1/n)$ by using a generalized local quasi-convexity property for

sufficiently parameterized networks. However, these methods depend on smooth activations, and whether similar or even better bound can be established for deep ReLU networks is still unknown. The recent work derived excess risk bounds of order $\widetilde{O}(1/n)$ for shallow ReLU networks [Lei et al., 2026].

# 3 Preliminaries

**Notation** Throughout the paper, we denote $a \lesssim b$ if there exists a constant $c > 0$ such that $a \leq cb$, and denote $a \asymp b$ if both $a \lesssim b$ and $b \lesssim a$ hold. We use the standard notation $O(\cdot), \Omega(\cdot)$ and use $\widetilde{O}(\cdot), \widetilde{\Omega}(\cdot)$ to hide polylogarithmic factors. Denote by $\mathbb{I}\{\cdot\}$ the indicator function (i.e., taking the value 1 if the argument holds true, and 0 otherwise). We use $\mathcal{N}(\mu, \sigma^2)$ to denote the Gaussian distribution of mean $\mu$ and variance $\sigma^2$. For a positive integer $n$, we denote $[n] := \{1, \ldots, n\}$. For a vector $x \in \mathbb{R}^d$, we use $\|x\|_2$ to denote its Euclidean norm. For a matrix $A \in \mathbb{R}^{m \times n}$, we denote $\|A\|_2$ and $\|A\|_F$ the corresponding spectral norm and Frobenius norm respectively. The $(2, 1)$-norm of $A$ is defined as $\|A\|_{2,1} = \sum_{j=1}^{n} \|A_{:j}\|_2$. Let $\langle \cdot, \cdot \rangle$ be the inner product of a vector or a matrix, i.e., for any matrices $A, B \in \mathbb{R}^{m \times n}$, we have $\|A\|_F^2 = tr(A^\top A)$ and $\langle A, B \rangle = tr(A^\top B)$. Let $L \in \mathbb{N}$, $\mathbf{A} = (A_1, \ldots, A_L)$ and $\mathbf{B} = (B_1, \ldots, B_L)$ be two collections of arbitrary matrices such that $A_i$ and $B_i$ have the same size for all $i \in [L]$. We define $\langle \mathbf{A}, \mathbf{B} \rangle = \sum_{i=1}^{L} tr(A_i^\top B_i)$. Denote $\|\mathbf{A}\|_{2,\infty} = \max_l \|A_l\|_2$. For a matrix $\mathbf{W}$, we define $(\mathbf{w}_r)^\top$ the $r$-th row of $\mathbf{W}$. Denote $\|\cdot\|_0$ the $l^0$-norm which is the number of nonzero entries of a matrix or a vector. We denote $C \geq 1$ as an absolute value, which may differ from line to line.

Let $\mathcal{X} \subseteq \mathbb{R}^d$ be the input space, $\mathcal{Y} = \{1, -1\}$ be the output space, and $\mathcal{Z} = \mathcal{X} \times \mathcal{Y}$. Let $\rho$ be a probability measure defined on $\mathcal{Z}$. Let $S = \{z_i = (\mathbf{x}_i, y_i) : i = 1, \ldots, n\}$ be a training dataset drawn from $\rho$. Let $\mathcal{W} := \mathbb{R}^{m \times d} \times (\mathbb{R}^{m \times m})^{L-1}$ be the parameter space. $\mathbf{W}^1 \in \mathbb{R}^{m \times d}$ and $\mathbf{W}^l \in \mathbb{R}^{m \times m}$ for $l = 2, \ldots, L$ is the weight of the $l$-th hidden layer. $\mathbf{W} = (\mathbf{W}^1, \ldots, \mathbf{W}^L) \in \mathcal{W}$ denotes the collection of weight matrices for all layers. Let $\mathbf{a} = (a_1, \ldots, a_m)^\top \in \mathbb{R}^m$ be the weight vector of the output layer and $\sigma(\cdot) = \max\{\cdot, 0\}$ denote the ReLU activation function. For $\mathbf{x} \in \mathcal{X}$, we consider the $L$-layer deep ReLU neural networks with width $m$ as follows,

$$f_{\mathbf{W}}(\mathbf{x}) = \mathbf{a}^\top \sqrt{\frac{2}{m}} \sigma \left( \mathbf{W}^L \cdots \sqrt{\frac{2}{m}} \sigma(\mathbf{W}^1 \mathbf{x}) \right). \tag{1}$$

Given an input $\mathbf{x} \in \mathcal{X}$ and parameter matrix $\mathbf{W} = (\mathbf{W}^1, \cdots, \mathbf{W}^L)$ of an $L$-layer ReLU network $f_{\mathbf{W}}(\mathbf{x})$. We denote the output of the $l$-th layer by $h^l(\mathbf{x}) = \sqrt{\frac{2}{m}} \sigma(\mathbf{W}^l h^{l-1}(\mathbf{x}))$ with $h^0(\mathbf{x}) = \mathbf{x}$. Then $f_{\mathbf{W}}(\mathbf{x}) = \mathbf{a}^\top h^L(\mathbf{x})$. We define $\mathcal{B}_R(\mathbf{W}) = \{\widetilde{\mathbf{W}} \in \mathcal{W} : \max_l \|\mathbf{W} - \widetilde{\mathbf{W}}^l\|_2 \leq R\}$. The performance of the network $f_{\mathbf{W}}(\mathbf{x})$ is measured by the following empirical risk $\mathcal{L}_S(\mathbf{W})$ and population risk $\mathcal{L}(\mathbf{W})$, respectively:

$$\mathcal{L}_S(\mathbf{W}) = \frac{1}{n} \sum_{i=1}^{n} \ell(y_i f_{\mathbf{W}}(\mathbf{x}_i)) \quad \text{and} \quad \mathcal{L}(\mathbf{W}) = \mathbb{E}_z \ell(y f_{\mathbf{W}}(\mathbf{x})),$$

where we use logistic loss $\ell(z) := \log(1 + \exp(-z))$ throughout this paper. We further assume the following symmetric initialization [Nitanda and Suzuki, 2020, Kuzborskij and Szepesvári, 2023, Xu and Zhu, 2024]:

**Assumption 1** (Symmetric initialization). *Without loss of generality, we assume the network width $m$ is even, and $a_{r+\frac{m}{2}} = -a_r \in \{-1, +1\}$ for $1 \leq r \leq m/2$. $\mathbf{W}(0) \in \mathcal{W}$ satisfies*

$$\mathbf{w}_r^1(0) \sim \mathcal{N}(0, \mathbf{I}_d), \mathbf{w}_r^l(0) \sim \mathcal{N}(0, \mathbf{I}_m) \quad 2 \leq l \leq L - 1 \text{ and } r \in [m],$$

$$\mathbf{w}_r^L(0) \sim \mathcal{N}(0, \mathbf{I}_m) \text{ for } r = \{1, \ldots, \frac{m}{2}\}, \text{ and } \mathbf{w}_{r+\frac{m}{2}}^L(0) = \mathbf{w}_r^L(0). \tag{2}$$

We remark that this initialization is for theoretical simplicity, using general initialization techniques will not affect the main results. We fix the output weights $\{a_r\}$ and use Gradient Descent (GD) to train the weight matrix $\mathbf{W}$ [Ji and Telgarsky, 2020, Arora et al., 2019a, Zou et al., 2018].

**Definition 1** (Gradient Descent). *GD updates* $\{\mathbf{W}(k)\}$ *by*

$$\mathbf{W}^l(t+1) = \mathbf{W}^l(t) - \eta \frac{\partial \mathcal{L}_S(\mathbf{W}(t))}{\partial \mathbf{W}^l(t)} \text{ for all } l \in [L], t = 0, \cdots, T-1, \quad (3)$$

*where* $\eta > 0$ *is the step size.*

Note that in each layer we employ $\sqrt{2/m}$ as the regularization factor instead of the conventional $\sqrt{1/m}$ [Ji and Telgarsky, 2020], which is due to $\mathbb{E}_{x \sim \mathcal{N}(0,1)} \sigma^2(x) = 1/2$ for our activation function $\sigma(\cdot)$. This scaling matches both the theoretical framework of Du et al. [2019a] and the initialization scheme of [He et al., 2015] (where weights $\mathbf{w}_r^l \sim \mathcal{N}(0, 2/m)$). As will be shown later (Appendix A), this regularization ensures stable gradient propagation and maintains consistent variance across layers.

The following assumption is standard in the literature [Cao and Gu, 2019, Ji and Telgarsky, 2020, Chen et al., 2021].

**Assumption 2.** *We assume* $\mathcal{X} = S^{d-1}$ *be the sphere.*

Throughout the paper, we assume that Assumptions 1 and 2 always hold true.

**Error decomposition**  In this work, we analyze the performance of gradient descent through the lens of population risk. To facilitate this analysis, we decompose the population risk $\mathcal{L}(\mathbf{W}(T))$ as follows

$$\mathcal{L}(\mathbf{W}(T)) = (\mathcal{L}(\mathbf{W}(T)) - \mathcal{L}_S(\mathbf{W}(T))) + \mathcal{L}_S(\mathbf{W}(T)),$$

where the first term captures the generalization gap, quantifying the network's performance on unseen data. The second term represents the optimization error, which reflects GD's ability to find global minima. We will use tools in the optimization theory to study the optimization error [Ji and Telgarsky, 2020, Schliserman and Koren, 2022], and Rademacher complexity to control the generalization gap [Mohri et al., 2018].

## 4  Main Results

In this section, we present the main results. In Section 4.1, we show the optimization analysis of gradient descent. In Section 4.2, we use Rademacher complexity to control the generalization gap. In Section 4.3, we apply our generalization results to NTK-separable data with a margin $\gamma$.

### 4.1  Optimization Analysis

We introduce the following notations for a reference model $\overline{\mathbf{W}}$

$$F_S(\overline{\mathbf{W}}) := 3\eta T \mathcal{L}_S(\overline{\mathbf{W}}) + \|\mathbf{W}(0) - \overline{\mathbf{W}}\|_F^2, \quad \tilde{F}_S(\overline{\mathbf{W}}) = \frac{1}{n} \sum_{i=1}^{n} |\ell'(y_i f_{\overline{\mathbf{W}}}(\mathbf{x}_i)|.$$

Without loss of generality, we assume $F_S(\overline{\mathbf{W}}) \geq 1$.

**Theorem 1.** *Let Assumptions 1, 2 hold. If* $m \gtrsim L^{16}(\log m)^4 \log(nL/\delta) F_S^4(\overline{\mathbf{W}}), \eta \leq \min\{4/(5L), 1/(20L\tilde{F}_S(\overline{\mathbf{W}}))\}$, *then with probability at least* $1 - \delta$, *for all* $t \leq T$ *we have*

$$\max_l \|\mathbf{W}^l - \overline{\mathbf{W}}^l\|_2^2 \leq \|\mathbf{W}(t) - \overline{\mathbf{W}}\|_F^2 \leq F_S(\overline{\mathbf{W}}) \quad and \quad \eta \sum_{k=0}^{t-1} \mathcal{L}_S(\mathbf{W}(k)) \leq F_S(\overline{\mathbf{W}}).$$

*Remark* 1. Our theorem shows that the convergence rate is bounded by the optimization error of a reference model, implying that any low-loss reference point guarantees good convergence. While prior works relied on NTK-induced solutions [Richards and Kuzborskij, 2021, Arora et al., 2019a], we prove that there exists a reference model near initialization under the milder Assumption 3. Furthermore, our analysis implies that all training iterates remain within a neighborhood of the reference point, and thus near initialization, aligned with previous observations but without studying the kernel or the corresponding Gram matrix directly [Du et al., 2019a,b].

Here we provide the proof sketch and compare it with previous works. The starting point is to show deep ReLU networks admit almost convexity ( Lemma 19 ):

$$f_{\mathbf{W}}(\mathbf{x}_i) - f_{\overline{\mathbf{W}}}(\mathbf{x}_i) - \left\langle \frac{\partial f_{\mathbf{W}}(\mathbf{x}_i)}{\partial \mathbf{W}}, \mathbf{W} - \overline{\mathbf{W}} \right\rangle = \widetilde{O}\left( \frac{L^{8/3} R^{4/3}}{m^{1/6}} \right) \tag{4}$$

for $\mathbf{W}, \overline{\mathbf{W}} \in \mathcal{B}_R(\mathbf{W}(0))$.

Then we can show all the iterates remain in $\mathcal{B}_{2\sqrt{F_S(\overline{\mathbf{W}})}}(\mathbf{W}(0))$ and the following inequality holds (Lemma 21),

$$\|\mathbf{W}(t+1) - \overline{\mathbf{W}}\|_F^2 \leq \|\mathbf{W}(t) - \overline{\mathbf{W}}\|_F^2 - \eta \mathcal{L}_S(\mathbf{W}(t)) + 3\eta \mathcal{L}_S(\overline{\mathbf{W}}). \tag{5}$$

Telescoping gives the theorem. Chen et al. [2021] introduce the following neural tangent random feature (NTRF) function class:

$$\mathcal{F}(\mathbf{W}(0), R) = \left\{ F_{\mathbf{W}(0), \mathbf{w}}(\mathbf{x}) = f_{\mathbf{W}(0)}(\mathbf{x}) + \left\langle \frac{\partial f_{\mathbf{W}(0)}(\mathbf{x})}{\partial \mathbf{W}(0)}, \mathbf{W} - \mathbf{W}(0) \right\rangle : \mathbf{W} \in \mathcal{B}_R(\mathbf{W}(0)) \right\}.$$

They show that gradient descent achieves a training loss of at most $3\epsilon_{\text{NTRF}}$, where $\epsilon_{\text{NTRF}}$ denotes the minimal loss over the NTRF function class (see Theorem 3.3 therein). In contrast, our approach directly analyzes the GD iterates and shows that the existence of a nearby reference point with small training error is sufficient to ensure convergence. For a fair comparison, under Assumption 3, both analyses yield an optimization error of $\widetilde{O}(1/T)$. However, our method significantly relaxes the overparameterization requirement, improving the width dependence by a factor of $L^6$ (see Remark 5).

## 4.2 Generalization Analysis

We use Rademacher complexity to study the generalization gap, which measures the ability of a function class to correlate random noises.

**Definition 2** (Rademacher complexity). *Let $\mathcal{F}$ be a class of real-valued functions over a space $\mathcal{X}$, $S_1 = \{\mathbf{x}_1, \cdots, \mathbf{x}_n\} \subset \mathcal{X}$. We define the following empirical Rademacher complexity as*

$$\mathfrak{R}_{S_1}(\mathcal{F}) = \mathbb{E}_\epsilon \left[ \sup_{f \in \mathcal{F}} \frac{1}{n} \sum_{i \in [n]} \epsilon_i f(\mathbf{x}_i) \right],$$

*where $\epsilon = (\epsilon_i)_{i \in [n]} \sim \{\pm 1\}^n$ are independent Rademacher variables, i.e., taking values in $\{\pm 1\}$ with the same probability.*

We further define the following worst-case Rademacher complexity,

$$\mathfrak{R}_{S_1, n}(\mathcal{F}) = \sup_{\widetilde{S} \subset S_1 : |\widetilde{S}| = n} \mathfrak{R}_{\widetilde{S}}(\mathcal{F}).$$

We define $G = \sup_z \ell(y f_{\overline{\mathbf{W}}}(\mathbf{x}))$, and

$$F(\overline{\mathbf{W}}) = 3\eta T \left( 2\mathcal{L}(\overline{\mathbf{W}}) + \frac{7G \log(2/\delta)}{6n} \right) + \|\mathbf{W}(0) - \overline{\mathbf{W}}\|_F^2. \tag{6}$$

We consider the following function space

$$\mathcal{F} := \{\mathbf{x} \to f_{\mathbf{W}}(\mathbf{x}) : \mathbf{W} \in \mathcal{W}_1\}, \tag{7}$$

where the parameter space is defined as

$$\mathcal{W}_1 = \left\{ \mathbf{W} \in \mathcal{W} : \|\mathbf{W} - \overline{\mathbf{W}}\|_F^2 \leq F(\overline{\mathbf{W}}) \right\}. \tag{8}$$

Here we use $F(\overline{\mathbf{W}})$ instead of $F_S(\overline{\mathbf{W}})$ to get a data-independent hypothesis space. We will show $F(\overline{\mathbf{W}})$ is an upper bound of $F_S(\overline{\mathbf{W}})$ with high probability. According to Theorem 1, all the iterations fall into $\mathcal{W}_1$ with high probability. We use the following lemma to relate the generalization gap of smooth loss function with Rademacher complexity.

**Lemma 1** (Srebro et al. [2010]). *Let $G' = \sup_{z, \mathbf{W} \in \mathcal{W}_1} \ell(y f_{\mathbf{W}}(\mathbf{x}))$. For any $0 < \delta < 1$, we have with probability at least $1 - \delta/2$ over $S$, for any $\mathbf{W} \in \mathcal{W}_1$,*

$$\mathcal{L}(\mathbf{W}) - \mathcal{L}_S(\mathbf{W}) \lesssim \mathcal{L}_S^{1/2}(\mathbf{W}) \left( \frac{1}{2} (\log n)^{3/2} \mathfrak{R}_{S_1,n}(\mathcal{F}) + \left( \frac{G' \log(2/\delta)}{n} \right)^{1/2} \right)$$
$$+ \frac{1}{4} (\log n)^3 \mathfrak{R}_{S_1,n}^2(\mathcal{F}) + \frac{G' \log(2/\delta)}{n}.$$

Now we need to control $\mathfrak{R}_{S_1,n}(\mathcal{F})$ and $G'$. As will be shown in Lemma 22, with high probability there holds

$$\mathfrak{R}_{S_1,n}(\mathcal{F}) = \widetilde{O}\left( L^2 \sqrt{\frac{F(\overline{\mathbf{W}})}{n}} \right). \tag{9}$$

To estimate $G'$, we employ covering numbers to derive a uniform upper bound of $f_{\mathbf{W}}(\mathbf{x}) - f_{\overline{\mathbf{W}}}(\mathbf{x})$. Then we use the smoothness of $\ell$ to show that for all $G' - 2G \lesssim L^4 \log m F(\overline{\mathbf{W}})$. Plugging these bounds into Lemma 1 gives the generalization gap. Combined with Theorem 1, we derive the following excess risk error. The full proofs are provided in Appendix C.

**Theorem 2.** *Let Assumptions 1, 2 hold. If $m \gtrsim L^{16} d (\log m)^5 \log(nL/\delta) F_S^4(\overline{\mathbf{W}}), \eta \leq \min\{4/(5L), 1/(20L\tilde{F}_S(\overline{\mathbf{W}}))\}, \eta T \asymp n$, then with probability at least $1 - \delta$, we have*

$$\frac{1}{T} \sum_{t=0}^{T-1} \mathcal{L}(\mathbf{W}(t)) = \widetilde{O}\left( \frac{L^4 F(\overline{\mathbf{W}}) + G \log(2/\delta)}{n} \right).$$

*Remark* 2 (Improved Rademacher complexity). In previous work [Chen et al., 2021], they derived the bound of $\widetilde{O}\big(4^L L \sqrt{m F(\overline{\mathbf{W}})/n}\big)$. Specifically, they use the generalization analysis in Bartlett et al. [2017], which requires to estimate the following term

$$\left[ \prod_{l=1}^{L} \|\mathbf{W}^l\|_2 \right] \cdot \left[ \sum_{l=1}^{L} \frac{\|(\mathbf{W}^l)^\top - (\overline{\mathbf{W}}^l)^\top\|_{2,1}^{2/3}}{\|\mathbf{W}^l\|_2^{2/3}} \right]^{3/2}.$$

Note that $\prod_{l=1}^{L} \|\mathbf{W}^l\|_2$ could lead to exponential dependence on the depth. Moreover, $\|(\mathbf{W}^l)^\top - (\overline{\mathbf{W}}^l)^\top\|_{2,1} \leq \sqrt{m} \|\mathbf{W}^l - \overline{\mathbf{W}}^l\|_F$, inducing an explicit $\sqrt{m}$ term. However, we reduce the dependence of $L$ to polynomial $(L^2)$. Furthermore, we sharpen the dependence on width from $\sqrt{m}$ to logarithmic terms. The key idea is to introduce the following expression

$$f_{\mathbf{W}}(\mathbf{x}_i) - f_{\mathbf{W}(0)}(\mathbf{x}_i) = \mathbf{a}^\top \sum_{l=1}^{L} \widehat{\mathbf{G}}_{L,0}^l(\mathbf{x}_i)(\mathbf{W}^l - \mathbf{W}^l(0)) h_0^{l-1}(\mathbf{x}_i),$$

where $\widehat{\mathbf{G}}_{L,0}^l(\mathbf{x}_i)$ is a matrix defined in (36), $h_0^l$ is the output of $l$-th layer of $f_{\mathbf{W}(0)}$. We further show that $\widehat{\mathbf{G}}_{L,0}^l(\mathbf{x}_i)$ is of the order $\widetilde{O}(L/\sqrt{m})$ with high probability, from which we can derive $\mathfrak{R}_{S_1,n}(\mathcal{F}) = \widetilde{O}(L^2/\sqrt{n})$. We will show that this improved Rademacher complexity is crucial to get almost optimal risk bounds in NTK separable data.

*Remark* 3 (Analysis of Lipschitzness). To bound the term $G' = \sup_{z, \mathbf{W} \in \mathcal{W}_1} \ell(y f_{\mathbf{W}}(\mathbf{x}))$, we analyze the difference $f_{\mathbf{W}}(\mathbf{x}) - f_{\overline{\mathbf{W}}}(\mathbf{x})$. Since both $\mathbf{W}, \overline{\mathbf{W}} \in \mathcal{B}_R(\mathbf{W}(0))$ for some $R$, we only need to study the local variation $f_{\mathbf{W}}(\mathbf{x}) - f_{\mathbf{W}(0)}(\mathbf{x})$. This approach necessitates characterizing the uniform behavior of deep networks in $\mathcal{B}_R(\mathbf{W}(0))$, specifically establishing control over their Lipschitz constants near initialization. Existing works usually lead to an exponential dependence on $L$ [Xu and Zhu, 2024, Taheri et al., 2025], thus resulting in a $e^{O(L)}$ term in the generalization bound. In particular, Lemma F.3 and F.5 in Liu et al. [2020] pointed out that $\|h^l(\mathbf{x})\| \leq C^L, \|\partial f_{\mathbf{W}}(\mathbf{x})/\partial h^l(\mathbf{x})\|_2 \leq C^{L-l+1}\sqrt{m}$. Based on these observations, Taheri et al. [2025] showed that

$$\left\| \frac{\partial f_{\mathbf{W}}(\mathbf{x})}{\partial \mathbf{W}^l} \right\|_2 \leq C^L.$$

On the other hand, to analyze the output difference near initialization, we observe that

$$f_{\mathbf{W}}(\mathbf{x}) - f_{\mathbf{W}(0)}(\mathbf{x}) = \mathbf{a}^\top (h^L(\mathbf{x}) - h_0^L(\mathbf{x})) \leq \sqrt{m} \|h^L(\mathbf{x}) - h_0^L(\mathbf{x})\|_2,$$

reducing our task to bounding the hidden layer perturbation. Previous approaches, including Xu and Zhu [2024] and Du et al. [2019b], employ a recursive estimation:

$$\|h^L(\mathbf{x}) - h_0^L(\mathbf{x})\|_2 = \sqrt{\frac{2}{m}} \|\sigma(\mathbf{W}^L h^{L-1}(\mathbf{x})) - \sigma(\mathbf{W}^L(0) h_0^{L-1}(\mathbf{x}))\|_2$$

$$\leq \sqrt{\frac{2}{m}} (\|(\mathbf{W}^L - \mathbf{W}^L(0)) h^{L-1}(\mathbf{x})\|_2 + \|\mathbf{W}^L(0)(h^{L-1}(\mathbf{x}) - h_0^{L-1}(\mathbf{x})\|_2)$$

$$\lesssim \frac{R}{\sqrt{m}} (\|h^{L-1}(\mathbf{x}) - h_0^{L-1}(\mathbf{x})\|_2 + C^L) + \|h^{L-1}(\mathbf{x}) - h_0^{L-1}(\mathbf{x})\|_2 \leq \frac{C^L R}{\sqrt{m}},$$

where in the second inequality they used $\|h_0^l(\mathbf{x})\|_2 \leq C^L$ and $\|\mathbf{W}^l(0)\|_2 \lesssim \sqrt{m}$. Although this method provides a straightforward bound, it leads to an exponential dependence on depth $L$ due to the recursive nature of the estimation.

In contrast to previous work, we develop the covering-number strategy to avoid the exponential dependence on depth. Specifically, we first show that for any finite set of size $N$: $K = \{\mathbf{x}^1, \cdots, \mathbf{x}^N\}$, if $m = \widetilde{\Omega}(L^{10} \log(N) R^2)$, then $\|h^l(\mathbf{x}^i) - h_0^l(\mathbf{x}^i)\|_2 = \widetilde{O}\left(\frac{L^2 R}{\sqrt{m}}\right)$ holds for $i \in [N], l \in [L]$ (Lemma 15). We further take a $1/(C^L \sqrt{m})$-covering $D = \{\mathbf{x}^j : j = 1, \ldots, |D|\}$ of the input space. Recall that the input space $\mathcal{X} = S^{d-1}$, it is well known from Corollary 4.2.13 in Vershynin [2018] that the number of $1/(C^L \sqrt{m})$-covering is given by $|D| \leq (1 + 2C^L \sqrt{m})^d$. Applying Lemma 15 to $D$ derives that if $m = \widetilde{\Omega}(L^{10} \log(|D|) R^2)$, then

$$\|h^l(\mathbf{x}^j) - h_0^l(\mathbf{x}^j)\|_2 = \widetilde{O}\left(\frac{L^2 R}{\sqrt{m}}\right), \quad \mathbf{x}^j \in D, l \in [L].$$

Note that although the covering number could be exponential in $L$, we only require logarithm of it, thus leading to polynomial dependence. For any input $\mathbf{x}$, we use the closest cover point $\mathbf{x}^j \in D$ to approximate $\|h^l(\mathbf{x}) - h^l(\mathbf{x}^j)\|_2, \|h_0^l(\mathbf{x}) - h_0^l(\mathbf{x}^j)\|_2$. Combining these yields the key technical lemma (Lemma 16):

$$\sup_{\mathbf{x} \in \mathcal{X}} \|h^l(\mathbf{x}) - h_0^l(\mathbf{x})\|_2 = \widetilde{O}\left(\frac{L^2 R}{\sqrt{m}}\right).$$

This implies that the network is $\widetilde{O}(L^2)$-Lipschitz near initialization. More details can be found in Lemma 16 and its proof.

### 4.3 Optimal rates on NTK-separable data

In this section, we apply our general analysis to NTK-separable data [Ji and Telgarsky, 2020, Nitanda et al., 2020, Chen et al., 2021, Taheri and Thrampoulidis, 2024, Deora et al., 2023], and obtain the optimal rates.

**Assumption 3.** *There exists $\gamma > 0$ and a collection of matrices $\mathbf{W}_* = \{\mathbf{W}_*^1, \cdots, \mathbf{W}_*^L\}$ satisfying $\sum_{l=1}^L \|\mathbf{W}_*^l\|_F^2 = 1$, such that*

$$y_i \left\langle \mathbf{W}_*, \frac{\partial f_{\mathbf{W}(0)}(\mathbf{x}_i)}{\partial \mathbf{W}(0)} \right\rangle \geq \gamma, \quad i \in [n].$$

This means that the dataset is separable by the NTK feature at initialization with a margin $\gamma$. Nitanda et al. [2020] pointed out that this assumption is weaker than positive eigenvalues of NTK Gram matrix, which has been widely used in the literature [Du et al., 2019b,a, Arora et al., 2019a]. With the above assumption, we have the following optimal risk bound on NTK separable data. The proof is given in Appendix D.

**Theorem 3.** *Let Assumptions 1, 2, 3 hold. If $m \gtrsim L^{16} d (\log m)^5 \log(nL/\delta)(\log T)^8/\gamma^8, \eta \leq 4/(5L), \eta T \asymp n$, then with probability at least $1 - \delta$, we have*

$$\frac{1}{T} \sum_{t=0}^{T-1} \mathcal{L}(\mathbf{W}(t)) = \widetilde{O}\left(\frac{L^6 (\log T)^2}{n \gamma^2}\right).$$

*Remark* 4 (Proof sketch). To apply the result in Theorem 2, we need to estimate $F(\overline{\mathbf{W}})$, for which it suffices to bound $\mathcal{L}(\overline{\mathbf{W}})$ and $G = \sup_z \ell(y f_{\overline{\mathbf{W}}}(\mathbf{x}))$. For the first part, we control it by $\mathcal{L}_S(\overline{\mathbf{W}})$ using Bernstein inequality (Eq.(14)). Let $\overline{\mathbf{W}} = \mathbf{W}(0) + 2\log T \mathbf{W}_*/\gamma$, plugging into (4) obtains $\ell(y_i f_{\overline{\mathbf{W}}}(\mathbf{x}_i)) \le 1/T$ and further $\mathcal{L}_S(\overline{\mathbf{W}}) \le 1/T$, implying $F_S(\overline{\mathbf{W}}) = \widetilde{O}(1/\gamma^2)$. In order to control $\ell(y f_{\overline{\mathbf{W}}}(\mathbf{x}))$, we leverage the $\widetilde{O}(L^2)$-Lipschitzness of $f_{\mathbf{W}}(\mathbf{x})$. Indeed, for any $\mathbf{x} \in \mathcal{X}$, there holds

$$|f_{\overline{\mathbf{W}}}(\mathbf{x})| \le |f_{\mathbf{W}(0)}(\mathbf{x})| + |f_{\overline{\mathbf{W}}}(\mathbf{x}) - f_{\mathbf{W}(0)}(\mathbf{x})| = \widetilde{O}\left(\frac{L^2}{\gamma}\right).$$

It then follows that $G = \widetilde{O}\left(\frac{L^2}{\gamma}\right)$ and $F(\overline{\mathbf{W}}) = \widetilde{O}\left(\frac{(\log T)^2 L^2}{\gamma^2}\right)$.

*Remark* 5 (Discussion on optimization error). Under Assumption 3, Theorem 3.3 and Proposition 4.2 of Chen et al. [2021] show that when the network width satisfies $m = \widetilde{\Omega}(L^{22}/\gamma^8)$, the training error is of the order $\widetilde{O}(1/T)$. We achieve the same guarantee under a significantly milder width condition of $m = \widetilde{\Omega}(L^{16}/\gamma^8)$. This improvement is enabled by two key technical advances: a sharper bound for (4), and a tighter estimate of the iterate distance $\|\mathbf{W}(t) - \overline{\mathbf{W}}\|_F$. Specifically, we improve the bound in (4) by a factor of $L^{1/3}$, and show that $\|\mathbf{W}(t) - \overline{\mathbf{W}}\|_F = \widetilde{O}(1/\gamma)$, improving upon the previous $\widetilde{O}(\sqrt{L}/\gamma)$ bound. Together, these refinements reduce the required network width by a factor of $L^6$.

*Remark* 6 (Comparison). Under similar assumptions, Ji and Telgarsky [2020] derived the bound $\widetilde{O}\left(\frac{1}{\gamma^2\sqrt{n}}\right)$ for shallow networks, which was recently improved to $\widetilde{O}(\frac{1}{\gamma^2 n})$ based on an improved control of the Rademacher complexity [Lei et al., 2026]. For deep ReLU networks, Chen et al. [2021] developed the bound of the order $\widetilde{O}\left(\frac{e^{O(L)}}{\gamma}\sqrt{\frac{m}{n}}\right)$ via Rademacher complexity [Bartlett et al., 2017], suffering from exponential depth dependence, explicit width requirement and suboptimal $\sqrt{1/n}$ scaling. Taheri et al. [2025] improved the result to $\widetilde{O}\left(\frac{e^{O(L)}}{\gamma^2 n}\right)$ for deep networks. The dependence on $n, \gamma$ is optimal up to a logarithmic factor [Shamir, 2021, Schliserman and Koren, 2023]. However, their results require smooth activations and exponential width in $L$. Our rate is almost-optimal and enjoys polynomial dependence over the network depth. Furthermore, our bound holds under the overparameterization $\widetilde{\Omega}(1/\gamma^8)$, matching the requirement in Ji and Telgarsky [2020], Chen et al. [2021]. This is much better than $1/\gamma^{6L+4}$ in Taheri et al. [2025]. To the best of our knowledge, these are the best generalization bound and width condition for deep neural networks.

## 5 Experiments

In this section, we make some experimental verifications to support our theoretical analysis. Our excess risk analysis in Theorem 3 imposes an NTK separability assumption, which has been validated in the literature. For example, [Ji and Telgarsky, 2020] demonstrates that Assumption 3 holds for a noisy 2-XOR distribution, where the dataset is structured as follows:

$$(x_1, x_2, y, \dots, x_d) \in \left\{\left(\frac{1}{\sqrt{d-1}}, 0, 1\right), \left(0, \frac{1}{\sqrt{d-1}}, -1\right),\right.$$
$$\left.\left(-\frac{1}{\sqrt{d-1}}, 0, 1\right), \left(0, -\frac{1}{\sqrt{d-1}}, -1\right)\right\} \times \left\{-\frac{1}{\sqrt{d-1}}, \frac{1}{\sqrt{d-1}}\right\}^{d-2}.$$

Here, the factor $\frac{1}{\sqrt{d-1}}$ ensures that $\|x\|_2 = 1$, $\times$ above denotes the Cartesian product, and the label $y$ only depends on the first two coordinates of the input $x$. As shown in Ji and Telgarsky [2020], this dataset satisfies Assumption 3 with $1/\gamma = O(d)$, which implies that our excess risk bound in Theorem 3 becomes $O(d^2/n)$ for this dataset. We conducted numerical experiments and observed that the test error decays linearly with $d^2/n$. The population loss for the test error is computed over all $2^d$ points in the distribution.

**Settings** We train two-layer ReLU networks by gradient descent on noisy 2-XOR data. We fix the width $m = 128, T = 500, \eta = 0.1$. We have conducted two experiments. With a fixed dimension $d = 6$, we vary the sample size $n$. The results are presented in Figure 1a. With a fixed sample size $n = 64$, we vary the dimension $d$ and the corresponding table is provided in Figure 1b.

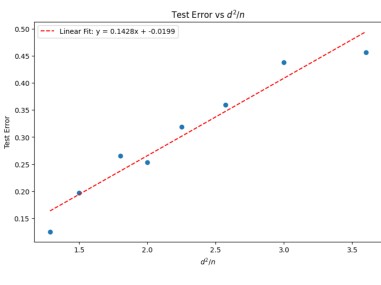
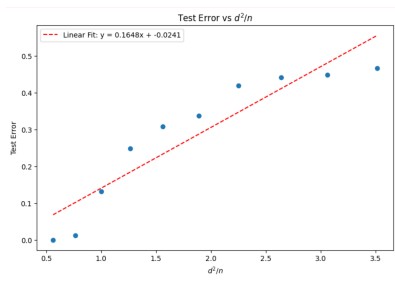

| (a) Test error for different $n$ | (b) Test error for different $d$ |

In both experiments, we observe that the test error is of the order $d^2/n$ (approximately $0.15d^2/n$). This shows the consistency between our excess risk bounds in Theorem 3 and experimental results. We conducted the experiments on Google Colab. A simple demonstration reproducing our numerical experiments is available as a Google Colab notebook at: `https://github.com/YuanfanLi2233/nips2025-optimal`.

## 6 Conclusion and Future Work

In this paper, we present optimization and generalization analysis of gradient descent-trained deep ReLU networks for classification tasks. We explore the optimization error of $F_S(\overline{\mathbf{W}})/(\eta T)$ under a milder overparameterization requirement than before. We establish sharper bound of Rademacher complexity and Lipschtiz constant for neural networks. This helps to derive generalization bound of order $\widetilde{O}(F(\overline{\mathbf{W}})/n)$. For NTK-separable data with a margin $\gamma$, our methods lead to the optimal rate of $\widetilde{O}(1/(n\gamma^2))$. We improve the existing analysis and require less overparameterization than previous works.

There remain several interesting questions for future works. First, it is an interesting question to extend our methods to SGD. Second, while we establish polynomial Lipschitz constants near initialization, investigating whether similar bounds hold far from initialization would deepen our theoretical understanding. Finally, we only consider fully-connected neural networks. It is interesting to study the generalization analysis of networks with other architectures, such as CNNs and Resnets [Du et al., 2019b].

## Acknowledgement

The authors are grateful to the anonymous reviewers for their thoughtful comments and constructive suggestions. The work of Yuanfan Li and Zheng-Chu Guo is partially supported by the National Natural Science Foundation of China (Grants No. 12271473 and U21A20426). The work by Yunwen Lei is partially supported by the Research Grants Council of Hong Kong [Project No. 17302624]. Yiming's work is partially supported by Australian Research Council (ARC) DP250101359.

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

# A Technical Lemmas

We define the diagonal sign matrix $\Sigma^l(\mathbf{x})$ with $l \in [L]$ by

$$\Sigma^l(\mathbf{x}) = \text{diag}\{\mathbb{I}\{\langle \mathbf{w}_r^l, h^{l-1}(\mathbf{x})\rangle \geq 0\}\} \in \mathbb{R}^{m \times m}. \tag{10}$$

Then the deep ReLU network has the following matrix product representation:

$$f_{\mathbf{W}}(\mathbf{x}) = \mathbf{a}^\top \sqrt{\frac{2}{m}}\Sigma^L(\mathbf{x})\mathbf{W}^L \cdots \sqrt{\frac{2}{m}}\Sigma^1(\mathbf{x})\mathbf{W}^1\mathbf{x}, \tag{11}$$

together with the presentation of $h^l(\mathbf{x})$:

$$h^l(\mathbf{x}) = \sqrt{\frac{2}{m}}\Sigma^l(\mathbf{x})\mathbf{W}^l \cdots \sqrt{\frac{2}{m}}\Sigma^1(\mathbf{x})\mathbf{W}^1\mathbf{x}, \quad l \in [L]. \tag{12}$$

We further define $\mathbf{G}_l^l(\mathbf{x}) = \sqrt{2/m}\Sigma^l(\mathbf{x})$ and

$$\mathbf{G}_b^a(\mathbf{x}) = \sqrt{\frac{2}{m}}\Sigma^b(\mathbf{x})\mathbf{W}^b \cdots \sqrt{\frac{2}{m}}\Sigma^a(\mathbf{x}), \quad 1 \leq a \leq b \leq L, \tag{13}$$

from which we can rewrite $f_{\mathbf{W}}(\mathbf{x})$ as

$$f_{\mathbf{W}}(\mathbf{x}) = \mathbf{a}^\top \mathbf{G}_L^l(\mathbf{x})\mathbf{W}^l h^{l-1}(\mathbf{x}) = \langle (\mathbf{G}_L^l(\mathbf{x}))^\top \mathbf{a}(h^{l-1}(\mathbf{x}))^\top, \mathbf{W}^l\rangle.$$

Hence, for $l \in [L]$, we have

$$\frac{\partial f_{\mathbf{W}}(\mathbf{x})}{\partial \mathbf{W}^l} = (\mathbf{G}_L^l(\mathbf{x}))^\top \mathbf{a}(h^{l-1}(\mathbf{x}))^\top.$$

Similarly, we define

$$\mathbf{H}_b^a(\mathbf{x}) = \sqrt{\frac{2}{m}}\Sigma^b(\mathbf{x})\mathbf{W}^b \cdots \sqrt{\frac{2}{m}}\Sigma^a(\mathbf{x})\mathbf{W}^a, \quad 2 \leq a \leq b \leq L. \tag{14}$$

We denote $\Sigma_0^l(\mathbf{x}), h_0^l(\mathbf{x}), \mathbf{G}_{b,0}^a(\mathbf{x}), \mathbf{H}_{b,0}^a(\mathbf{x})$ as (10), (12), (13) and (14) with $\mathbf{W} = \mathbf{W}(0)$.

## A.1 Properties of the Initialization

Given a set of $N$ points on the sphere $K = \{x^1, \cdots, x^N\}$. We provide general results for any finite set $K$, then it can be applied to specific choices of $K$, for example, the training dataset $S_1 = \{\mathbf{x}_1, \cdots, \mathbf{x}_n\}$.

**Lemma 2** (Theorem 4.4.5 in Vershynin [2018]). *With probability at least $1 - L\exp(-Cm)$ over the random choice of $\mathbf{W}(0)$, there exists an absolute constant $c_0 > 1$ such that for any $l \in [L]$, there holds*

$$\|\mathbf{W}^l(0)\|_2 \leq c_0\sqrt{m}. \tag{15}$$

For a sub-exponential random variable $X$, its sub-exponential norm is defined as follows:

$$\|X\|_{\phi_1} = \inf\{t > 0 : \mathbb{E}\exp(|X|/t) \leq 2\}.$$

$X - \mathbb{E}X$ is sub-exponential too, satisfying

$$\|X - \mathbb{E}X\|_{\phi_1} \leq 2\|X\|_{\phi_1}. \tag{16}$$

If $Y$ is a sub-gaussian random variable, we define the sub-gaussian norm of $Y$ by

$$\|Y\|_{\phi_2} = \inf\{t > 0 : \mathbb{E}\exp(Y^2/t^2) \leq 2\}.$$

Suppose $Y \sim \mathcal{N}(0, r^2)$, then $\sigma(Y)$ is also sub-guassian and we have

$$\|\sigma(Y)\|_{\phi_2} \leq \|Y\|_{\phi_2} \leq Cr. \tag{17}$$

We have the following lemma:

**Lemma 3** (Lemma 2.7.6 in Vershynin [2018]). *A random variable $X$ is sub-gaussian if and only if $X^2$ is sub-exponential. Moreover,*

$$\|X\|_{\phi_2}^2 = \|X^2\|_{\phi_1}.$$

Now we introduce Bernstein inequality with respect to $\|\cdot\|_{\phi_1}$,

**Lemma 4** (Theorem 2.8.2 in Vershynin [2018]). *Let $X_1, \cdots, X_m$ be independent, mean zero, sub-exponential random variables, and $d = (d_1, \cdots, d_m) \in \mathbb{R}^m$, $K \geq \max_r \|X_r\|_{\phi_1}$. Then for every $t \geq 0$, we have*

$$\mathbb{P}\left(\left|\sum_{r=1}^m d_r X_r\right| \geq t\right) \leq 2\exp\left[-c\min\left(\frac{t^2}{K^2\|d\|_2^2}, \frac{t}{K\|d\|_\infty}\right)\right].$$

*for some absolute constant $c$.*

We introduce the following technical lemma related to the conditional expectation of Gaussian indicator function.

**Lemma 5.** *Suppose $\mathbf{w}$ is a $m$-dim Gaussian random vector with distribution $\mathcal{N}(0, \mathbf{I})$. Let $\mathbf{c} \neq 0$, $\mathbf{b}$ be two given vectors of $m$-dim. Then we have the following property*

$$\mathbb{E}[\mathbb{I}\{\langle \mathbf{w}, \mathbf{c}\rangle \geq 0\}\langle \mathbf{w}, \mathbf{b}\rangle^2] = \frac{\|\mathbf{b}\|_2^2}{2}.$$

*Proof.* Let $\mathbf{u} = \langle \mathbf{w}, \mathbf{c}\rangle, \mathbf{v} = \langle \mathbf{w}, \mathbf{b}\rangle$. Then $\mathbf{u} \sim \mathcal{N}(0, \|\mathbf{c}\|_2^2), \mathbf{v} \sim \mathcal{N}(0, \|\mathbf{b}\|_2^2)$. We decompose $\mathbf{v}$ into a component dependent on $\mathbf{u}$ and an independent residual $\mathbf{z}$:

$$\mathbf{v} = \frac{\mathrm{Cov}(\mathbf{u}, \mathbf{v})}{\mathrm{Var}(\mathbf{u})}\mathbf{u} + \mathbf{z} = \frac{\langle \mathbf{c}, \mathbf{b}\rangle}{\|\mathbf{c}\|_2^2}\mathbf{u} + \mathbf{z},$$

where $\mathbf{z} \sim \mathcal{N}\left(0, \|\mathbf{b}\|_2^2 - \frac{\langle \mathbf{c}, \mathbf{b}\rangle^2}{\|\mathbf{c}\|_2^2}\right)$ is independent of $\mathbf{u}$. Hence, we have

$$\mathbb{E}[\mathbb{I}\{\langle \mathbf{w}, \mathbf{c}\rangle \geq 0\}\langle \mathbf{w}, \mathbf{b}\rangle^2] = \mathbb{E}[\mathbb{I}\{\mathbf{u} \geq 0\}\mathbf{v}^2]$$

$$= \frac{\langle \mathbf{c}, \mathbf{b}\rangle^2}{\|\mathbf{c}\|_2^4}\mathbb{E}[\mathbb{I}\{\mathbf{u} \geq 0\}\mathbf{u}^2] + \frac{2\langle \mathbf{c}, \mathbf{b}\rangle}{\|\mathbf{c}\|_2^2}\mathbb{E}[\mathbb{I}\{\mathbf{u} \geq 0\}\mathbf{u}\mathbf{z}] + \mathbb{E}[\mathbb{I}\{\mathbf{u} \geq 0\}\mathbf{z}^2]$$

$$= \frac{\langle \mathbf{c}, \mathbf{b}\rangle^2}{2\|\mathbf{c}\|_2^2} + 0 + \frac{1}{2}\mathbb{E}\mathbf{z}^2$$

$$= \frac{\langle \mathbf{c}, \mathbf{b}\rangle^2}{2\|\mathbf{c}\|_2^2} + \frac{1}{2}\cdot\left(\|\mathbf{b}\|_2^2 - \frac{\langle \mathbf{c}, \mathbf{b}\rangle^2}{\|\mathbf{c}\|_2^2}\right) = \frac{\|\mathbf{b}\|_2^2}{2},$$

where the second equality is due to $\mathbb{E}[\mathbb{I}\{\mathbf{u} \geq 0\}\mathbf{u}^2] = \frac{\|\mathbf{c}\|_2^2}{2}$ and the independence of $\mathbf{u}, \mathbf{z}$. The third equality follows from the distribution of $\mathbf{z}$. The proof is completed. $\square$

The following lemma studies the output of initialization at each layer.

**Lemma 6.** *For any $\delta > 0$, if $m \gtrsim L^2\log(NL/\delta)$, then with probability at least $1 - \delta$,*

$$\|h_0^l(\mathbf{x}^i)\|_2 \in \left[\sqrt{\frac{2}{3}}, \sqrt{\frac{4}{3}}\right], \quad i = 1, \cdots, N \quad and \quad l = 1, \cdots, L.$$

*Proof.* This result directly follows Corollary A.2 in Zou et al. [2018], and we give the proof here for completeness. Note that for $1 \leq i \leq N, 1 \leq l \leq L$,

$$\left\|h_0^l(\mathbf{x}^i)\right\|_2^2 = \left\|\sqrt{\frac{2}{m}}\sigma\left(\mathbf{W}^l(0)h_0^{l-1}(\mathbf{x}^i)\right)\right\|_2^2 = \frac{1}{m}\sum_{r=1}^m 2\sigma^2(\langle \mathbf{w}_r^l(0), h_0^{l-1}(\mathbf{x}^i)\rangle).$$

Condition on $h_0^{l-1}(\mathbf{x}^i)$, we have $\langle \mathbf{w}_j^l(0), h_0^{l-1}(\mathbf{x}^i)\rangle \sim \mathcal{N}(0, \|h_0^{l-1}(\mathbf{x}^i)\|_2^2)$, hence

$$\mathbb{E}2\sigma^2(\langle \mathbf{w}_j^l(0), h_0^{l-1}(\mathbf{x}^i)\rangle) = 2\left\|h_0^{l-1}(\mathbf{x}^i)\right\|_2^2\mathbb{E}_{u\sim\mathcal{N}(0,1)}\sigma^2(u) = \left\|h_0^{l-1}(\mathbf{x}^i)\right\|_2^2 \quad (18)$$

By (16) and Lemma 3, we have

$$\|2\sigma^2(\langle \mathbf{w}_j^l(0), h_0^{l-1}(\mathbf{x}^i)\rangle) - \|h_0^{l-1}(\mathbf{x}^i)\|_2^2\|_{\phi_1}$$

$$\leq 2\|2\sigma^2(\langle \mathbf{w}_j^l(0), h_0^{l-1}(\mathbf{x}^i)\rangle)\|_{\phi_1} \leq C\|\sigma(\langle \mathbf{w}_j^l(0), h_0^{l-1}(\mathbf{x}^i)\rangle)\|_{\phi_2}^2 \leq C\|h_0^{l-1}(\mathbf{x}^i)\|_2^2,$$

where the last inequality is due to (17). Let $X_r = 2\sigma^2(\langle \mathbf{w}_r^l(0), h_0^{l-1}(\mathbf{x}^i)\rangle) - \|h_0^{l-1}(\mathbf{x}^i)\|_2^2, d_r = 1/m, K = C\left\|h_0^{l-1}(\mathbf{x}^i)\right\|_2^2$ and apply Lemma 4. We have for any $0 \leq t \leq 1$,

$$\mathbb{P}(|\|h_0^l(\mathbf{x}^i)\|_2^2 - \|h_0^{l-1}(\mathbf{x}^i)\|_2^2| \leq Ct\|h_0^{l-1}(\mathbf{x}^i)\|_2^2|h_0^{l-1}(\mathbf{x}^i))$$

$$=\mathbb{P}\left(\left|\frac{1}{m}\sum_{r=1}^m (\mathbb{E}2\sigma^2(\langle w_r^l(0), h_0^{l-1}(\mathbf{x}^i)\rangle) - \|h_0^{l-1}(\mathbf{x}^i)\|_2^2\right| \leq Ct\|h_0^{l-1}(\mathbf{x}^i)\|_2^2|h_0^{l-1}(\mathbf{x}^i)\right)$$

$$\geq 1 - 2\exp(-Cm\min\{t^2, t\}) = 1 - 2\exp(-Cmt^2).$$

Taking union bounds over $i, l$, there holds for any $1 \leq i \leq N$ and $1 \leq l \leq L$,

$$\mathbb{P}(|\|h_0^l(\mathbf{x}^i)\|_2^2 - \|h_0^{l-1}(\mathbf{x}^i)\|_2^2| \leq Ct\|h_0^{l-1}(\mathbf{x}^i)\|_2^2) \geq 1 - 2nL\exp(-Cmt^2).$$

Since $m \gtrsim L^2\log(NL/\delta)$, let $t = \sqrt{\log(NL/\delta)/m}$, we have with probability at least $1 - \delta$, there holds

$$|\|h_0^l(\mathbf{x}^i)\|_2^2 - \|h_0^{l-1}(\mathbf{x}^i)\|_2^2| \leq C\sqrt{\frac{\log(NL/\delta)}{m}}\|h_0^{l-1}(\mathbf{x}^i)\|_2^2, \quad 1 \leq i \leq N, 1 \leq l \leq L. \quad (19)$$

Now we show the following inequality holds with probability at least $1 - \delta$,

$$|\|h_0^l(\mathbf{x}^i)\|_2^2 - 1| \leq \frac{4lC}{3}\sqrt{\frac{\log(NL/\delta)}{m}} \leq \frac{1}{3}. \quad (20)$$

When $l = 0$, it is true. If (20) holds for $l \in [L-1]$, then $\|h_0^l(\mathbf{x}^i)\|_2^2 \leq 4/3$. Combined with (19), we have with probability at least $1 - \delta$,

$$|\|h_0^{l+1}(\mathbf{x}^i)\|_2^2 - 1| \leq |\|h_0^l(\mathbf{x}^i)\|_2^2 - 1| + |\|h_0^{l+1}(\mathbf{x}^i)\|_2^2 - \|h_0^l(\mathbf{x}^i)\|_2^2|$$

$$\leq \frac{4lC}{3}\sqrt{\frac{\log(NL/\delta)}{m}} + C\sqrt{\frac{\log(NL/\delta)}{m}}\|h_0^l(\mathbf{x}^i)\|_2^2$$

$$\leq \frac{4(l+1)C}{3}\sqrt{\frac{\log(NL/\delta)}{m}} \leq \frac{1}{3}.$$

Hence, (20) holds for all $i \in [N], l \in [L]$, which implies the lemma. $\qquad\square$

**Lemma 7.** *For any* $\mathbf{x} \in \mathcal{X}$, *we have*

$$\mathbf{a}^\top\Sigma_0^L(\mathbf{x})\mathbf{W}^L(0) = 0 \quad and \quad \frac{\partial f_{\mathbf{W}(0)}(\mathbf{x})}{\partial\mathbf{W}^l(0)} = 0, \quad l \in [L-1].$$

*Proof.* Note the $r$-th row of $\Sigma_0^L(\mathbf{x})\mathbf{W}^L(0)$ is $\mathbb{I}\{\langle \mathbf{w}_r^L(0), h_0^{L-1}(\mathbf{x})\rangle \geq 0\}\mathbf{w}_r^L(0)$. Since $a_r = -a_{r+\frac{m}{2}}$ and $\mathbf{w}_r^L(0) = \mathbf{w}_{r+\frac{m}{2}}^L(0)$ for all $r \in [\frac{m}{2}]$, we have

$$\mathbf{a}^\top\Sigma_0^L(\mathbf{x})\mathbf{W}^L(0)$$

$$=\sum_{r=1}^m a_r\mathbb{I}\{\langle \mathbf{w}_r^L(0), h_0^{L-1}(\mathbf{x})\rangle \geq 0\}\mathbf{w}_r^L(0)$$

$$=\sum_{r=1}^{\frac{m}{2}} a_r\mathbb{I}\{\langle \mathbf{w}_r^L(0), h_0^{L-1}(\mathbf{x})\rangle \geq 0\}\mathbf{w}_r^L(0) + \sum_{r=1}^{\frac{m}{2}} a_{r+\frac{m}{2}}\mathbb{I}\{\langle \mathbf{w}_{r+\frac{m}{2}}^L(0), h_0^{L-1}(\mathbf{x})\rangle \geq 0\}\mathbf{w}_r^L(0)$$

$$=\sum_{r=1}^{\frac{m}{2}} (a_r - a_r)\mathbb{I}\{\langle \mathbf{w}_r^L(0), h_0^{L-1}(\mathbf{x})\rangle \geq 0\}\mathbf{w}_r^L(0) = 0.$$

It then follows that for all $l \in [L-1]$,

$$\left(\frac{\partial f_{\mathbf{W}(0)}(\mathbf{x})}{\partial\mathbf{W}^l(0)}\right)^\top = h_0^{l-1}(\mathbf{x})\mathbf{a}^\top(\mathbf{G}_{L,0}^l(\mathbf{x}))^\top = h_0^{l-1}(\mathbf{x})\mathbf{a}^\top\sqrt{\frac{2}{m}}\Sigma_0^L(\mathbf{x})\mathbf{W}^L(0)\cdots\sqrt{\frac{2}{m}}\Sigma_0^l(\mathbf{x}) = 0.$$

Hence, the proof is completed. $\qquad\square$

**Lemma 8.** *Suppose $m \gtrsim L^2 \log(NL/\delta)$, then with probability at least $1 - \delta$, for all $i \in [N]$,*

$$\left\| \frac{\partial f_{\mathbf{W}(0)}(\mathbf{x}^i)}{\partial \mathbf{W}^L(0)} \right\|_F \leq \sqrt{2}.$$

*Proof.* By Hoeffding inequality, condition on $h_0^{L-1}(\mathbf{x}^i)$, with probability at least $1 - \delta$, there holds

$$\frac{1}{m} \mathbf{a}^\top \Sigma_0^L(\mathbf{x}^i) \Sigma_0^L(\mathbf{x}^i) \mathbf{a} = \frac{1}{m} \sum_{j=1}^m \mathbb{I}\{\langle \mathbf{w}_j^L(0), h_0^{L-1}(\mathbf{x}^i) \rangle \geq 0\} \leq \frac{1}{2} + C\sqrt{\frac{\log(N/\delta)}{m}} \leq \frac{3}{4}.$$

Combined with Lemma 6, we have

$$\left\| \frac{\partial f_{\mathbf{W}(0)}(\mathbf{x}^i)}{\partial \mathbf{W}^L(0)} \right\|_F^2 = \frac{2}{m} \|h_0^{L-1}(\mathbf{x}^i)\|_2^2 \mathbf{a}^\top \Sigma_0^L(\mathbf{x}^i) \Sigma_0^L(\mathbf{x}^i) \mathbf{a} \leq 2 \cdot \frac{4}{3} \cdot \frac{3}{4} = 2.$$

The proof is completed. $\qquad\qquad\square$

Let $\Sigma_1, \Sigma_2 \in \mathbb{R}^{m \times m}$ be two diagonal matrices with entries in $\{0, 1\}$.

**Lemma 9.** *Suppose $m \gtrsim L^2 \log(NL/\delta), s \lesssim m/(L^2 \log m)$, then with probability at least $1 - \delta$ we have for all $i \in [N], 2 \leq a \leq b \leq L$,*

$$\sup_{\|\Sigma_1\|_0 \leq s} \|\mathbf{H}_{b,0}^a(\mathbf{x}^i)\Sigma_1\|_2 \lesssim 1. \tag{21}$$

*Proof.* We need to prove that for any $v \in S^{m-1}$, there holds

$$\|\mathbf{H}_{b,0}^a(\mathbf{x}^i)\Sigma_1 v\|_2 \lesssim 1. \tag{22}$$

Note that $\|\Sigma_1 v\|_0 \leq s$ and $\|\Sigma_1 v\|_2 \leq 1$. Let $\mathcal{P} = \{v \in S^{m-1} : \|v\|_0 \leq s\}$. We only need to prove that the following inequality holds with probability at least $1 - \delta$:

$$\sup_{v \in \mathcal{P}} \|\mathbf{H}_{b,0}^a(\mathbf{x}^i) v\|_2 \lesssim 1. \tag{23}$$

Let $\mathcal{S}$ be a subspace of $S^{m-1}$ that has at most $s$ non-zero coordinates. For such a subspace, we choose a $1/2$-cover of it and denote this cover by $\mathcal{Q}$. By Lemma 4.2.13 in Vershynin [2018],

$$|\mathcal{Q}| \leq 5^s.$$

The number of such subspaces is $M = \binom{m}{s}$. We denote all subspaces by $\mathcal{S}_1, \cdots, \mathcal{S}_M$, and the corresponding covers $\mathcal{Q}_1, \cdots, \mathcal{Q}_M$. Let $\bigcup \mathcal{Q} = \{v_1, \cdots, v_{M'}\}$ with $M' \leq \binom{m}{s} 5^s$. We first prove that (23) is true for all $v_j$, then it holds simultaneously for all elements in $\mathcal{P}$.

For a unit vector $v$, we define

$$v^l(\mathbf{x}_i) = \mathbf{H}_{l,0}^a(\mathbf{x}^i) v, \quad a \leq l \leq b.$$

and $v^{a-1}(\mathbf{x}^i) = v$. Note that condition on $h_0^{l-1}(\mathbf{x}^i), v^{l-1}(\mathbf{x}^i)$, we take expectation over $\mathbf{w}_r^l(0)$, applying Lemma 5 implies that

$$\mathbb{E}[2\mathbb{I}\{\langle \mathbf{w}_r^l(0), h_0^{l-1}(\mathbf{x}^i) \rangle \geq 0\}(\langle \mathbf{w}_r^l(0), v^{l-1}(\mathbf{x}^i) \rangle)^2] = \|v^{l-1}(\mathbf{x}^i)\|_2^2.$$

It then follows that

$$\|v^l(\mathbf{x}^i)\|_2^2 = \|\mathbf{H}_{l,0}^a(\mathbf{x}^i) v\|_2^2 = \left\| \sqrt{\frac{2}{m}} \Sigma_0^l(\mathbf{x}^i) \mathbf{W}^l(0) v^{l-1}(\mathbf{x}^i) \right\|_2^2$$

$$= \frac{1}{m} \sum_{r=1}^m \mathbb{E} 2\mathbb{I}\{\langle \mathbf{w}_r^l(0), h_0^{l-1}(\mathbf{x}^i) \rangle \geq 0\}(\langle \mathbf{w}_r^l(0), v^{l-1}(\mathbf{x}^i) \rangle)^2 = \|v^{l-1}(\mathbf{x}^i)\|_2^2.$$

Similar to the proof of Lemma 6, for every $v_j, 1 \leq j \leq M'$, we apply Lemma 4 to get

$$\mathbb{P}\left( |\|v_j^l(\mathbf{x}^i)\|_2^2 - \|v_j^{l-1}(\mathbf{x}^i)\|_2^2| \leq \frac{C\|v_j^{l-1}(\mathbf{x}^i)\|_2^2}{L} |v_j^{l-1}(\mathbf{x}^i) \right) \geq 1 - 2\exp\left( -Cm \min\left\{ \frac{1}{L^2}, \frac{1}{L} \right\} \right).$$

Taking the union bounds for all $j, l, i$ yields

$$\mathbb{P}\left(|\|v_j^l(\mathbf{x}^i)\|_2^2 - \|v_j^{l-1}(\mathbf{x}^i)\|_2^2| \leq \frac{C\|v_j^{l-1}(\mathbf{x}^i)\|_2^2}{L}\right)$$

$$\geq 1 - 2 \cdot \binom{m}{s} 5^s NL \exp\left(\frac{-Cm}{L^2}\right)$$

$$\geq 1 - 2 \cdot (5em)^s NL \exp\left(\frac{-Cm}{L^2}\right)$$

$$= 1 - \exp\left[\log \delta + s \log m + s \log(5e) + \log\left(\frac{2NL}{\delta}\right) - \frac{C}{L^2}m\right] \geq 1 - \delta,$$

where we have used $\binom{m}{s} \leq (em/s)^s \leq (em)^s$ in the second inequality and the last inequality is due to $m \gtrsim L^2 \log(NL/\delta), s \lesssim m/(L^2 \log m)$. Hence, we have with probability at least $1 - \delta$,

$$\|v_j^l(\mathbf{x}^i)\|_2^2 \leq \left(1 + \frac{C}{L}\right)\|v_j^{l-1}(\mathbf{x}^i)\|_2^2 \leq \left(1 + \frac{C}{L}\right)^L \|v_j\|_2^2 \lesssim 1.$$

For any unit vector $v$ with $\|v\|_0 \leq s$, consider the subspace $S$ containing it and the corresponding $1/2$- cover $\mathcal{Q}$. There exists a unit vector $v_j, j \in [M'], \|v - v_j\|_0 \leq s$ and $\|v - v_j\|_2 \leq 1/2$. Thus,

$$\|\mathbf{H}_{b,0}^a(\mathbf{x}^i)v\|_2$$

$$\leq \|\mathbf{H}_{b,0}^a(\mathbf{x}^i)v_j\|_2 + \|\mathbf{H}_{b,0}^a(\mathbf{x}^i)(v - v_j)\|_2$$

$$= \|v_j^b(\mathbf{x}^i)\|_2 + \|v - v_j\|_2 \left\|\mathbf{H}_{b,0}^a(\mathbf{x}^i)\frac{v - v_j}{\|v - v_j\|_2}\right\|_2$$

$$\lesssim 1 + \|v - v_j\|_2 \sup_{v \in \mathcal{P}} \|\mathbf{H}_{b,0}^a(\mathbf{x}^i)v\|_2 \leq 1 + \frac{1}{2} \sup_{v \in \mathcal{P}} \|\mathbf{H}_{b,0}^a(\mathbf{x}^i)v\|_2.$$

Taking $\sup$ to the both sides yields

$$\sup_{v \in \mathcal{P}} \|\mathbf{H}_{b,0}^a(\mathbf{x}^i)v\|_2 \lesssim 1 + \frac{1}{2} \sup_{v \in \mathcal{P}} \|\mathbf{H}_{b,0}^a(\mathbf{x}^i)v\|_2,$$

which implies $\sup_{v \in \mathcal{P}} \|\mathbf{H}_{b,0}^a(\mathbf{x}^i)v\|_2 \lesssim 1$. Hence (23) holds and the proof is completed. $\qquad\square$

*Remark* 7. Our proofs are inspired by Lemma A.9 in Zou et al. [2018], which establishes the estimates under the condition $s \gtrsim \log(NL/\delta)$. However, we eliminate this assumption via a more refined analysis.

**Lemma 10.** *Suppose $m \gtrsim L^2 \log(NL/\delta)$, then with probability at least $1 - \delta$, we have for all $i \in [N], 2 \leq a \leq b \leq L$,*

$$\|\mathbf{H}_{b,0}^a(\mathbf{x}^i)\|_2 \lesssim L\sqrt{\log m}.$$

Although the left-hand side of above inequality could be the production of $L$ terms, it is bounded by $\widetilde{O}(L)$. This lemma shows that the introduction of ReLU activation can avoid exponential explosion.

*Proof.* For any unit vector $v$, we decompose it as $v = v_1 + \cdots + v_q$, where $v_j, j \in [q]$ are all $s$-sparse vectors on different coordinates. Therefore,

$$\|v\|_2^2 = \sum_{j=1}^q \|v_j\|_2^2.$$

Here we choose $s \asymp m/(L^2 \log m)$, then $q \lesssim m/s \lesssim L^2 \log m$. Applying Lemma 9, we have

$$\|\mathbf{H}_{b,0}^a(\mathbf{x}^i)v\|_2 = \left\|\sum_{j=1}^q \mathbf{H}_{b,0}^a(\mathbf{x}^i)v_j\right\|_2 \leq \sum_{j=1}^q \|\mathbf{H}_{b,0}^a(\mathbf{x}^i)v_j\|_2 \leq \sqrt{q \sum_{j=1}^q \|\mathbf{H}_{b,0}^a(\mathbf{x}^i)v_j\|_2^2}$$

$$\lesssim \sqrt{q \sum_{j=1}^q \|v_j\|_2^2} = \sqrt{q\|v\|_2^2} = \sqrt{q} \lesssim L\sqrt{\log m}.$$

where we have used Cauchy-Schwartz's inequality in the second inequality. Hence,

$$\|\mathbf{H}_{b,0}^a(\mathbf{x}^i)\|_2 \lesssim L\sqrt{\log m}.$$

The proof is completed. $\qquad\square$

From the above lemma, we know that if $m \gtrsim L^2 \log(NL/\delta) \log m$, then

$$\left\|\mathbf{G}_{b,0}^a(\mathbf{x}^i)\right\|_2 \lesssim L\sqrt{\frac{\log m}{m}}, \quad i \in [N], 1 \le a \le b \le L. \tag{24}$$

*Remark* 8. In Lemma 10, we introduce a useful technique that decomposes the unit vector into sparse components. This approach reduces the covering number from $5^m$ to $5^s\binom{m}{s}$, making it easier for high-probability bounds to hold. A related method appears in Lemma 7.3 of Allen-Zhu et al. [2019], but their width exhibits polynomial dependence on $n$. In contrast, our analysis achieves polylogarithmic width, substantially relaxing the overparameterization requirement.

Using similar techniques we can obtain the following lemma:

**Lemma 11.** *Suppose* $m \gtrsim L^2 \log(NL/\delta)$, *then with probability at least* $1 - \delta$, *for all* $i \in [N], 2 \le a \le b \le L, \|\Sigma_1\|_0, \|\Sigma_2\|_0 \le s \lesssim m/(L^2 \log m)$,

$$\left\|\Sigma_1 \sqrt{\frac{2}{m}}\mathbf{W}^b(0)\mathbf{H}_{b-1,0}^a(\mathbf{x}^i)\Sigma_2\right\|_2 \lesssim \frac{1}{L}.$$

*Proof.* If $s = 0$, the above inequality becomes $0 \lesssim 1/L$, which holds true. Now we assume $s \ge 1$. Similar to Lemma 9, we only need to prove that for any $s$-sparse unit vector $v$ there holds

$$\left\|\Sigma_1 \sqrt{\frac{2}{m}}\mathbf{W}^b(0)\mathbf{H}_{b-1,0}^a(\mathbf{x}^i)v\right\|_2 \lesssim \frac{1}{L}. \tag{25}$$

We use the same notation as in Lemma 9, it then follows that for all $j \in [M'], a \le l \le b, i \in [N]$,

$$\mathbb{P}\left(|\|v_j^l(\mathbf{x}^i)\|_2^2 - \|v_j^{l-1}(\mathbf{x}^i)\|_2^2| \le \frac{C\|v_j^{l-1}(\mathbf{x}^i)\|_2^2}{L}|v_j^{l-1}(\mathbf{x}^i)\right) \ge 1 - 2\exp\left(-Cm\min\left\{\frac{1}{L^2},\frac{1}{L}\right\}\right).$$

For a fixed $\Sigma_1$, we assume $\Sigma_1 = diag\{d_1, \cdots, d_m\}$ with $d_r \in \{0, 1\}, \sum_r d_r \le s, r \in [m]$. We have

$$\left\|\Sigma_1 \sqrt{\frac{2}{m}}\mathbf{W}^b(0)v_j^{b-1}(\mathbf{x}^i)\right\|_2^2 = \sum_{r=1}^m d_r^2 \frac{2}{m}(\langle \mathbf{w}_r^b, v_j^{b-1}(\mathbf{x}^i)\rangle)^2. \tag{26}$$

Condition on $v_j^{b-1}(\mathbf{x}^i)$, there holds

$$\mathbb{E}\frac{2}{m}(\langle \mathbf{w}_r^b(0), v_j^{b-1}(\mathbf{x}^i)\rangle)^2 = \frac{2}{m}\|v_j^{b-1}(\mathbf{x}^i)\|_2^2.$$

Let $X_r = \frac{2}{m}(\langle \mathbf{w}_r^b(0), v_j^{b-1}(\mathbf{x}^i)\rangle)^2 - \frac{2}{m}\|v_j^{b-1}(\mathbf{x}^i)\|_2^2$ and $d = (d_1, \cdots, d_m)$. Then $X_r$ are mean-zero sub-exponential random variables, following similar discussions in Lemma 6, we have $\|X_r\|_{\phi_1} \le \frac{C}{m}\|v_j^{b-1}(\mathbf{x}^i)\|_2^2$. Moreover, $\|d\|_2^2 = \sum_{r=1}^m d_r^2 = \sum_{r=1}^m d_r \le s, \|d\|_\infty = 1$. Applying Lemma 4, we have

$$\mathbb{P}\left(\left|\sum_{r=1}^m d_r\left(\frac{2}{m}(\langle \mathbf{w}_r^b(0), v_j^{b-1}(\mathbf{x}^i)\rangle)^2 - \frac{2}{m}\|v_j^{b-1}(\mathbf{x}^i)\|_2^2\right)\right| \ge t\|v_j^{b-1}(\mathbf{x}^i)\|_2^2|v_j^{b-1}(\mathbf{x}^i)\right)$$
$$\le 2\exp\left[-C\min\left(\frac{t^2m^2}{s}, tm\right)\right]. \tag{27}$$

Choosing $t = 1/L^2$ and note that $s \leq m/(L^2 \log m)$, we have

$$\mathbb{P}\left(\left\|\Sigma_1\sqrt{\frac{2}{m}}\mathbf{W}^b(0)v_j^{b-1}(\mathbf{x}^i)\right\|_2^2 \leq \left(\frac{2s}{m}+\frac{1}{L^2}\right)\|v_j^{b-1}(\mathbf{x}^i)\|_2^2|v_j^{b-1}(\mathbf{x}^i)\right)$$

$$\geq \mathbb{P}\left(\sum_{r=1}^m d_r^2\frac{2}{m}(\langle\mathbf{w}_r^b(0),v_j^{b-1}(\mathbf{x}^i)\rangle)^2 \leq \sum_{r=1}^m \frac{2d_r}{m}\|v_j^{b-1}(\mathbf{x}^i)\|_2^2 + \frac{1}{L^2}\|v_j^{b-1}(\mathbf{x}^i)\|_2^2|v_j^{b-1}(\mathbf{x}^i)\right)$$

$$\geq 1 - 2\exp\left(-\frac{Cm}{L^2}\right),$$

where the first inequality is due to (26) and $\sum_{r=1}^m d_r \leq s$, the last inequality results from (27).

Taking union bounds over all $\mathbf{x}^i, \Sigma_1, v_j, l$, and note that

$$2nL\binom{m}{s}5^s\binom{m}{s}\exp\left(-\frac{Cm}{L^2}\right) \leq \frac{\delta}{2}.$$

We have with probability at least $1 - \delta$,

$$\left\|\Sigma_1\sqrt{\frac{2}{m}}\mathbf{W}^b(0)v_j^{b-1}(\mathbf{x}^i)\right\|_2^2 \leq \left(\frac{2s}{m}+\frac{1}{L^2}\right)\|v_j^{b-1}(\mathbf{x}^i)\|_2^2$$

$$\leq \left(\frac{2s}{m}+\frac{1}{L^2}\right)\left(1+\frac{C}{L}\right)\|v_j^{b-2}(\mathbf{x}^i)\|_2^2 \leq \left(\frac{2s}{m}+\frac{1}{L^2}\right)\left(1+\frac{C}{L}\right)^L\|v_j\|_2^2 \lesssim \frac{1}{L^2}.$$

Hence, we have

$$\left\|\Sigma_1\sqrt{\frac{2}{m}}\mathbf{W}^b(0)\mathbf{H}_{b-1,0}^a(\mathbf{x}^i)v_j\right\|_2 \lesssim \frac{1}{L}.$$

Following same techniques of using $1/2$-cover in the proof of Lemma 9, we can prove (25), and then complete the proof of the lemma. $\square$

Additionally, apply similar methods in Lemma 10 by decomposing the unit vector into $s$-sparse vectors, we have

$$\left\|\Sigma_1\sqrt{\frac{2}{m}}\mathbf{W}_0^b(\mathbf{x}^i)\mathbf{H}_{b-1,0}^a(\mathbf{x}^i)\right\|_2 \lesssim \sqrt{\log m}. \tag{28}$$

*Remark* 9. To analyze the influence of the sparse matrix $\Sigma_1$ on (25), we propose a key technical improvement: instead of resorting to covering number arguments as in Zou et al. [2018], Allen-Zhu et al. [2019], we leverage a weighted Bernstein inequality. Particularly, existing methods require taking another union bound over both all $s$-sparse subspaces and their covers (of size $\sim \binom{m}{s}9^s$), whereas our analysis only needs to union over the sparse subspaces themselves (of cardinality $\binom{m}{s}$). Our method directly demonstrates that sparsity inherently lowers computational costs by avoiding the need for dense covers. The simplicity of our technique also underscores the intrinsic benefits of sparse structures in optimization.

## A.2 Properties of Perturbation Terms

Recall that

$$\mathbf{G}_{b,0}^a(\mathbf{x}) = \sqrt{\frac{2}{m}}\Sigma_0^b(\mathbf{x})\mathbf{W}^b(0)\cdots\sqrt{\frac{2}{m}}\Sigma_0^{a+1}(\mathbf{x})\mathbf{W}^{a+1}(0)\sqrt{\frac{2}{m}}\Sigma_0^a(\mathbf{x}).$$

For any $l \in [L]$, let $\widehat{\mathbf{W}}^l$ and the diagonal matrix $\widehat{\Sigma}^l(\mathbf{x})$ be the matrices with the same size of $\mathbf{W}^l(0)$ and $\Sigma_0^l(\mathbf{x})$, respectively. Define

$$\widehat{\mathbf{G}}_b^a(\mathbf{x}) = \sqrt{\frac{2}{m}}(\Sigma_0^b(\mathbf{x})+\widehat{\Sigma}^b(\mathbf{x}))(\mathbf{W}^b(0)+\widehat{\mathbf{W}}^b)\cdots\sqrt{\frac{2}{m}}(\Sigma_0^{a+1}(\mathbf{x})+\widehat{\Sigma}^{a+1}(\mathbf{x}))$$

$$\times(\mathbf{W}^{a+1}(0)+\widehat{\mathbf{W}}^{a+1})\sqrt{\frac{2}{m}}(\Sigma_0^a(\mathbf{x})+\widehat{\Sigma}^a(\mathbf{x})), \quad 1 \leq a \leq b \leq L \tag{29}$$

and $\widehat{\mathbf{G}}_l^l(\mathbf{x}) = \sqrt{\frac{2}{m}}(\Sigma_0^l(\mathbf{x})+\widehat{\Sigma}^l(\mathbf{x}))$ for all $l \in [L]$.

**Lemma 12.** *Let* $\widehat{\mathbf{G}}_b^a(\mathbf{x})$ *with* $1 \leq a \leq b \leq L$ *be the matrix defined in (29). Assume* $\max_{l \in [L]} \|\widehat{\mathbf{W}}^l\|_2 \leq R \lesssim \sqrt{m}/(L^2 \sqrt{\log m}), m \gtrsim L^2 \log(NL/\delta)$ *and* $\widehat{\Sigma}^l(\mathbf{x}^i), \widehat{\Sigma}^l(\mathbf{x}^i) + \Sigma_0^l(\mathbf{x}^i) \in [-1, 1]^{m \times m}, \|\widehat{\Sigma}^l(\mathbf{x}^i)\|_0 \leq s \lesssim m/(L^2 \log m)$ *for all* $i \in [N], l \in [L]$. *Then, with probability at least* $1 - \delta$ *for all* $1 \leq a \leq b \leq L, i \in [N]$, *there holds*

$$\left\| \widehat{\mathbf{G}}_b^a(\mathbf{x}^i) \right\|_2 \lesssim L\sqrt{\frac{\log m}{m}}.$$

*Proof.* The proof is similar to that of Lemma 8.6 in Allen-Zhu et al. [2019], the differences lie in the dependence of $L$. We first prove that for any $1 \leq a \leq b \leq L$,

$$\left\| \sqrt{\frac{2}{m}}(\Sigma_0^b(\mathbf{x}^i) + \widehat{\Sigma}^b(\mathbf{x}^i))\mathbf{W}^b(0) \cdots \sqrt{\frac{2}{m}}(\Sigma_0^{a+1}(\mathbf{x}^i) + \widehat{\Sigma}^{a+1}(\mathbf{x}^i))\mathbf{W}^{a+1}(0) \right\|_2 \lesssim L\sqrt{\log m}.$$
(30)

We define a diagonal matrix $(\widehat{\Sigma}_1^l(\mathbf{x}^i))_{k,k} = \mathbb{I}\{\widehat{\Sigma}^l(\mathbf{x}^i)_{k,k} \neq 0\}$, and $\|\widehat{\Sigma}_1^l(\mathbf{x}^i)\|_0 \leq s$. Therefore, $\widehat{\Sigma}^l(\mathbf{x}^i) = \widehat{\Sigma}_1^l(\mathbf{x}^i)\widehat{\Sigma}^l(\mathbf{x}^i)\widehat{\Sigma}_1^l(\mathbf{x}^i)$. We decompose the left term of (30) into $2^{b-a}$ terms and control them respectively. Each matrix can be written as (ignoring the superscripts and $\mathbf{x}^i$).

$$\left( \Sigma_0 \sqrt{\frac{2}{m}}\mathbf{W}(0) \cdots \widehat{\Sigma}_1 \right) \widehat{\Sigma} \left( \widehat{\Sigma}_1 \sqrt{\frac{2}{m}}\mathbf{W}(0) \cdots \sqrt{\frac{2}{m}}\mathbf{W}(0)\widehat{\Sigma}_1 \right) \widehat{\Sigma} \cdots \widehat{\Sigma}$$
$$\times \left( \widehat{\Sigma}_1 \sqrt{\frac{2}{m}}\mathbf{W}(0) \cdots \Sigma_0 \sqrt{\frac{2}{m}}\mathbf{W}(0) \right).$$

Then, with probability at least $1 - \delta$, there holds:

- By Lemma 9, $\left\| \Sigma_0 \sqrt{\frac{2}{m}}\mathbf{W}(0) \cdots \widehat{\Sigma}_1 \right\|_2 \lesssim 1$.

- By Lemma 11, $\left\| \widehat{\Sigma}_1 \sqrt{\frac{2}{m}}\mathbf{W}(0) \cdots \sqrt{\frac{2}{m}}\mathbf{W}(0)\widehat{\Sigma}_1 \right\|_2 \lesssim 1/L$.

- By (28), $\left\| \widehat{\Sigma}_1 \sqrt{\frac{2}{m}}\mathbf{W}(0) \cdots \Sigma_0 \sqrt{\frac{2}{m}}\mathbf{W}(0) \right\|_2 \lesssim \sqrt{\log m}$.

- When there is no $\widehat{\Sigma}$, by Lemma 10, $\left\| \Sigma_0 \sqrt{\frac{2}{m}}\mathbf{W}(0) \cdots \Sigma_0 \sqrt{\frac{2}{m}}\mathbf{W}(0) \right\|_2 \lesssim L\sqrt{\log m}$.

Combined with these results, counting the number of $\widehat{\Sigma}$, we obtain

$$\left\| \sqrt{\frac{2}{m}}(\Sigma_0^b(\mathbf{x}^i) + \widehat{\Sigma}^b(\mathbf{x}^i))\mathbf{W}^b(0) \cdots \sqrt{\frac{2}{m}}(\Sigma_0^{a+1}(\mathbf{x}^i) + \widehat{\Sigma}^{a+1}(\mathbf{x}^i))\mathbf{W}^{a+1}(0) \right\|_2$$
$$\lesssim L\sqrt{\log m} + \sum_{j=1}^{b-a} \binom{b-a}{j} \left(\frac{1}{L}\right)^{j-1} 1^j \sqrt{\log m}$$
$$\leq L\sqrt{\log m} \left( 1 + \sum_{j=1}^{L} \left(\frac{eL}{j}\right)^j \left(\frac{1}{L}\right)^j \right) \lesssim L\sqrt{\log m},$$

where in the second inequality we have used $\binom{b-a}{j} \leq (e(b-a)/j)^j \leq (eL/j)^j$, the last inequality is due to $\sum_{j=1}^{L}(e/j)^j$ converges and it is bounded by a constant. Now we have proved (30). Denote $\Sigma' = \Sigma_0 + \widehat{\Sigma}$, through similar expansion, $\Sigma' \sqrt{\frac{2}{m}}(\mathbf{W}(0) + \widehat{\mathbf{W}}) \cdots (\Sigma') \sqrt{\frac{2}{m}}(\mathbf{W}(0) + \widehat{\mathbf{W}})$

is the sum of following terms

$$\left(\Sigma'\sqrt{\frac{2}{m}}\mathbf{W}(0)\cdots\Sigma'\right)\sqrt{\frac{2}{m}}\widehat{\mathbf{W}}\left(\Sigma'\sqrt{\frac{2}{m}}\mathbf{W}(0)\cdots\Sigma'\right)\sqrt{\frac{2}{m}}\widehat{\mathbf{W}}\cdots\sqrt{\frac{2}{m}}\widehat{\mathbf{W}}$$
$$\times\left(\Sigma'\sqrt{\frac{2}{m}}\mathbf{W}(0)\cdots\Sigma'\sqrt{\frac{2}{m}}\mathbf{W}(0)\right).$$

Since $\|\Sigma'\|_2 \lesssim 1$, using Eq. (30), we have

$$\left\|\Sigma'\sqrt{\frac{2}{m}}\mathbf{W}(0)\cdots\Sigma'\right\|_2 \lesssim L\sqrt{\log m},$$

$$\left\|\Sigma'\sqrt{\frac{2}{m}}\mathbf{W}(0)\cdots\Sigma'\sqrt{\frac{2}{m}}\mathbf{W}(0)\right\|_2 \lesssim L\sqrt{\log m}.$$

Note that $\max_{l\in[L]}\|\widehat{\mathbf{W}}^l\|_2 \le R \lesssim \sqrt{m}/(L^2\sqrt{\log m})$, then by counting the number of $\widehat{\mathbf{W}}$, we have

$$\left\|\widehat{\mathbf{G}}_b^a(\mathbf{x}_i)\right\|_2 \lesssim \sqrt{\frac{1}{m}}\left(L\sqrt{\log m}+\sum_{j=1}^{b-a}\binom{b-a}{j}\left(\frac{1}{L^2}\sqrt{\frac{1}{\log m}}\right)^j(L\sqrt{\log m})^{j+1}\right)$$

$$=L\sqrt{\frac{\log m}{m}}\left(1+\sum_{j=1}^{b-a}\binom{b-a}{j}\left(\frac{1}{L}\right)^j\right)\lesssim L\sqrt{\frac{\log m}{m}}.$$

The proof is completed. $\qquad\square$

Denote $\widetilde{\Sigma}(\mathbf{x}), \tilde{h}^l(\mathbf{x}), \widetilde{\mathbf{G}}_b^a(\mathbf{x})$ as (10), (12),(13) when $\mathbf{W}=\widetilde{\mathbf{W}}$.

**Lemma 13** (Claim 11.2 and Proposition 11.3 in Allen-Zhu et al. [2019]). *For any* $\mathbf{W},\widetilde{\mathbf{W}} \in \mathcal{B}_R(\mathbf{W}(0))$. *There exists a series of diagonal matrices* $\{(\Sigma'')^l \in \mathbb{R}^{m\times m}\}_{l\in[L]}$ *with entries in* $[-1,1]$ *such that for any* $l \in [L]$, *there holds*

*(a)* $h^l(\mathbf{x}) - \tilde{h}^l(\mathbf{x}) = \sum_{k=1}^l\left[\prod_{j=k+1}^l\sqrt{\frac{2}{m}}(\widetilde{\Sigma}^j(\mathbf{x})+(\Sigma'')^j)\widetilde{\mathbf{W}}^j\right]\sqrt{\frac{2}{m}}(\widetilde{\Sigma}^k(\mathbf{x})+(\Sigma'')^k)(\mathbf{W}^k-\widetilde{\mathbf{W}}^k)h^{k-1}(\mathbf{x}).$

*(b)* $\|(\Sigma'')^l\|_0 \le \|\Sigma^l(\mathbf{x})-\widetilde{\Sigma}^l(\mathbf{x})\|_0.$

The above lemma shows that the difference of ReLU networks can be expressed explicitly as the operations of matrices. The main idea is to show that $\sigma(a) - \sigma(b) = (\mathbb{I}[a \ge 0] - \xi)(a-b)$ for $\xi \in [-1,1]$. Now we introduce the following Bernstein inequality under bounded distributions.

**Lemma 14** (Theorem 2.8.4 in Vershynin [2018]). *Let* $X_1,\cdots,X_N$ *be independent, mean-zero random variables, such that* $|X_i| \le K$ *for all* $i$. *Then for every* $t \ge 0$, *we have*

$$\mathbb{P}\left(\left|\sum_{i=1}^N X_i\right| \ge t\right) \le 2\exp\left(-\frac{t^2/2}{\lambda^2+Kt/3}\right),$$

*where* $\lambda^2 = \sum_{i=1}^N \mathbb{E}X_i^2$ *is the sum of the variance.*

The following lemma shows that under overparameterized setting, the outputs and activation patterns for deep relu networks near initialization do not change much.

**Lemma 15.** *Suppose* $m \gtrsim L^{10}\log(NL/\delta)(\log m)^4 R^2$. *Then with probability at least* $1-\delta$, *for any* $\mathbf{W} \in \mathcal{B}_R(\mathbf{W}(0)), i \in [N]$ *and* $l \in [L]$, *there holds*

$$\|h^l(\mathbf{x}^i)-h_0^l(\mathbf{x}^i)\|_2 \lesssim L^2\sqrt{\frac{\log m}{m}}R \text{ and } \|\Sigma^l(\mathbf{x}^i)-\Sigma_0^l(\mathbf{x}^i)\|_0 \lesssim L^{4/3}(\log m)^{1/3}(mR)^{2/3}. \quad (31)$$

*Proof.* We prove these two inequalities by induction. Note that (31) holds for $l = 0$. Now we suppose (31) holds for $l - 1$. Let $\kappa > 0$ be a constant. For $i \in [N]$ and $l \in [L]$, we define $A^l(\mathbf{x}^i) = \{r \in [m] : \mathbb{I}\{\langle \mathbf{w}_r^l, h^{l-1}(\mathbf{x}^i)\rangle \geq 0\} \neq \mathbb{I}\{\langle \mathbf{w}_r^l(0), h_0^{l-1}(\mathbf{x}^i)\rangle \geq 0\}\}$, then $\|\Sigma^l(\mathbf{x}^i) - \Sigma_0^l(\mathbf{x}^i)\|_0 = |A^l(\mathbf{x}^i)|$. Furthermore, we decompose $A^l(\mathbf{x}^i)$ into two parts based on the behavior of $\mathbf{w}_r^l(0)$:

$$A_1^l(\mathbf{x}^i) = \{r \in A^l(\mathbf{x}^i) : |\langle \mathbf{w}_r^l(0), h_0^{l-1}(\mathbf{x}^i)\rangle| \leq \kappa\} \quad \text{and} \quad A_2^l(\mathbf{x}^i) = \{r \in A^l(\mathbf{x}^i) : |\langle \mathbf{w}_r^l(0), h_0^{l-1}(\mathbf{x}^i)\rangle| > \kappa\}.$$

We will control $|A_1^l(\mathbf{x}^i)|$ and $|A_2^l(\mathbf{x}^i)|$ respectively.

For $r \in [m]$, we define $F_{r,i}^l = \mathbb{I}\{|\langle \mathbf{w}_r^l(0), h_0^{l-1}(\mathbf{x}^i)\rangle| \leq \kappa\}$. From Lemma 6 we know that $2/3 \leq \|h_0^{l-1}(\mathbf{x}^i)\|_2^2 \leq 4/3$. Condition on $h_0^{l-1}(\mathbf{x}^i)$, $\langle \mathbf{w}_r^l(0), h_0^{l-1}(\mathbf{x}^i)\rangle \sim \mathcal{N}(0, \|h_0^{l-1}(\mathbf{x}^i)\|_2^2)$, then we have

$$\text{var}(F_{r,i}^l) \leq \mathbb{E}(F_{r,i}^l)^2 = \mathbb{E}(F_{r,i}^l) = \mathbb{P}(-\kappa \leq \langle \mathbf{W}_r^l(0), h_0^{l-1}(\mathbf{x}^i)\rangle \leq \kappa)$$

$$\leq \frac{3}{2\sqrt{\pi}} \int_{-\kappa}^{\kappa} e^{-3x^2/8} dx \leq C\kappa.$$

Then by Lemma 14, choose $K = 1, t = mC\kappa, \lambda^2 \leq mC\kappa$, it then follows that

$$\mathbb{P}\left(\left|\sum_{r=1}^m F_{r,i}^l - m\mathbb{E}(F_{r,i}^l)\right| \leq mC\kappa|h_0^{l-1}(\mathbf{x}^i)\right) \geq 1 - 2\exp\left(-\frac{(mC\kappa)^2/2}{mC\kappa + mC\kappa/3}\right).$$

Hence, taking union bounds over $l, i$, with probability at least $1 - CnL\exp(-m\kappa)$, there holds for all $i, l$,

$$|A_1^l(\mathbf{x}^i)| \leq \sum_{r=1}^m F_{r,i}^l \lesssim m\kappa. \tag{32}$$

For $r \in A_2^l(\mathbf{x}_i)$, since $\mathbb{I}\{\langle \mathbf{w}_r^l, h^{l-1}(\mathbf{x}^i)\rangle \geq 0\} \neq \mathbb{I}\{\langle \mathbf{w}_r^l(0), h_0^{l-1}(\mathbf{x}^i)\rangle \geq 0\}$, we have

$$(\langle \mathbf{w}_r^l, h^{l-1}(\mathbf{x}^i)\rangle - \langle \mathbf{w}_r^l(0), h_0^{l-1}(\mathbf{x}^i)\rangle)^2 \geq |\langle \mathbf{w}_r^l(0), h_0^{l-1}(\mathbf{x}^i)\rangle|^2 > \kappa^2.$$

We deduce that

$$\|\mathbf{W}^l h^{l-1}(\mathbf{x}^i) - \mathbf{W}^l(0)h_0^{l-1}(\mathbf{x}^i)\|_2^2 \geq \sum_{r \in A_2^l(\mathbf{x}_i)} (\langle \mathbf{w}_r^l, h^{l-1}(\mathbf{x}^i)\rangle - \langle \mathbf{w}_r^l(0), h_0^{l-1}(\mathbf{x}^i)\rangle)^2$$

$$> \sum_{r \in A_2^l(\mathbf{x}^i)} \kappa^2 = \kappa^2|A_2^l(\mathbf{x}^i)|. \tag{33}$$

By assumption $\|h^{l-1}(\mathbf{x}^i) - h_0^{l-1}(\mathbf{x}^i)\|_2 \lesssim L^2 R\sqrt{\log m/m}$ and Lemma 6, we get

$$\|\mathbf{W}^l h^{l-1}(\mathbf{x}^i) - \mathbf{W}^l(0)h_0^{l-1}(\mathbf{x}^i)\|_2^2$$
$$\leq(\|\mathbf{W}^l - \mathbf{W}^l(0)\|_2\|h^{l-1}(\mathbf{x}^i) - h_0^{l-1}(\mathbf{x}^i) + h_0^{l-1}(\mathbf{x}^i)\|_2 + \|\mathbf{W}^l(0)\|_2\|h^{l-1}(\mathbf{x}^i) - h_0^{l-1}(\mathbf{x}^i)\|_2)^2$$
$$\lesssim(R(\|h^{l-1}(\mathbf{x}^i) - h_0^{l-1}(\mathbf{x}^i)\|_2 + 1) + \sqrt{m}\|h^{l-1}(\mathbf{x}^i) - h_0^{l-1}(\mathbf{x}^i)\|_2)^2 \lesssim L^4 R^2 \log m.$$

Combined with (33), we have

$$|A_2^l(\mathbf{x}^i)| \lesssim \frac{L^4 R^2 \log m}{\kappa^2}. \tag{34}$$

From (32) and (34) we know that

$$\|\Sigma^l(\mathbf{x}^i) - \Sigma_0^l(\mathbf{x}^i)\|_0 = |A^l(\mathbf{x}^i)| = |A_1^l(\mathbf{x}^i)| + |A_2^l(\mathbf{x}^i)|$$

$$\lesssim m\kappa + \frac{L^4 R^2 \log m}{(\kappa)^2} \lesssim L^{4/3}(\log m)^{1/3}(mR)^{2/3},$$

where in the last inequality we choose $\kappa = L^{4/3}(\log m)^{1/3}R^{2/3}m^{-1/3}$. Hence, due to the overparameterization of $m$, we have with probability at least $1 - \delta$, for $i \in [N]$,

$$\|\Sigma^l(\mathbf{x}^i) - \Sigma_0^l(\mathbf{x}^i)\|_0 \lesssim L^{4/3}(\log m)^{1/3}(mR)^{2/3} \lesssim \frac{m}{L^2 \log m}.$$

Applying Lemma 13, we have

$$h^l(\mathbf{x}^i) - h_0^l(\mathbf{x}^i) = \sum_{k=1}^{l} \widehat{\mathbf{G}}_{l,0}^k(\mathbf{x}^i)(\mathbf{W}^k - \mathbf{W}^k(0))h_0^{k-1}(\mathbf{x}^i), \tag{35}$$

where $\widehat{\mathbf{G}}_{l,0}^k(\mathbf{x}^i)$ is defined as

$$\widehat{\mathbf{G}}_{l,0}^k(\mathbf{x}^i) = \left[ \prod_{j=k+1}^{l} \sqrt{\frac{2}{m}}(\Sigma^j(\mathbf{x}^i) + (\Sigma'')^j)\mathbf{W}^j \right] \sqrt{\frac{2}{m}}(\Sigma^k(\mathbf{x}^i) + (\Sigma'')^k). \tag{36}$$

It then follows that

$$\|\Sigma^j(\mathbf{x}^i) + (\Sigma'')^j - \Sigma_0^j(\mathbf{x}^i)\|_0$$
$$\leq \|\Sigma^j(\mathbf{x}^i) - \Sigma_0^j(\mathbf{x}^i)\|_0 + \|(\Sigma'')^j\|_0 \leq 2\|\Sigma^j(\mathbf{x}^i) - \Sigma_0^j(\mathbf{x}^i)\|_0 \lesssim \frac{m}{L^2 \log m}.$$

Our overparameterization requirement implies that $R \lesssim \sqrt{m}/(L^2\sqrt{\log m})$. Hence, by Lemma 12, we have

$$\|\widehat{\mathbf{G}}_{l,0}^k(\mathbf{x}^i)\|_2 \lesssim L\sqrt{\frac{\log m}{m}}. \tag{37}$$

Therefore,

$$\|h^l(\mathbf{x}^i) - h_0^l(\mathbf{x}^i)\|_2$$
$$= \left\| \sum_{k=1}^{l} \widehat{\mathbf{G}}_{l,0}^k(\mathbf{x}^i)(\mathbf{W}^k - \mathbf{W}^k(0))h_0^{k-1}(\mathbf{x}^i) \right\|_2$$
$$\lesssim \sum_{k=1}^{l} L\sqrt{\frac{\log m}{m}}R\|h_0^{k-1}(\mathbf{x}^i)\|_2 \lesssim L^2\sqrt{\frac{\log m}{m}}R,$$

where the last inequality results from Lemma 6. As a result, (31) holds for $l$. We have completed the proof of the lemma. $\qquad\square$

The above lemma and Lemma 6 imply that with probability at least $1 - \delta$, for all $l \in [L], i \in [N]$,

$$\|h^l(\mathbf{x}^i)\|_2 \lesssim L^2\sqrt{\frac{\log m}{m}}R + 1 \lesssim 1. \tag{38}$$

*Remark* 10. Although our approach shares similarities with Lemma B.3 in Zou et al. [2018], our analysis relaxes the required conditions. Specifically, we only require $R/\sqrt{m} = \widetilde{O}(L^{-5})$, whereas their result demands the stricter scaling $R/\sqrt{m} = \widetilde{O}(L^{-11})$. Furthermore, compared to Lemma 8.2 in Allen-Zhu et al. [2019], they derive the bound $\|h^l(\mathbf{x}^i) - h_0^l(\mathbf{x}^i)\|_2 \lesssim RL^{5/2}\sqrt{\log m}/\sqrt{m}$, which is worse than our result by a factor of $\sqrt{L}$.

The following lemma shows the uniform concentration property of deep ReLU networks, which is crucial in the generalization analysis.

**Lemma 16.** *Let $R \geq 1$ be a constant. Assume $m \gtrsim L^{11}d(\log m)^5 \log(L/\delta)R^2$. Then with probability at least $1 - \delta$, for $\mathbf{W} \in \mathcal{B}_R(\mathbf{W}(0)), l \in [L]$, we have*

$$\sup_{\mathbf{x} \in \mathcal{X}} \|h^l(\mathbf{x}) - h_0^l(\mathbf{x})\|_2 \lesssim L^2\sqrt{\frac{\log m}{m}}R. \tag{39}$$

*Proof.* We consider the $1/(C^L\sqrt{m})$-cover of $S^{d-1}$ and denote it by $D = \{\mathbf{x}^1, \cdots, \mathbf{x}^{|D|}\}$. By Lemma 4.2.13 in Vershynin [2018],

$$|D| \leq (1 + 2C^L\sqrt{m})^d.$$

Note that Lemma 15 holds for any finite set $K = \{\mathbf{x}^1, \cdots, \mathbf{x}^N\}$. Letting $K = D$, we obtain that if $m \gtrsim L^{11} d(\log m)^5 \log(L/\delta) R^2 \gtrsim L^{10} \log(|D|L/\delta)(\log m)^4 R^2$, then

$$\|h^l(\mathbf{x}^j) - h_0^l(\mathbf{x}^j)\|_2 \lesssim L^2 \sqrt{\frac{\log m}{m}} R, \quad 1 \leq j \leq |D|. \tag{40}$$

For any $\mathbf{x} \in \mathcal{X}$, there exists $\mathbf{x}^j \in D$ with $\|\mathbf{x} - \mathbf{x}^j\|_2 \leq 1/(C^L \sqrt{m})$. It then follows that

$$\|h^l(\mathbf{x}) - h^l(\mathbf{x}^j)\|_2^2$$
$$= \frac{2}{m} \sum_{r=1}^m (\sigma(\langle \mathbf{w}_r^l, h^{l-1}(\mathbf{x}) \rangle) - \sigma(\langle \mathbf{w}_r^l, h^{l-1}(\mathbf{x}^j) \rangle))^2$$
$$\leq \frac{2}{m} \sum_{r=1}^m (\langle \mathbf{w}_r^l, h^{l-1}(\mathbf{x}) \rangle - \langle \mathbf{w}_r^l, h^{l-1}(\mathbf{x}^j) \rangle)^2$$
$$= \frac{2}{m} \|\mathbf{W}^l(h^{l-1}(\mathbf{x}) - h^{l-1}(\mathbf{x}^j))\|_2^2 \leq C \|h^{l-1}(\mathbf{x}) - h^{l-1}(\mathbf{x}^j)\|_2^2$$
$$\leq C^L \|\mathbf{x} - \mathbf{x}^j\|_2^2 \leq \frac{1}{m},$$

where the first inequality is due to $\sigma(\cdot)$ is 1-Lipschitz. In the second inequality we have used $\|\mathbf{W}^l\|_2 \leq \|\mathbf{W}^l(0)\|_2 + R \lesssim \sqrt{m}$ due to Lemma 2. Similarly, we derive that

$$\|h_0^l(\mathbf{x}) - h_0^l(\mathbf{x}^j)\|_2^2 \leq \frac{1}{m}.$$

Therefore, combined with (40), we have

$$\|h^l(\mathbf{x}) - h_0^l(\mathbf{x})\|_2$$
$$\leq \|h^l(\mathbf{x}) - h^l(\mathbf{x}^j)\|_2 + \|h^l(\mathbf{x}^j) - h_0^l(\mathbf{x}^j)\|_2 + \|h_0^l(\mathbf{x}) - h_0^l(\mathbf{x}^j)\|_2$$
$$\lesssim \frac{1}{\sqrt{m}} + L^2 \sqrt{\frac{\log m}{m}} R + \frac{1}{\sqrt{m}} \lesssim L^2 \sqrt{\frac{\log m}{m}} R,$$

where the last inequality results from $R \geq 1$.

The proof is completed. $\qquad\square$

*Remark* 11. This lemma is a property of deep ReLU networks near initialization that does not depend on the training data. Compared to prior work, while Allen-Zhu et al. [2019], Zou et al. [2018] only establishes bounds for the training data, we prove the uniform convergence over the entire input space. Previous work on uniform concentration demonstrated that $\sup_{\mathbf{x} \in \mathcal{X}} \|h^l(\mathbf{x}) - h_0^l(\mathbf{x})\|_2 \lesssim C^L R/\sqrt{m}$ [Xu and Zhu, 2024]. We present a significant improvement, reducing the dependence on $L$ from exponential to polynomial.

In the following part, we apply previous technical lemmas to $K = S_1$ and get properties of deep neural networks over the training dataset.

**Lemma 17.** *Suppose $m \gtrsim L^{10} \log(nL/\delta)(\log m)^4 R^2$. Then with probability at least $1 - \delta$ for all $\mathbf{W} \in \mathcal{B}_R(\mathbf{W}(0)), l \in [L], i \in [n]$*

$$\|\mathbf{a}^\top (\mathbf{G}_L^l(\mathbf{x}_i) - \mathbf{G}_{L,0}^l(\mathbf{x}_i))\|_2 \lesssim \frac{L^{5/3}(\log m)^{2/3} R^{1/3}}{m^{1/6}}$$

*Proof.* For the case $l = L$,

$$\|\mathbf{a}^\top (\mathbf{G}_L^L(\mathbf{x}_i) - \mathbf{G}_{L,0}^L(\mathbf{x}_i))\|_2 = \sqrt{\frac{2}{m}} \|\mathbf{a}^\top (\Sigma^L(\mathbf{x}_i) - \Sigma_0^L(\mathbf{x}_i))\|_2$$

$$= \sqrt{\frac{2}{m}} \sqrt{\sum_{r=1}^m a_r^2 (\mathbb{I}\{\langle \mathbf{w}_r^l, h^{l-1}(\mathbf{x}_i) \rangle \geq 0\} - \mathbb{I}\{\langle \mathbf{w}_r^l(0), h_0^{l-1}(\mathbf{x}_i) \rangle \geq 0\})^2}$$

$$= \sqrt{\frac{2}{m}} \sqrt{\sum_{r \in A^l(\mathbf{x}_i)} |a_r|} = \sqrt{\frac{2|A^l(\mathbf{x}_i)|}{m}} \lesssim \frac{L^{2/3}(\log m)^{1/6} R^{1/3}}{m^{1/6}}, \tag{41}$$

where the last inequality is due to Lemma 15.

Now we suppose $l < L$, then

$$\mathbf{a}^\top(\mathbf{G}_L^l(\mathbf{x}_i) - \mathbf{G}_{L,0}^l(\mathbf{x}_i))$$

$$=\mathbf{a}^\top\sqrt{\frac{2}{m}}(\Sigma^L(\mathbf{x}_i)\mathbf{W}^L\mathbf{G}_{L-1}^l(\mathbf{x}_i) - \Sigma_0^L(\mathbf{x}_i)\mathbf{W}^L(0)\mathbf{G}_{L-1,0}^l(\mathbf{x}_i))$$

$$=\sqrt{\frac{2}{m}}\mathbf{a}^\top(\Sigma^L(\mathbf{x}_i) - \Sigma_0^L(\mathbf{x}_i))\mathbf{W}^L\mathbf{G}_{L-1}^l(\mathbf{x}_i) + \sqrt{\frac{2}{m}}\mathbf{a}^\top\Sigma_0^L(\mathbf{x}_i)(\mathbf{W}^L - \mathbf{W}^L(0))\mathbf{G}_{L-1}^l(\mathbf{x}_i))$$

$$+\sqrt{\frac{2}{m}}\mathbf{a}^\top\Sigma_0^L(\mathbf{x}_i)\mathbf{W}^L(0)(\mathbf{G}_{L-1}^l(\mathbf{x}_i) - \mathbf{G}_{L-1,0}^l(\mathbf{x}_i))$$

$$=\sqrt{\frac{2}{m}}(\mathbf{a}^\top(\Sigma^L(\mathbf{x}_i) - \Sigma_0^L(\mathbf{x}_i))\mathbf{W}^L\mathbf{G}_{L-1}^l(\mathbf{x}_i) + \mathbf{a}^\top\Sigma_0^L(\mathbf{x}_i)(\mathbf{W}^L - \mathbf{W}^L(0))\mathbf{G}_{L-1}^l(\mathbf{x}_i)),$$

where the last equality is according to Lemma 7. Applying Lemma 12 and Lemma 15, there holds

$$\|\mathbf{G}_{L-1}^l(\mathbf{x}_i)\|_2 \lesssim L\sqrt{\frac{\log m}{m}}.$$

This implies that

$$\|\mathbf{a}^\top(\mathbf{G}_L^l(\mathbf{x}_i) - \mathbf{G}_{L,0}^l(\mathbf{x}_i))\|_2$$

$$\lesssim\sqrt{\frac{2}{m}}\left(\|\mathbf{a}^\top(\Sigma^L(\mathbf{x}_i) - \Sigma_0^L(\mathbf{x}_i))\mathbf{W}^L\mathbf{G}_{L-1}^l(\mathbf{x}_i)\|_2 + \|\mathbf{a}^\top\Sigma_0^L(\mathbf{x}_i)(\mathbf{W}^L - \mathbf{W}^L(0))\mathbf{G}_{L-1}^l(\mathbf{x}_i)\|_2\right)$$

$$\leq\sqrt{\frac{2}{m}}\|\mathbf{a}^\top(\Sigma^L(\mathbf{x}_i) - \Sigma_0^L(\mathbf{x}_i))\|_2\|\mathbf{W}^L\|_2\|\mathbf{G}_{L-1}^l(\mathbf{x}_i)\|_2$$

$$+\sqrt{\frac{2}{m}}\|\mathbf{a}\|_2\|\Sigma_0^L(\mathbf{x}_i)\|_2\|\mathbf{W}^L - \mathbf{W}^L(0)\|_2\|\mathbf{G}_{L-1}^l(\mathbf{x}_i)\|_2$$

$$\lesssim\frac{L^{2/3}(\log m)^{1/6}R^{1/3}}{m^{1/6}}\sqrt{m}L\sqrt{\frac{\log m}{m}} + L\sqrt{\frac{\log m}{m}}R$$

$$\lesssim\frac{L^{5/3}(\log m)^{2/3}R^{1/3}}{m^{1/6}},$$

where we have used (41) in the third inequality. The proof is completed. $\qquad\square$

**Lemma 18.** *Assume $m \gtrsim L^{10}\log(nL/\delta)(\log m)^4 R^2$. Then with probability at least $1 - \delta$, for any $\mathbf{W} \in \mathcal{B}_R(\mathbf{W}(0))$, $i \in [n]$ and $l \in [L]$, there holds*

$$\left\|\frac{\partial f_\mathbf{W}(\mathbf{x}_i)}{\partial\mathbf{W}^l} - \frac{\partial f_{\mathbf{W}(0)}(\mathbf{x}_i)}{\partial\mathbf{W}^l(0)}\right\|_F \lesssim \frac{L^{5/3}(\log m)^{2/3}R^{1/3}}{m^{1/6}}. \tag{42}$$

*Proof.* Since $\|xy^\top\|_F = \|x\|_2\|y\|_2$ for two vectors $x, y$, we have

$$\left\|\frac{\partial f_\mathbf{W}(\mathbf{x}_i)}{\partial\mathbf{W}^l} - \frac{\partial f_{\mathbf{W}(0)}(\mathbf{x}_i)}{\partial\mathbf{W}^l(0)}\right\|_F$$

$$=\|h^{l-1}(\mathbf{x}_i)\mathbf{a}^\top\mathbf{G}_L^l(\mathbf{x}_i) - h_0^{l-1}(\mathbf{x}_i)\mathbf{a}^\top\mathbf{G}_{L,0}^l(\mathbf{x}_i)\|_F$$

$$\leq\|h^{l-1}(\mathbf{x}_i)\mathbf{a}^\top(\mathbf{G}_L^l(\mathbf{x}_i) - \mathbf{G}_{L,0}^l(\mathbf{x}_i))\|_F + \|(h^{l-1}(\mathbf{x}_i) - h_0^{l-1}(\mathbf{x}_i))\mathbf{a}^\top\mathbf{G}_{L,0}^l(\mathbf{x}_i)\|_F$$

$$=\|h^{l-1}(\mathbf{x}_i)\|_2\|\mathbf{a}^\top\left(\mathbf{G}_L^l(\mathbf{x}_i) - \mathbf{G}_{L,0}^l(\mathbf{x}_i)\right)\|_2 + \|h^{l-1}(\mathbf{x}_i) - h_0^{l-1}(\mathbf{x}_i)\|_2\|\mathbf{a}^\top\mathbf{G}_{L,0}^l(\mathbf{x}_i)\|_2.$$

Using (38) and Lemma 17, we have

$$\|h^{l-1}(\mathbf{x}_i)\|_2\|\mathbf{a}^\top\left(\mathbf{G}_L^l(\mathbf{x}_i) - \mathbf{G}_{L,0}^l(\mathbf{x}_i)\right)\|_2 \lesssim \frac{L^{5/3}(\log m)^{2/3}R^{1/3}}{m^{1/6}}.$$

Applying Lemma 15 and (24), we obtain

$$\|h^{l-1}(\mathbf{x}_i) - h_0^{l-1}(\mathbf{x}_i)\|_2\|\mathbf{a}^\top\mathbf{G}_{L,0}^l(\mathbf{x}_i)\|_2 \lesssim L^2\sqrt{\frac{\log m}{m}}R\sqrt{m}L\sqrt{\frac{\log m}{m}} = \frac{L^3R\log m}{\sqrt{m}}.$$

It then follows that

$$\left\| \frac{\partial f_{\mathbf{W}}(\mathbf{x}_i)}{\partial \mathbf{W}^l} - \frac{\partial f_{\mathbf{W}(0)}(\mathbf{x}_i)}{\partial \mathbf{W}^l(0)} \right\|_F \lesssim \frac{L^{5/3}(\log m)^{2/3}R^{1/3}}{m^{1/6}} + \frac{L^3 R \log m}{\sqrt{m}} \lesssim \frac{L^{5/3}(\log m)^{2/3}R^{1/3}}{m^{1/6}}.$$

The proof is completed. $\qquad\qquad\qquad\qquad\qquad\qquad\qquad\qquad\qquad\qquad\qquad\qquad\qquad\qquad\square$

## B  Proofs for Optimization

**Lemma 19.** *Suppose $m \gtrsim L^{10}\log(nL/\delta)(\log m)^4 R^2$, then with probability at least $1 - \delta$, for $i \in [n], \widetilde{\mathbf{W}}, \mathbf{W} \in \mathcal{B}_R(\mathbf{W}(0))$, we have*

$$\left| f_{\widetilde{\mathbf{W}}}(\mathbf{x}_i) - f_{\mathbf{W}}(\mathbf{x}_i) - \left\langle \frac{\partial f_{\widetilde{\mathbf{W}}}(\mathbf{x}_i)}{\partial \widetilde{\mathbf{W}}}, \widetilde{\mathbf{W}} - \mathbf{W} \right\rangle \right| \lesssim \frac{L^{8/3}R^{4/3}(\log m)^{2/3}}{m^{1/6}}.$$

This lemma shows that deep ReLU networks near initialization are almost linear.

*Proof.* Note that

$$\left\langle \frac{\partial f_{\widetilde{\mathbf{W}}}(\mathbf{x}_i)}{\partial \widetilde{\mathbf{W}}}, \widetilde{\mathbf{W}} - \mathbf{W} \right\rangle = \sum_{l=1}^{L} \left\langle \frac{\partial f_{\widetilde{\mathbf{W}}}(\mathbf{x}_i)}{\partial \widetilde{\mathbf{W}}^l}, \widetilde{\mathbf{W}}^l - \mathbf{W}^l \right\rangle$$

$$= \sum_{l=1}^{L} \left\langle (\widetilde{\mathbf{G}}_L^l(\mathbf{x}_i))^\top \mathbf{a}(\tilde{h}^{l-1}(\mathbf{x}_i))^\top, \widetilde{\mathbf{W}}^l - \mathbf{W}^l \right\rangle$$

$$= \sum_{l=1}^{L} \mathbf{a}^\top \widetilde{\mathbf{G}}_L^l(\mathbf{x}_i)(\widetilde{\mathbf{W}}^l - \mathbf{W}^l)\tilde{h}^{l-1}(\mathbf{x}_i).$$

Since $f_{\mathbf{W}}(\mathbf{x}_i) = \mathbf{a}^\top h^L(\mathbf{x}_i)$, applying Lemma 13, we obtain

$$f_{\widetilde{\mathbf{W}}}(\mathbf{x}_i) - f_{\mathbf{W}}(\mathbf{x}_i) = \mathbf{a}^\top(\tilde{h}^L(\mathbf{x}_i) - h^L(\mathbf{x}_i))$$

$$= \sum_{l=1}^{L} \left[ \prod_{j=l+1}^{L} \sqrt{\frac{2}{m}}(\Sigma^j(\mathbf{x}_i) + (\Sigma'')^j)\mathbf{W}^j \right] \sqrt{\frac{2}{m}}(\Sigma^l(\mathbf{x}_i) + (\Sigma'')^l)(\widetilde{\mathbf{W}}^l - \mathbf{W}^l)\tilde{h}^{l-1}(\mathbf{x}_i)$$

with $\|(\Sigma'')^l\|_0 \le \|\Sigma^l(\mathbf{x}_i) - \widetilde{\Sigma}^l(\mathbf{x}_i)\|_0, \Sigma^l(\mathbf{x}_i) + (\Sigma'')^l - \Sigma_0^l(\mathbf{x}_i) \in [-1, 1]^m$. Then

$$\|\Sigma^l(\mathbf{x}_i) + (\Sigma'')^l - \Sigma_0^l(\mathbf{x}_i)\|_0 \le \|\widetilde{\Sigma}^l(\mathbf{x}_i) - \Sigma_0^l(\mathbf{x}_i)\|_0 + 2\|\Sigma^l(\mathbf{x}_i) - \Sigma_0^l(\mathbf{x}_i)\|_0$$

$$\lesssim L^{4/3}(\log m)^{1/3}(mR)^{\frac{2}{3}},$$

the last inequality is due to Lemma 15. We further let

$$\widehat{\mathbf{G}}_b^a(\mathbf{x}_i) = \left[ \prod_{j=a+1}^{b} \sqrt{\frac{2}{m}}(\Sigma^j(\mathbf{x}_i) + (\Sigma'')^j)\mathbf{W}^j \right] \sqrt{\frac{2}{m}}(\Sigma^a(\mathbf{x}_i) + (\Sigma'')^a).$$

By Lemma 12, we have

$$\|\widehat{\mathbf{G}}_b^a(\mathbf{x}_i)\|_2 \lesssim L\sqrt{\frac{\log m}{m}}.$$

Following the proof of Lemma 17, we have

$$\|\mathbf{a}^\top(\widehat{\mathbf{G}}_L^l(\mathbf{x}_i) - \mathbf{G}_{L,0}^l(\mathbf{x}_i))\|_2 \le \frac{L^{5/3}(\log m)^{2/3}R^{1/3}}{m^{1/6}},$$

which implies that

$$\|\mathbf{a}^\top(\widehat{\mathbf{G}}_L^l(\mathbf{x}_i) - \widetilde{\mathbf{G}}_L^l(\mathbf{x}_i))\|_2 \le \|\mathbf{a}^\top(\widehat{\mathbf{G}}_L^l(\mathbf{x}_i) - \mathbf{G}_{L,0}^l(\mathbf{x}_i))\|_2 + \|\mathbf{a}^\top(\mathbf{G}_{L,0}^l(\mathbf{x}_i) - \widetilde{\mathbf{G}}_L^l(\mathbf{x}_i))\|_2$$

$$\lesssim \frac{L^{5/3}(\log m)^{2/3}R^{1/3}}{m^{1/6}}.$$

Hence,

$$
\left| f_{\widetilde{\mathbf{W}}}(\mathbf{x}_i) - f_{\mathbf{W}}(\mathbf{x}_i) - \left\langle \frac{\partial f_{\widetilde{\mathbf{W}}}(\mathbf{x}_i)}{\partial \widetilde{\mathbf{W}}}, \widetilde{\mathbf{W}} - \mathbf{W} \right\rangle \right|
$$

$$
= \left| \sum_{l=1}^{L} \mathbf{a}^\top (\widehat{\mathbf{G}}_L^l(\mathbf{x}_i) - \widetilde{\mathbf{G}}_L^l(\mathbf{x}_i))(\widetilde{\mathbf{W}}^l - \mathbf{W}^l) \tilde{h}^{l-1}(\mathbf{x}_i) \right|
$$

$$
\leq \sum_{l=1}^{L} \|\mathbf{a}^\top (\widehat{\mathbf{G}}_L^l(\mathbf{x}_i) - \widetilde{\mathbf{G}}_L^l(\mathbf{x}_i))^\top\|_2 \|\widetilde{\mathbf{W}}^l - \mathbf{W}^l\|_2 \|\tilde{h}^{l-1}(\mathbf{x}_i)\|_2
$$

$$
\lesssim L \frac{L^{5/3} (\log m)^{2/3} R^{1/3}}{m^{1/6}} R = \frac{L^{8/3} R^{4/3} (\log m)^{2/3}}{m^{1/6}},
$$

where in the last inequality we have used (38). The proof is completed. $\qquad \square$

The following lemma shows that $\mathcal{L}_S$ is almost convex near initialization. It becomes more convex as the width grows.

**Lemma 20.** *Suppose $m \gtrsim L^{10} \log(nL/\delta)(\log m)^4 R^2$, then with probability at least $1 - \delta$, we have for $\widetilde{\mathbf{W}}, \mathbf{W} \in \mathcal{B}_R(\mathbf{W}(0))$,*

$$
\left\langle \widetilde{\mathbf{W}} - \mathbf{W}, \frac{\partial \mathcal{L}_S(\widetilde{\mathbf{W}})}{\partial \widetilde{\mathbf{W}}} \right\rangle \geq \mathcal{L}_S(\widetilde{\mathbf{W}}) - \mathcal{L}_S(\mathbf{W}) + \frac{2}{n} \sum_{i=1}^{n} (l'(y_i f_{\widetilde{\mathbf{W}}}(\mathbf{x}_i) - l'(y_i f_{\mathbf{W}}(\mathbf{x}_i))^2
$$

$$
- \frac{CL^{8/3} (\log m)^{2/3} R^{4/3}}{m^{1/6}} \mathcal{L}_S(\widetilde{\mathbf{W}}).
$$

*Proof.* Since $\ell$ is $1/4$-smooth, it enjoys the co-coercivity, i.e., $\ell(a) \geq \ell(b) + (a-b)\ell'(b) + 2(\ell'(a) - \ell'(b))^2$, which implies that

$$
y_i \ell'(y_i f_{\widetilde{\mathbf{W}}}(\mathbf{x}_i))(f_{\widetilde{\mathbf{W}}}(\mathbf{x}_i) - f_{\mathbf{W}}(\mathbf{x}_i))
$$

$$
\geq \ell(y_i f_{\widetilde{\mathbf{W}}}(\mathbf{x}_i)) - \ell(y_i f_{\mathbf{W}}(\mathbf{x}_i)) + 2(\ell'(y_i f_{\widetilde{\mathbf{W}}}(\mathbf{x}_i)) - \ell'(y_i f_{\mathbf{W}}(\mathbf{x}_i)))^2.
$$

We combine the above inequality with Lemma 19 and obtain

$$
\left\langle \widetilde{\mathbf{W}} - \mathbf{W}, \frac{\partial \mathcal{L}_S(\widetilde{\mathbf{W}})}{\partial \widetilde{\mathbf{W}}} \right\rangle
$$

$$
= \frac{1}{n} \sum_{i=1}^{n} \left\langle \widetilde{\mathbf{W}} - \mathbf{W}, \frac{\partial f_{\widetilde{\mathbf{W}}}(\mathbf{x}_i)}{\partial \widetilde{\mathbf{W}}} \right\rangle y_i \ell'(y_i f_{\widetilde{\mathbf{W}}}(\mathbf{x}_i))
$$

$$
= \frac{1}{n} \sum_{i=1}^{n} y_i \ell'(y_i f_{\widetilde{\mathbf{W}}}(\mathbf{x}_i)) \left( f_{\widetilde{\mathbf{W}}}(\mathbf{x}_i) - f_{\mathbf{W}}(\mathbf{x}_i) - \left( f_{\widetilde{\mathbf{W}}}(\mathbf{x}_i) - f_{\mathbf{W}}(\mathbf{x}_i) - \left\langle \frac{\partial f_{\widetilde{\mathbf{W}}}(\mathbf{x}_i)}{\partial \widetilde{\mathbf{W}}}, \widetilde{\mathbf{W}} - \mathbf{W} \right\rangle \right) \right)
$$

$$
\geq \frac{1}{n} \sum_{i=1}^{n} \left( \ell(y_i f_{\widetilde{\mathbf{W}}}(\mathbf{x}_i)) - \ell(y_i f_{\mathbf{W}}(\mathbf{x}_i)) + 2(\ell'(y_i f_{\widetilde{\mathbf{W}}}(\mathbf{x}_i)) - \ell'(y_i f_{\mathbf{W}}(\mathbf{x}_i)))^2 \right)
$$

$$
- \frac{1}{n} \sum_{i=1}^{n} \left| f_{\widetilde{\mathbf{W}}}(\mathbf{x}_i) - f_{\mathbf{W}}(\mathbf{x}_i) - \left\langle \frac{\partial f_{\widetilde{\mathbf{W}}}(\mathbf{x}_i)}{\partial \widetilde{\mathbf{W}}}, \widetilde{\mathbf{W}} - \mathbf{W} \right\rangle \right| \ell(y_i f_{\widetilde{\mathbf{W}}}(\mathbf{x}_i))
$$

$$
\geq \mathcal{L}_S(\widetilde{\mathbf{W}}) - \mathcal{L}_S(\mathbf{W}) + \frac{2}{n} \sum_{i=1}^{n} (l'(y_i f_{\widetilde{\mathbf{W}}}(\mathbf{x}_i) - l'(y_i f_{\mathbf{W}}(\mathbf{x}_i)))^2
$$

$$
- CL^{8/3} (\log m)^{2/3} R^{4/3} m^{-1/6} \mathcal{L}_S(\widetilde{\mathbf{W}}),
$$

where in the first inequality we have used $|y_i \ell'(y_i f_{\widetilde{\mathbf{W}}}(\mathbf{x}_i))| \leq \ell(y_i f_{\widetilde{\mathbf{W}}}(\mathbf{x}_i))$. The proof is completed. $\qquad \square$

The following lemma shows how the distance between gradient descent iterators and the reference model would change after a single gradient descent.

**Lemma 21.** *Suppose $m \gtrsim L^{10} \log(nL/\delta)(\log m)^4 R^2$. Then with probability at least $1 - \delta$, for $\eta \le 4/(5L)$ and $\widetilde{\mathbf{W}}, \mathbf{W} \in \mathcal{B}_R(\mathbf{W}(0))$,*

$$\left\| \mathbf{W} - \eta \frac{\partial \mathcal{L}_S(\mathbf{W})}{\partial \mathbf{W}} - \widetilde{\mathbf{W}} \right\|_F^2 \le \|\mathbf{W} - \widetilde{\mathbf{W}}\|_F^2 - 2\eta(\mathcal{L}_S(\mathbf{W}) - \mathcal{L}_S(\widetilde{\mathbf{W}}))$$
$$+ 2\eta C L^{8/3}(\log m)^{2/3} R^{4/3} m^{-1/6} \mathcal{L}_S(\mathbf{W}) + 20\eta^2 L \tilde{F}_S^2(\widetilde{\mathbf{W}}).$$

*Proof.* By Lemma 8 and Lemma 18 we know that $\left\| \frac{\partial f_{\mathbf{w}}(\mathbf{x}_i)}{\partial \mathbf{W}^l} \right\|_F \le 2$, hence $\left\| \frac{\partial f_{\mathbf{w}}(\mathbf{x}_i)}{\partial \mathbf{W}} \right\|_F \le 2\sqrt{L}$, which implies that

$$\left\| \frac{\partial \mathcal{L}_S(\mathbf{W})}{\partial \mathbf{W}} \right\|_F^2 = \left\| \frac{1}{n} \sum_{i=1}^n y_i \ell'(y_i f_{\mathbf{w}}(\mathbf{x}_i)) \frac{\partial f_{\mathbf{w}}(\mathbf{x}_i)}{\partial \mathbf{W}} \right\|_F^2$$

$$\le \left( \frac{1}{n} \sum_{i=1}^n |\ell'(y_i f_{\mathbf{w}}(\mathbf{x}_i))| \left\| \frac{\partial f_{\mathbf{w}}(\mathbf{x}_i)}{\partial \mathbf{W}} \right\|_F \right)^2$$

$$\le 4L \left( \frac{1}{n} \sum_{i=1}^n |\ell'(y_i f_{\mathbf{w}}(\mathbf{x}_i))| \right)^2$$

$$\le 5L \left( \frac{1}{n} \sum_{i=1}^n (|\ell'(y_i f_{\mathbf{w}}(\mathbf{x}_i))| - |\ell'(y_i f_{\widetilde{\mathbf{W}}}(\mathbf{x}_i))|) \right)^2 + 20L \left( \frac{1}{n} \sum_{i=1}^n |\ell'(y_i f_{\widetilde{\mathbf{W}}}(\mathbf{x}_i))| \right)^2,$$

where we have used the standard inequality $(a + b)^2 \le 5a^2 + 5b^2/4$. Then by applying Lemma 20 we have

$$\left\| \mathbf{W} - \eta \frac{\partial \mathcal{L}_S(\mathbf{W})}{\partial \mathbf{W}} - \widetilde{\mathbf{W}} \right\|_F^2$$

$$= \|\mathbf{W} - \widetilde{\mathbf{W}}\|_F^2 + \eta^2 \left\| \frac{\partial \mathcal{L}_S(\mathbf{W})}{\partial \mathbf{W}} \right\|_F^2 - 2\eta \left\langle \mathbf{W} - \widetilde{\mathbf{W}}, \frac{\partial \mathcal{L}_S(\mathbf{W})}{\partial \mathbf{W}} \right\rangle$$

$$\le \|\mathbf{W} - \widetilde{\mathbf{W}}\|_F^2 + 5\eta^2 L \left( \frac{1}{n} \sum_{i=1}^n (|\ell'(y_i f_{\mathbf{w}}(\mathbf{x}_i))| - |\ell'(y_i f_{\widetilde{\mathbf{W}}}(\mathbf{x}_i))|) \right)^2$$

$$+ 20\eta^2 L \left( \frac{1}{n} \sum_{i=1}^n |\ell'(y_i f_{\widetilde{\mathbf{W}}}(\mathbf{x}_i))| \right)^2 - 2\eta(\mathcal{L}_S(\mathbf{W}) - \mathcal{L}_S(\widetilde{\mathbf{W}})) - \frac{4\eta}{n} \sum_{i=1}^n (l'(y_i f_{\widetilde{\mathbf{W}}}(\mathbf{x}_i) - l'(y_i f_{\mathbf{w}}(\mathbf{x}_i)))^2$$

$$+ \eta C L^{8/3}(\log m)^{2/3} R^{4/3} m^{-1/6} \mathcal{L}_S(\mathbf{W}).$$

Since $\eta \le 4/(5L)$ and

$$\left( \frac{1}{n} \sum_{i=1}^n (|\ell'(y_i f_{\mathbf{w}}(\mathbf{x}_i))| - |\ell'(y_i f_{\widetilde{\mathbf{W}}}(\mathbf{x}_i))|) \right)^2 \le \frac{1}{n} \sum_{i=1}^n (l'(y_i f_{\widetilde{\mathbf{W}}}(\mathbf{x}_i) - l'(y_i f_{\mathbf{w}}(\mathbf{x}_i)))^2,$$

the proof is completed. $\qquad\square$

*Proof of Theorem 1.* We prove it through induction. It holds for $t = 0$. Suppose it holds for $k = 0, \cdots, t - 1$, then we have,

$$\max_l \|\mathbf{W}^l(k) - \mathbf{W}^l(0)\|_2 \le \|\mathbf{W}^l(k) - \overline{\mathbf{W}^l}\|_2 + \|\mathbf{W}^l(0) - \overline{\mathbf{W}^l}\|_2 \le 2\sqrt{F_S(\overline{\mathbf{W}})}.$$

plugging $R = 2\sqrt{F_S(\overline{\mathbf{W}})}$ and $m$ into Lemma 21, we have

$$\|\mathbf{W}(k+1) - \overline{\mathbf{W}}\|_F^2 \le \|\mathbf{W}(k) - \overline{\mathbf{W}}\|_F^2 - 2\eta(\mathcal{L}_S(\mathbf{W}(k)) - \mathcal{L}_S(\overline{\mathbf{W}}))$$
$$+ 2\eta C L^{8/3}(\log m)^{2/3} F_S^{2/3}(\overline{\mathbf{W}}) m^{-1/6} \mathcal{L}_S(\mathbf{W}(k)) + 20\eta^2 L \tilde{F}_S^2(\overline{\mathbf{W}}).$$

Telescoping and note that $\tilde{F}_S(\overline{\mathbf{W}}) \leq \mathcal{L}_S(\overline{\mathbf{W}})$, we obtain

$$\|\mathbf{W}(t) - \overline{\mathbf{W}}\|_F^2 + 2\eta \sum_{k=0}^{t-1}(\mathcal{L}_S(\mathbf{W}(k)) - \mathcal{L}_S(\overline{\mathbf{W}})) \leq \|\mathbf{W}(0) - \overline{\mathbf{W}}\|_F^2$$

$$+2\eta CL^{8/3}(\log m)^{2/3}F_S^{2/3}(\overline{\mathbf{W}})m^{-1/6}\sum_{k=0}^{t-1}\mathcal{L}_S(\mathbf{W}(k)) + 20\eta^2 LT\tilde{F}_S(\overline{\mathbf{W}})\mathcal{L}_S(\overline{\mathbf{W}}),$$

which implies

$$\|\mathbf{W}(t) - \overline{\mathbf{W}}\|_F^2 + 2\eta \sum_{k=0}^{t-1}(\mathcal{L}_S(\mathbf{W}(k))(1 - CL^{8/3}(\log m)^{2/3}F_S^{2/3}(\overline{\mathbf{W}})m^{-1/6})$$

$$\leq \|\mathbf{W}(0) - \overline{\mathbf{W}}\|_F^2 + (2 + 20\eta L\tilde{F}_S(\overline{\mathbf{W}}))\eta T\mathcal{L}_S(\overline{\mathbf{W}}).$$

Hence, when $m \gtrsim F_S^4(\overline{\mathbf{W}})(\log m)^4 L^{16}, \eta \leq 1/(20L\tilde{F}_S(\overline{\mathbf{W}}))$, there holds

$$\|\mathbf{W}(t) - \overline{\mathbf{W}}\|_F^2 + \eta \sum_{k=0}^{t-1}(\mathcal{L}_S(\mathbf{W}(k)) \leq F_S(\overline{\mathbf{W}}).$$

This implies that

$$\|\mathbf{W}^l(t) - \overline{\mathbf{W}^l}\|_2 \leq \|\mathbf{W}(t) - \overline{\mathbf{W}}\|_F \leq F_S(\overline{\mathbf{W}}), \quad \eta \sum_{k=0}^{t-1}(\mathcal{L}_S(\mathbf{W}(k)) \leq F_S(\overline{\mathbf{W}}).$$

Therefore, the induction holds for $t$, the proof is completed. $\qquad\square$

## C  Proofs for Generalization

From (35), we know that there exists $\widehat{\mathbf{G}}_{L,0}^l(\mathbf{x}_i), 1 \leq l \leq L, i \in [n]$, such that

$$h^L(\mathbf{x}_i) - h_0^L(\mathbf{x}_i) = \sum_{l=1}^{L}\widehat{\mathbf{G}}_{L,0}^l(\mathbf{x}_i)(\mathbf{W}^l - \mathbf{W}^l(0))h_0^{l-1}(\mathbf{x}_i).$$

Let $E_1$ be the event (w.r.t. $\mathbf{W}(0)$) that $\|\widehat{\mathbf{G}}_{L,0}^l(\mathbf{x}_i)\|_2 \leq CL\sqrt{\log m/m}$ for all $\mathbf{W} \in \mathcal{W}_1$. Let $E_2$ be the event such that $\|h_0^{l-1}(\mathbf{x}_i)\|_2^2 \leq 4/3, 1 \leq l \leq L, i \in [n]$. Then we have the following bound on Rademacher complexity:

**Lemma 22.** *Let $\mathcal{F}$ and $\mathcal{W}_1$ be defined in (7) and (8), respectively. If the events $E_1, E_2$ hold, then*

$$\mathfrak{R}_{S_1,n}(\mathcal{F}) \leq CL^2\sqrt{\frac{F(\overline{\mathbf{W}})\log m}{n}}, \tag{43}$$

*where*

$$\mathfrak{R}_{S_1,n}(\mathcal{F}) = \sup_{\widetilde{S} \subset S_1 : |\widetilde{S}| = n} \mathfrak{R}_{\widetilde{S}}(\mathcal{F}).$$

*Proof.* Let $\widetilde{S} = \{\tilde{\mathbf{x}}_1, \cdots, \tilde{\mathbf{x}}_n\}$. Then we have

$$\mathfrak{R}_{\widetilde{S}}(\mathcal{F}) = \mathbb{E}_\epsilon\left[\sup_{\mathbf{W}\in\mathcal{W}_1}\frac{1}{n}\sum_{i=1}^{n}\epsilon_i f_{\mathbf{W}}(\tilde{\mathbf{x}}_i)\right]$$

$$\leq \mathbb{E}_\epsilon\left[\sup_{\mathbf{W}\in\mathcal{W}_1}\frac{1}{n}\sum_{i=1}^{n}\epsilon_i(f_{\mathbf{W}}(\tilde{\mathbf{x}}_i) - f_{\mathbf{W}(0)}(\tilde{\mathbf{x}}_i))\right] + \mathbb{E}_\epsilon\left[\sup_{\mathbf{W}\in\mathcal{W}_1}\frac{1}{n}\sum_{i=1}^{n}\epsilon_i f_{\mathbf{W}(0)}(\tilde{\mathbf{x}}_i)\right]$$

$$= \mathbb{E}_\epsilon\left[\sup_{\mathbf{W}\in\mathcal{W}_1}\frac{1}{n}\sum_{i=1}^{n}\epsilon_i(f_{\mathbf{W}}(\tilde{\mathbf{x}}_i) - f_{\mathbf{W}(0)}(\tilde{\mathbf{x}}_i))\right] = \mathbb{E}_\epsilon\left[\sup_{\mathbf{W}\in\mathcal{W}_1}\frac{1}{n}\sum_{i=1}^{n}\epsilon_i \mathbf{a}^\top(h^L(\tilde{\mathbf{x}}_i) - h_0^L(\tilde{\mathbf{x}}_i))\right]$$

$$= \frac{1}{n}\mathbb{E}_\epsilon\left[\sup_{\mathbf{W}\in\mathcal{W}_1}\mathbf{a}^\top\sum_{i=1}^{n}\epsilon_i(h^L(\tilde{\mathbf{x}}_i) - h_0^L(\tilde{\mathbf{x}}_i))\right]$$

$$\leq \frac{\|\mathbf{a}\|_2}{n}\mathbb{E}_\epsilon\sup_{\mathbf{W}\in\mathcal{W}_1}\left\|\sum_{i=1}^{n}\epsilon_i(h^L(\tilde{\mathbf{x}}_i) - h_0^L(\tilde{\mathbf{x}}_i))\right\|_2, \tag{44}$$

where the last inequality is due to Cauchy-Schwartz inequality. Since the event $E_1$ holds, we have for all $\mathbf{W} \in \mathcal{W}_1, 1 \le k \le l \le L, i \in [n]$,

$$\|\widehat{\mathbf{G}}_{L,0}^l(\tilde{\mathbf{x}}_i)(\mathbf{W}^l - \mathbf{W}^l(0))\|_F \le \|\widehat{\mathbf{G}}_{L,0}^l(\tilde{\mathbf{x}}_i)\|_2 \|\mathbf{W}^l - \mathbf{W}^l(0)\|_F \le CL\sqrt{\frac{F(\overline{\mathbf{W}})\log m}{m}},$$

where we have used $\|AB\|_F \le \|A\|_2 \|B\|_F$ for two matrices $A, B$. Therefore,

$$\mathbb{E}_\epsilon \sup_{\mathbf{W} \in \mathcal{W}_1} \left\| \sum_{i=1}^n \epsilon_i (h^L(\tilde{\mathbf{x}}_i) - h_0^L(\tilde{\mathbf{x}}_i)) \right\|_2$$

$$= \mathbb{E}_\epsilon \sup_{\mathbf{W} \in \mathcal{W}_1} \left\| \sum_{i=1}^n \epsilon_i \sum_{l=1}^L \widehat{\mathbf{G}}_{L,0}^l(\tilde{\mathbf{x}}_i)(\mathbf{W}^l - \mathbf{W}^l(0)) h_0^{l-1}(\tilde{\mathbf{x}}_i) \right\|_2$$

$$\le \sum_{l=1}^L \mathbb{E}_\epsilon \sup_{\mathbf{W} \in \mathcal{W}_1} \left\| \sum_{i=1}^n \epsilon_i \widehat{\mathbf{G}}_{L,0}^l(\tilde{\mathbf{x}}_i)(\mathbf{W}^l - \mathbf{W}^l(0)) h_0^{l-1}(\tilde{\mathbf{x}}_i) \right\|_2.$$

For $l \in [L]$, let $(g_1^l)^\top, \cdots, (g_m^l)^\top$ be the rows of matrix $\widehat{\mathbf{G}}_{L,0}^l(\tilde{\mathbf{x}}_i)(\mathbf{W}^l - \mathbf{W}^l(0))$, we obtain

$$\left\| \sum_{i=1}^n \epsilon_i \widehat{\mathbf{G}}_l^k(\tilde{\mathbf{x}}_i)(\mathbf{W}^l - \mathbf{W}^l(0)) h_0^{l-1}(\tilde{\mathbf{x}}_i) \right\|_2^2$$

$$= \sum_{r=1}^m \|g_r^l\|_2^2 \left( \sum_{i=1}^n \epsilon_i \frac{(g_r^l)^\top}{\|g_r^l\|_2} h_0^{l-1}(\tilde{\mathbf{x}}_i) \right)^2$$

$$\le \sum_{r=1}^m \|g_r^l\|_2^2 \sup_{\mathbf{u}: \|\mathbf{u}\|_2 = 1} \left( \sum_{i=1}^n \epsilon_i \mathbf{u}^\top h_0^{l-1}(\tilde{\mathbf{x}}_i) \right)^2$$

$$= \sum_{r=1}^m \|g_r^l\|_2^2 \left\| \sum_{i=1}^n \epsilon_i h_0^{l-1}(\tilde{\mathbf{x}}_i) \right\|_2^2$$

$$= \|\widehat{\mathbf{G}}_{L,0}^l(\tilde{\mathbf{x}}_i)(\mathbf{W}^l - \mathbf{W}^l(0))\|_F^2 \left\| \sum_{i=1}^n \epsilon_i h_0^{l-1}(\tilde{\mathbf{x}}_i) \right\|_2^2.$$

Therefore,

$$\mathbb{E}_\epsilon \sup_{\mathbf{W} \in \mathcal{W}_1} \left\| \sum_{i=1}^n \epsilon_i \widehat{\mathbf{G}}_{L,0}^l(\tilde{\mathbf{x}}_i)(\mathbf{W}^l - \mathbf{W}^l(0)) h_0^{l-1}(\tilde{\mathbf{x}}_i) \right\|_2$$

$$= CL\sqrt{\frac{F(\overline{\mathbf{W}})\log m}{m}} \mathbb{E}_\epsilon \left\| \sum_{i=1}^n \epsilon_i h_0^{l-1}(\tilde{\mathbf{x}}_i) \right\|_2.$$

This implies that

$$
\mathbb{E}_\epsilon \sup_{\mathbf{W} \in \mathcal{W}_1} \left\| \sum_{i=1}^n \epsilon_i (h^L(\tilde{\mathbf{x}}_i) - h_0^L(\tilde{\mathbf{x}}_i)) \right\|_2
$$

$$
\leq \sum_{l=1}^L \mathbb{E}_\epsilon \sup_{\mathbf{W} \in \mathcal{W}_1} \| \widehat{\mathbf{G}}_{L,0}^l(\tilde{\mathbf{x}}_i)(\mathbf{W}^l - \mathbf{W}^l(0)) \|_F \left\| \sum_{i=1}^n \epsilon_i h_0^{l-1}(\tilde{\mathbf{x}}_i) \right\|_2
$$

$$
\leq CL \sqrt{\frac{F(\overline{\mathbf{W}}) \log m}{m}} \sum_{l=1}^L \mathbb{E}_\epsilon \left\| \sum_{i=1}^n \epsilon_i h_0^{l-1}(\tilde{\mathbf{x}}_i) \right\|_2
$$

$$
\leq CL \sqrt{\frac{F(\overline{\mathbf{W}}) \log m}{m}} \sum_{l=1}^L \sqrt{\mathbb{E}_\epsilon \left\| \sum_{i=1}^n \epsilon_i h_0^{l-1}(\tilde{\mathbf{x}}_i) \right\|_2^2}
$$

$$
= CL \sqrt{\frac{F(\overline{\mathbf{W}}) \log m}{m}} \sum_{l=1}^L \sqrt{\sum_{i=1}^n \left\| h_0^{l-1}(\tilde{\mathbf{x}}_i) \right\|_2^2} \leq CL^2 \sqrt{\frac{F(\overline{\mathbf{W}}) n \log m}{m}}.
$$

The last inequality is due to the event $E_2$. Combined with (44), we have

$$
\mathfrak{R}_{\widetilde{S}}(\mathcal{F}) \leq \frac{\|\mathbf{a}\|_2}{n} \mathbb{E}_\epsilon \sup_{\mathbf{W} \in \mathcal{W}_1} \left\| \sum_{i=1}^n \epsilon_i (h^L(\tilde{\mathbf{x}}_i) - h_0^L(\tilde{\mathbf{x}}_i)) \right\|_2 \leq CL^2 \sqrt{\frac{F(\overline{\mathbf{W}}) \log m}{n}}.
$$

The proof is completed. $\qquad\square$

Now we provide the proof for Theorem 2

*Proof of Theorem 2.* We first control $G' = \sup_{z, \mathbf{W} \in \mathcal{W}_1} \ell(y f_{\mathbf{W}}(\mathbf{x}))$. We denote $\bar{h}^l(\mathbf{x})$ as the output of $l$-th layer of the network $f_{\overline{\mathbf{W}}}(\mathbf{x})$. Then $f_{\overline{\mathbf{W}}}(\mathbf{x}) = \mathbf{a}^\top \bar{h}^L(\mathbf{x})$. By the definition of $F(\overline{\mathbf{W}})$, we have $\max_l \|\overline{\mathbf{W}}^l - \mathbf{W}^l(0)\|_2 \leq \|\overline{\mathbf{W}} - \mathbf{W}(0)\|_F \leq \sqrt{F(\overline{\mathbf{W}})}$. For $\mathbf{W} \in \mathcal{W}_1$, there holds $\|\mathbf{W}^l - \mathbf{W}^l(0)\|_2 \leq \|\overline{\mathbf{W}}^l - \mathbf{W}^l(0)\|_2 + \|\mathbf{W}^l - \overline{\mathbf{W}}^l\|_2 \leq 2\sqrt{F(\overline{\mathbf{W}})}$ for all $l \in [L]$. By the overparameterization of $m$ and Lemma 16,

$$
(f_{\mathbf{W}}(\mathbf{x}) - f_{\overline{\mathbf{W}}}(\mathbf{x}))^2 = (\mathbf{a}^\top (h^L(\mathbf{x}) - \bar{h}^L(\mathbf{x})))^2 \leq m \|h^L(\mathbf{x}) - \bar{h}^L(\mathbf{x})\|_2^2
$$
$$
\leq 2m (\|h^L(\mathbf{x}) - h_0^L(\mathbf{x})\|_2^2 + \|h_0^L(\mathbf{x}) - \bar{h}^L(\mathbf{x})\|_2^2) \leq CL^4 \log m F(\overline{\mathbf{W}}).
$$

Since logistic loss $\ell$ is $1/4$-smooth, the following property holds,

$$
|\ell'(x)| \leq \sqrt{\ell(x)/2}, \quad x \in \mathbb{R}.
$$

It then follows that for any $\mathbf{W} \in \mathcal{W}_1, z \in \mathcal{Z}$,

$$
\ell(y f_{\mathbf{W}}(\mathbf{x})) \leq \ell(y f_{\overline{\mathbf{W}}}(\mathbf{x})) + y(f_{\mathbf{W}}(\mathbf{x}) - f_{\overline{\mathbf{W}}}(\mathbf{x}))\ell'(y f_{\overline{\mathbf{W}}}(\mathbf{x})) + \frac{(f_{\mathbf{W}}(\mathbf{x}) - f_{\overline{\mathbf{W}}}(\mathbf{x}))^2}{8}
$$
$$
\leq \ell(y f_{\overline{\mathbf{W}}}(\mathbf{x})) + 2|\ell'(y f_{\overline{\mathbf{W}}}(\mathbf{x}))|^2 + \frac{(f_{\mathbf{W}}(\mathbf{x}) - f_{\overline{\mathbf{W}}}(\mathbf{x}))^2}{4}
$$
$$
\leq 2\ell(y f_{\overline{\mathbf{W}}}(\mathbf{x})) + CL^4 \log m F(\overline{\mathbf{W}}),
$$

where we have used $ab \leq 2a^2 + b^2/8$ in the second inequality. Hence, $G' \leq 2G + CL^4 \log m F(\overline{\mathbf{W}})$. According to Lemma 14, we have with probability at least $1 - \delta$,

$$
|\mathcal{L}_S(\overline{\mathbf{W}}) - \mathcal{L}(\overline{\mathbf{W}})| \leq \left( \frac{2G\mathcal{L}(\overline{\mathbf{W}}) \log(2/\delta)}{n} \right)^{1/2} + \frac{2G \log(2/\delta)}{3n}. \tag{45}
$$

It then follows that

$$
\mathcal{L}_S(\overline{\mathbf{W}}) \leq 2\mathcal{L}(\overline{\mathbf{W}}) + \frac{7G \log(2/\delta)}{6n},
$$

which implies that $F_S(\overline{\mathbf{W}}) \leq F(\overline{\mathbf{W}})$. Combined with Theorem 1, we know that with probability at least $1 - \delta$, $\mathbf{W}(t) \in \mathcal{W}_1$. It means that all the iterates are in the hypothsesis space. Furthermore, events $E_1, E_2$ hold due to Lemma 6 and (37) in Lemma 15. Hence, by Lemma 1 and Lemma 22, there holds

$$\mathcal{L}(\mathbf{W}(t)) - 2\mathcal{L}_S(\mathbf{W}(t)) \lesssim (\log n)^3 \mathfrak{R}_{S,n}^2(\mathcal{F}) + \frac{G' \log(2/\delta)}{n}$$

$$= \widetilde{O}\left( \frac{L^4 F(\overline{\mathbf{W}}) + G \log(2/\delta)}{n} \right).$$

As a result,

$$\eta \sum_{t=0}^{T-1} \mathcal{L}(\mathbf{W}(t)) = \eta \sum_{t=0}^{T-1} \left( L(\mathbf{W}(t) - 2\mathcal{L}_S(\mathbf{W}(t)) + 2\eta \sum_{t=0}^{T-1} \mathcal{L}_S(\mathbf{W}(t)) \right)$$

$$= \widetilde{O}\left( \frac{(\eta T L^4 + n) F(\overline{\mathbf{W}}) + \eta T G \log(2/\delta)}{n} \right),$$

from which we derive

$$\frac{1}{T} \sum_{t=0}^{T-1} \mathcal{L}(\mathbf{W}(t)) = \widetilde{O}\left( \frac{L^4 F(\overline{\mathbf{W}}) + G \log(2/\delta)}{n} \right),$$

where we have used $\eta T \asymp n$. The proof is completed. $\qquad\square$

## D  Proofs on NTK separability

*Proof of Theorem 3.* We show that there exists $\overline{\mathbf{W}}$ with small $F(\overline{\mathbf{W}})$ for NTK separable data. Let $\overline{\mathbf{W}} = \mathbf{W}(0) + \lambda \mathbf{W}_*$. Choose $\lambda = 2 \log T / \gamma$. Applying Lemma 19 and letting $R = \lambda$, we know that if $m \gtrsim L^{16} d (\log m)^5 \log(nL/\delta)(\log T)^2 / \gamma^8$, then with probability at least $1 - \delta$, for all $i \in [n]$, there holds

$$\left| \left\langle \lambda \mathbf{W}_*, \frac{\partial f_{\mathbf{W}(0)}(\mathbf{x}_i)}{\partial \mathbf{W}(0)} \right\rangle - f_{\overline{\mathbf{W}}}(\mathbf{x}_i) \right|$$

$$= \left| f_{\mathbf{W}(0)}(\mathbf{x}_i) - f_{\overline{\mathbf{W}}}(\mathbf{x}_i) - \left\langle \mathbf{W}(0) - \overline{\mathbf{W}}, \frac{\partial f_{\mathbf{W}(0)}(\mathbf{x}_i)}{\partial \mathbf{W}(0)} \right\rangle \right|$$

$$\leq C L^{8/3} \lambda^{4/3} m^{-1/6} (\log m)^{2/3} \leq \frac{\lambda \gamma}{2},$$

where we have used $f_{\mathbf{W}(0)}(\mathbf{x}) = 0$ for any $\mathbf{x} \in \mathcal{X}$ due to Lemma 7. Therefore, by Assumption 3, we have

$$y_i f_{\overline{\mathbf{W}}}(\mathbf{x}_i) = y_i \left\langle \lambda \mathbf{W}_*, \frac{\partial f_{\mathbf{W}(0)}(\mathbf{x}_i)}{\partial \mathbf{W}(0)} \right\rangle - y_i \left( \left\langle \lambda \mathbf{W}_*, \frac{\partial f_{\mathbf{W}(0)}(\mathbf{x}_i)}{\partial \mathbf{W}(0)} \right\rangle - f_{\overline{\mathbf{W}}}(\mathbf{x}_i) \right)$$

$$\geq \lambda y_i \left\langle \mathbf{W}_*, \frac{\partial f_{\mathbf{W}(0)}(\mathbf{x}_i)}{\partial \mathbf{W}(0)} \right\rangle - \left| \left\langle \lambda \mathbf{W}_*, \frac{\partial f_{\mathbf{W}(0)}(\mathbf{x}_i)}{\partial \mathbf{W}(0)} \right\rangle - f_{\overline{\mathbf{W}}}(\mathbf{x}_i) \right|$$

$$\geq \lambda \gamma - \frac{\lambda \gamma}{2} = \frac{\lambda \gamma}{2} = \log T.$$

As a result,

$$\mathcal{L}_S(\overline{\mathbf{W}}) = \frac{1}{n} \sum_{i=1}^{n} \ell(-y_i f_{\overline{\mathbf{W}}}(\mathbf{x}_i)) \leq \log(1 + \exp(-\log T)) \leq \frac{1}{T},$$

where we have used $\log(1 + x) \leq x$. For any $\mathbf{x} \in \mathcal{X}$, Lemma 16 implies that

$$y f_{\overline{\mathbf{W}}}(\mathbf{x}) \geq -|f_{\overline{\mathbf{W}}}(\mathbf{x})| \geq -|f_{\mathbf{W}(0)}(\mathbf{x})| - |f_{\overline{\mathbf{W}}}(\mathbf{x}) - f_{\mathbf{W}(0)}(\mathbf{x})|$$

$$\geq -\mathbf{a}^\top \|\bar{h}^L(\mathbf{x}) - h_0^L(\mathbf{x})\|_2 \geq -C L^2 \sqrt{\log m} \lambda.$$

Hence,

$$G = \sup_z \ell(y f_{\overline{\mathbf{W}}}(\mathbf{x})) \lesssim \log(1 + \exp(L^2 \sqrt{\log m} \log T / \gamma)) \lesssim \frac{L^2 \sqrt{\log m} \log T}{\gamma},$$

where the last inequality is due to $\log(1+t) \le \log(2t) \le 2\log(t)$ for $t \ge 2$. Note that if $x^2 \le \alpha x + \beta$, then $x^2 \le \alpha^2 + 2\beta$. Combined with (45), it then follows that (let $x = \sqrt{\mathcal{L}(\overline{\mathbf{W}})}$)

$$\mathcal{L}(\overline{\mathbf{W}}) \le 2\mathcal{L}_S(\overline{\mathbf{W}}) + \frac{4G\log(2/\delta)}{3n} + \frac{2G\log(2/\delta)}{n} \lesssim \frac{1}{T} + \frac{L^2\sqrt{\log m}\log T \log(2/\delta)}{\gamma n}.$$

Thus, we have

$$F(\overline{\mathbf{W}}) \lesssim 3\eta T \left( \frac{1}{T} + \frac{L^2\sqrt{\log m}\log T \log(2/\delta)}{\gamma n} \right) + \lambda^2 = \widetilde{O}\left( \frac{(\log T)^2(1+\gamma L^2)}{\gamma^2} \right). \tag{46}$$

Applying Theorem 2 and note that $\widetilde{F}_S(\overline{\mathbf{W}}) \le \mathcal{L}_S(\overline{\mathbf{W}}) \le \frac{1}{T}$, there holds

$$\frac{1}{T} \sum_{t=0}^{T-1} \mathcal{L}(\mathbf{W}(t)) = \widetilde{O}\left( \frac{L^4(\log T)^2(1+\gamma L^2)}{n\gamma^2} \right).$$

The proof is completed. $\square$

