# OpenReview forum: "Optimal Rates  for Generalization of Gradient Descent for Deep ReLU Classification"
_NeurIPS.cc/2025/Conference — NeurIPS 2025 poster_

### Official Review · Reviewer_e2P1 · 2025-06-09

**Clarity:** 3
**Significance:** 2
**Originality:** 2
**Rating:** 5
**Confidence:** 4

**Summary:**

Improved bounds on the excess risk rate for deep ReLU networks.

**Questions:**

- gradient decent -> gradient descent, line 57

**Ethical Concerns:**

["NO or VERY MINOR ethics concerns only"]

**Final Justification:**

I confirm the positive evaluation. The point of saying that their contribution is only theoretical is weak, since to my knowledge no work in NeurIPS is only theoretical. The authors could have missed a mathematical step, that the reviewers could have also missed, so a few experimental confirmations never hurt. To my understanding however the article is strong enough to be accepted.

**Limitations:**

The limitations are the assumptions of their theorems, which are clearly stated.

**Paper Formatting Concerns:**

Correct.

**Quality:**

3

**Strengths And Weaknesses:**

Strengths. Improved bounds wrt those known that are polynomial in depth.
Weaknesses. No attempt at an experimental confirmation, but they are on time to present it for the rebuttal.

---

> ### Author Rebuttal · Authors · 2025-07-31
>
> Thanks for acknowledging the strengths of our work. Our point-to-point response to your comments follows next.
> >**Strengths And Weaknesses** Strengths. Improved bounds wrt those known that are polynomial in depth. Weaknesses. No attempt at an experimental confirmation.
>
> - The improvement over $L$ is **only part of** our contribution. The main result is that we are **the first to prove that deep ReLU networks trained by gradient descent can achieve optimal rate** ($1/n$). The existing works either obtain a bound of suboptimal rate $1/\sqrt{n}$ or focus on smooth activations.
> - This is a paper on **learning theory**. We focus on whether and how deep ReLU networks could achieve optimal generalization error. Hence, we do not take experiments. We believe we have improved the understanding of deep neural networks **theoretically**.
>
> >**Q.** gradient decent -> gradient descent, line 57
>
> Thank you for pointing it out. We will revise it in the next version.

---

### Official Review · Reviewer_usBV · 2025-06-25

**Clarity:** 3
**Significance:** 2
**Originality:** 2
**Rating:** 4
**Confidence:** 3

**Summary:**

This paper studies the convergence and generalization analysis of deep fully connected neural networks. Specifically, this paper proves that GD achieves a sublinear convergence rate in the order of $1/T$. Furthermore, the derived generalization bound exhibits an optimal rate of $1/n$ with respect to the sample size $n$. Additionally, the dependence on $L$ has been improved from an exponential to a polynomial order by establishing a tighter Rademacher complexity bound.

**Questions:**

Since the major contribution of this paper is the improvement in the dependence of the generalization bound with respect to $L$, I have several questions:

1. What are the key insights that this paper aims to convey through this improved bound?

2. If Assumption 3 holds for real-world data, what would be the practical difference between using a deep neural network versus a shallow one? For example, how does $L$ relate to $\gamma$ in this context?

3. Could you specify lines that reflect "novel control of activation patterns near a reference model"?

**Ethical Concerns:**

["NO or VERY MINOR ethics concerns only"]

**Final Justification:**

I increased the score from 3 to 4 because the author has addressed my concerns regarding the gap between the theoretical results and real applications by providing numerical experiments. However, I still have concerns about Assumption 3, which deviates significantly from real-world applications.

**Limitations:**

Yes

**Quality:**

3

**Strengths And Weaknesses:**

Strength:

1. A tighter and more practically relevant generalization bound.

2. A thorough comparison with existing works.

Weakness:

Overall, I feel that while this paper shows improvements over prior work, it focuses on a less meaningful direction. The insights gained from this improvement are either unclear or do not align well with what is observed in practice. As a result, I am not convinced that this setting is appropriate for studying the generalization error of deep neural networks.

1. The assumption that the data is NTK-separable with a margin $\gamma$ lacks explanation in practical contexts. Specifically, what kinds of data will succeed or fail in this assumption? In addition, this assumption is extremely strong in the sense that it requires the network to be initialized in such a way that the data, when processed through this predefined model, is already linearly separable.

2. Compared with prior work, the main difference I observe is that the generalization error bound has been improved from an exponential dependence to a polynomial dependence. However, the key insight remains the same as in previous studies: increasing the number of layers leads to worse generalization. I do not see any fundamentally new insights presented in this paper.

3. The theoretical insights presented in this paper do not adequately explain the current success and widespread effectiveness of deep learning models that rely on large and deep architectures.

4. The paper does not include any experiments to validate the theoretical results it presents.

---

> ### Author Rebuttal · Authors · 2025-07-31
>
> Thanks for acknowledging the strengths of our work. Our point-to-point response to your comments follows next.
>
> >**W2.&Q1.** Compared with prior work, the main difference I observe is that the generalization error bound has been improved from an exponential dependence to a polynomial dependence. However, the key insight remains the same as in previous studies: increasing the number of layers leads to worse generalization. I do not see any fundamentally new insights presented in this paper. What are the key insights that this paper aims to convey through this improved bound?
> - The improvement over $L$ is **only part of** our contribution. **Our key insight is that we are the first to prove that deep ReLU networks trained by gradient descent can achieve optimal rate** ($1/n$). The existing works either obtain a bound of suboptimal rate $1/\sqrt{n}$ or focus on smooth activations. For more details, please refer to Table 1 of our paper.
> - Despite the success of deep learning in practice, understanding it in theory is often challanging due to its nonconvexity and non-smoothness. We address these problems by **developing two novel techniques**. Firstly, we derive a tighter Rademacher complexity bound for deep ReLU networks based on the control of activation patterns. Secondly, we prove the $\widetilde{O}(L^2)$ Lipschitzness of deep ReLU networks. These key improvements open the door to further analysis of deep learning, which is beyond the scope of this paper.
>
> >**W1.** The assumption that the data is NTK-separable with a margin $\gamma$ lacks explanation in practical contexts. Specifically, what kinds of data will succeed or fail in this assumption? In addition, this assumption is extremely strong in the sense that it requires the network to be initialized in such a way that the data, when processed through this predefined model, is already linearly separable.
>
> We would like to clarify the motivation and interpretation of the NTK-separability assumption, which has been widely used in the literature [2,3,4,5].
>
> - **Role of the NTK separability assumption:**
>
> The NTK separability assumption is used in our paper as **a tool to analyze $F(\overline{W})$** in Theorem 2, which reflects the performance achievable by the network class. This assumption enables a clean characterization of generalization in Theorem 2. Importantly, our main result (Theorem 2) does **not** require data to be NTK-separable—it holds for any $F(\overline{W})$ with bounded complexity. Thus, NTK separability is not a limiting assumption of our theory, but rather a benchmark scenario to **make comparisons** with existing literature.
> - **We believe Assumption 3 is not a strong assumption**
>
> NTK separability has been **theoretically justified in a range of settings**. For instance, [2] analyzes the margin through its dual form in different settings, including linearly separable case and the noisy $2$-XOR distribution. [8] also shows that NTK can learn polynomials. Hence, Assumption 3 holds for data that are separable by polynomials. In practice, it can be verified through random features by solving a regularized linear classification problem.
>
> >**Q2.**  If Assumption 3 holds for real-world data, what would be the practical difference between using a deep neural network versus a shallow one? For example, how does $L$ relate to $\gamma$ in this context?
>
> This is a good question. In standard NTK theory, the margin $\gamma$ does **not explicitly depend on $L$**. However, we **conjecture** that deeper networks tend to induce larger NTK margins $\gamma$. Our intuition is based on the structure of the NTK and its associated **reproducing kernel Hilbert space (RKHS)**.
> - Let $\Theta^{(L)}$ denote the NTK of an $L$-layer ReLU network, and $H^{\infty}_L$ its corresponding RKHS. As shown in [6], the NTK satisfies the recursive relation: $\Theta^{(L+1)}(x,y) = \Theta^{(L)}(x,y)*\dot{\Sigma}^{(L+1)}(x,y) + \Sigma^{(L+1)}(x,y)$, where $\dot{\Sigma}^{(L+1)}$ is some derivative covirance matrix and $\Sigma^{(L+1)}$ is the covariance matrix of a Gaussian process. This implies that $\Theta^{(L+1)}$ contains $\Theta^{(L)}$ as part of its structure. Consequently, the RKHSs satisfy a nested relationship: $H^{\infty}_L \subset H^{\infty}_{L+1}$.
> - Consider the finite-width version: $$
> H^m_L = \{f : \exists\, W \in \mathcal{W} \text{ s.t. } f(x) = \langle W, \partial f_{W(0)}(x)/\partial W(0) \rangle \}.
> $$ As $m\to \infty, H^m_{L}\to H^\infty_{L}$ (see [7]). Therefore, $H^m_L \subset H^m_{L+1}$, and an $(L+1)$-layer deep network has a larger NTK-margin than an $L$-layer network, according to the definition of $\gamma$.
>
>
> Taken together, these suggest that **deeper networks admit richer RKHSs, which can better separate the data and thus lead to a larger margin $\gamma$**. Therefore, although $\gamma$ is not explicitly parameterized by $L$, it is indirectly affected through the expressiveness of the induced RKHS. We emphasize that this argument remains **heuristic**, and we leave a precise characterization of how $\gamma$ scales with $L$ as an important open question for future work.
>
> >**W3.** The theoretical insights presented in this paper do not adequately explain the current success and widespread effectiveness of deep learning models that rely on large and deep architectures.
>
> Our theoretical framework could not explain the effectiveness of large and deep models adequately, but we **provide a partial explanation** here.
> - In Theorem 2, we establish a generalization bound of the form $\widetilde{O}\left(\frac{L^4 F(\overline{W})}{n}\right)$, where $F(\overline{W})$ represents the risk of a well-structured (but fixed) reference model near initialization in the same network class. **As long as there exists a deep network with small $F(\overline{W})$**, gradient descent is guaranteed to find a predictor with low generalization error.
> - Importantly, the quantity $F(\overline{W})$ is directly tied to the **expressive power of the network class**. For more complex data distributions, a shallow network may not be able to achieve a small $F(\overline{W})$, while a deep architecture can do so due to its ability to capture hierarchical structures. In particular, existing results (e.g., [1]) show that deep networks can approximate certain function classes with exponentially fewer parameters than shallow ones. Therefore, our theory does capture the advantage of depth: **deeper architectures can produce smaller $F(\overline{W})$**, and Theorem 2 ensures that gradient descent can generalize well in such cases. We believe this provides an explanation for why deep models succeed in practice.
> - In addition, our analysis is conducted in the **overparameterized regime**, where the network width $m$ is large. Most of our results require $m \gtrsim \mathrm{poly}(L, \log n)$ to ensure good optimization behavior (e.g., descent direction, weak convexity). From this perspective, **large models also help with optimization**—they make gradient-based methods more effective, which partially explains why large-scale networks work well in practice.
>
> >**Q3.** Could you specify lines that reflect "novel control of activation patterns near a reference model"?
>
> - In Lemma 22, we express the Rademacher complexity via products of sparse matrices, whose norms are tightly controlled using optimization informed estimates. This helps to derive a sharper bound, line 704. Please also refer to remark 2, line 214.
> - Specifically, we first levearage the useful expression
> $f_W(x_i) - f_{W(0)}(x_i) = \mathbf{a}^\top \sum_{l=1}^L \widehat{G}_{L,0}^l(x_i) (W^l - W^l(0)) h_0^{l-1}(x_i).$
>
> Then we control $\widehat{G}_{L,0}^l(x_i)$ by Theorem 1 and Lemma 15. This leads to a tight bound of Rademacher complexity $\widetilde{O}\left(L^2\sqrt{\frac{F(\overline{W})}{n}}\right)$.
> - However, previous works use the bound on the shelf, ignoring how activation patterns would affect the complexity. For example, [4] derive the bound of $\widetilde{O}\left(4^LL\sqrt{\frac{mF(\overline{W})}{n}}\right)$, which suffers from exponential term and explicit dependence on $m$.
>
> >**W4.** The paper does not include any experiments to validate the theoretical results it presents.
>
> This is a paper on **learning theory**. We focus on whether and how deep ReLU networks could achieve optimal generalization error. Hence, we do not take experiments. We believe we have improved the understanding of deep neural networks **theoretically**.
>
> [1] Telgarsky et al. Benefits of depth in neural networks.
>
> [2] Ji et al. Polylogarithmic width suffices for gradient descent to achieve arbitrarily small test error with shallow relu networks.
>
> [3] Taheri et al. Generalization and stability of interpolating neural networks with minimal width.
>
> [4] Chen et al. How much over-parameterization is sufficient to learn deep relu networks?
>
> [5] Taheri et al. Sharper guarantees for learning neural networkclassifiers with gradient methods.
>
> [6] Arora et al. On Exact Computation with an Infinitely Wide Neural Net.
>
> [7] Xu et al. Overparametrized multi-layer neural networks: Uniform concentration of neural tangent kernel and convergence of stochastic gradient descent.
>
> [8] Arora et al. Fine-grained analysis of optimization and generalization for overparameterized two-layer neural networks.

---

> > ### Comment · Reviewer_usBV · 2025-08-05
> >
> > Thanks for the rebuttal. I need to point out that I respectfully disagree with the last point. If the theoretical results cannot be justified in real applications, how can we trust and utilize these theoretical insights to genuinely push understanding or guide the design of practical systems? As you mentioned, your goal is to improve the theoretical understanding of deep neural networks (DNNs). However, any results derived under assumptions that significantly deviate from real-world conditions risk being detached from practice. For example, if applying gradient-based methods in real-world applications cannot practically achieve a generalization error that scales on the order of $1/{\gamma^2 n}$, then what is the point of establishing such a result? While the theoretical bound might be mathematically elegant, if it cannot be observed under realistic training conditions or architectures, its relevance becomes questionable.
> >
> > Unless you can justify your assumptions and ensure that the model architecture aligns with real-world applications, it is essential to include numerical experiments to support any theoretical results based on those assumptions.

---

> > > ### Author Response · Authors · 2025-08-07
> > > **Response on Empirical Verifications**
> > >
> > > Thank you for your constructive comments. We agree that numerical experiments are valuable in supporting our theoretical analysis. Accordingly, we have conducted some experimental verifications to support our excess risk analysis.
> > >
> > > Our excess risk analysis in Theorem 3 imposes an NTK separability assumption, which has been validated in the literature. For example, the work [1] demonstrates that Assumption 3 holds for a noisy 2-XOR distribution, where the dataset is structured as follows:
> > > $$
> > > (x_1, x_2, y, \ldots, x_d) \in \\{
> > >   (\tfrac{1}{\sqrt{d-1}}, 0, 1),
> > >   (0, \tfrac{1}{\sqrt{d-1}}, -1), \ldots, (-\tfrac{1}{\sqrt{d-1}}, 0, 1),
> > >   (0, -\tfrac{1}{\sqrt{d-1}}, -1)
> > > \\}
> > >  \times \\{-\tfrac{1}{\sqrt{d-1}}, \tfrac{1}{\sqrt{d-1}}\\}^{d-2}.
> > > $$
> > > Here, the factor $\frac{1}{\sqrt{d-1}}$ ensures that $\|x\|_2 = 1$, $\times$ above denotes the Cartesian product, and the label $y$ only depends on the first two coordinates of the input $x$. As shown in [1], this dataset satisfies Assumption 3 with $1/\gamma=O(d)$, which implies that our excess risk bound in Theorem 3 becomes $O(d^2/n)$ for this dataset.  We conducted numerical experiments and observed that the test error decays linearly with $d^2/n$. The population loss for the test error is computed over all $2^d$ points in the distribution.
> > > - We train two-layer ReLU networks by gradient descent on noisy 2-XOR data. We fix the width $m=128, T=500,\eta = 0.1$.
> > > - With a fixed dimension $d = 6$, we vary the sample size $n$ and obtain the following results
> > >
> > > |$n$ |$d^2/n$|Test Error|
> > > |:----|:-------:|:----------|
> > > |  10 |  3.60 | 0.4625|
> > >  |   12 |  3.00 | 0.4500|
> > >  |   14 |  2.57 | 0.3625|
> > >  |   16 |  2.25 | 0.3438|
> > >  |   18 |  2.00 | 0.3219|
> > > |   20 |  1.80 | 0.3063|
> > >  |   24 |  1.50 | 0.2469|
> > >  |   28 |  1.29 | 0.2062|
> > >
> > >
> > >
> > >  - With a fixed sample size $n=64$, we vary the dimension $d$ and obtain the following results
> > >
> > >  |$d$ |$d^2/n$|Test Error|
> > > |:----|:-------:|:----------|
> > > | 7 |    0.77 | 0.0125|
> > > |8 |    1.00 | 0.1313|
> > > |9 |     1.27 | 0.2484|
> > > |10 |    1.56 | 0.3080|
> > > |11 |     1.89 | 0.3365|
> > > |12 |   2.25 | 0.4190|
> > >
> > > In both experiments, we observe that the test error is of the order $d^2/n$ (approximately $0.15d^2/n$). This shows the consistency between our excess risk bounds in Theorem 3 and experimental results. We will include these numerical experiments in the revised version. We will also conduct more experimental verifications.
> > >
> > > [1] Ji, Z., & Telgarsky, M. (2019). Polylogarithmic width suffices for gradient descent to achieve arbitrarily small test error with shallow relu networks.

---

> > > > ### Comment · Reviewer_usBV · 2025-08-09
> > > >
> > > > Thank you for the additional experiments. I have increased my score from 3 to 4.

---

> > > > > ### Author Response · Authors · 2025-08-09
> > > > >
> > > > > Dear Reviewer usBV,
> > > > >
> > > > > Thank you for recognizing the contribution of our paper. We really appreciate your constructive suggestions and comments.
> > > > >
> > > > > Best regards,
> > > > > Authors

---

### Official Review · Reviewer_iHRn · 2025-06-30

**Clarity:** 3
**Significance:** 3
**Originality:** 2
**Rating:** 5
**Confidence:** 3

**Summary:**

The authors establish a convergence rate for Gradient Descent applied on deep neural network with rail activations that is polynomial in the depth of the network $L$.

**Questions:**

Why is it necessary to assume that $R \geq 1$?

**Ethical Concerns:**

["NO or VERY MINOR ethics concerns only"]

**Final Justification:**

The paper is clear and well written. The result is significant and improves results presented in previous work. For these reasons, I consider the paper to be above the acceptance threshold.

**Limitations:**

Yes.

**Paper Formatting Concerns:**

N.A

**Quality:**

2

**Strengths And Weaknesses:**

**Strengths**

1. The paper is well written and easy to follow.
2. The authors present their proof technique and the relation to prior work in a very clear manner.
3. The result is significant and improves the exponential bound given in previous work.

**Weaknesses**

1. My main concern is about the correctness of Lemma 16, which is a central component of the paper. While Lemma 15 is proved only for the data points \$x\_i\$ in the training set, Lemma 16 applies it to points in the covering set \$D\$, which includes points outside the training set. Why is this use of Lemma 15 justified? Could the authors elaborate on this point? If I’m mistaken, I will revise my score accordingly. Another reason this transition may not be valid is that the network’s weights depends on the training set. Therefore, a bound that holds for the training set may not necessarily apply to points outside it.

2. Additionally, I am uncertain about the correctness of the proof of Lemma 15. In the inductive argument showing that $A_l \leq O(B_l) $, it is crucial that the constant hidden in the big-$O$ notation remains uniform across all induction steps. Otherwise, for example, if the constant grows multiplicatively at each step, the resulting bound would be exponential in $ l $. Could you clarify why this issue does not arise in the proof of Lemma 15?


3. A minor concern: The authors obtain a bound only in the case where \$m \geq d\$.

---

> ### Author Rebuttal · Authors · 2025-07-31
>
> Thanks for acknowledging the strengths of our work. Our point-to-point response to your comments follows next.
>
> > **W1.** My main concern is about the correctness of Lemma 16, which is a central component of the paper. While Lemma 15 is proved only for the data points $x_i$
>  in the training set, Lemma 16 applies it to points in the covering set $D$, which includes points outside the training set. Why is this use of Lemma 15 justified? Another reason this transition may not be valid is that the network’s weights depend on the training set. Therefore, a bound that holds for the training set may not necessarily apply to points outside it.
>
> We appreciate the opportunity to clarify the connection between Lemma 15 and Lemma 16. We claim that **all of our technical lemmas (including Lemma 15) hold for any finite set**. In particular, **Lemma 15 should be stated as follows**:
>
> - **Lemma 15'**
>
> Given a set of $N$ points on the sphere $K=\{x^1,\cdots,x^N\}$. Suppose $m\gtrsim L^{10}\log(NL/\delta)(\log m)^4R^2$. Then with probability at least $1-\delta$ over the randomness of initialization, for any $W \in \mathcal{B}_R(W(0)),j \in [N],l \in [L]$, there holds
> $$
> \|h^l(x^j)-h^l_0(x^j)\|_2\lesssim L^2R\sqrt{\frac{\log m}{m}},
> \|\Sigma^l(x^j)-\Sigma^l_0(x^j)\|_0\lesssim L^{4/3}(\log m)^{1/3}(mR)^{2/3}. \quad \text{(1)}
> $$
> As $K,N$ vary, we could get concentration inequalites on **different sets**.
> - **Applying Lemma 15' to the training set**
>
> We get the uniform control over $n$ points $x_1,\cdots,x_n$ (the original Lemma 15):
> Suppose $m\gtrsim L^{10}\log(nL/\delta)(\log m)^4R^2$. Then with probability at least $1-\delta$ over the randomness of initialization, for any $W \in \mathcal{B}_R(W(0)),i \in [n],l \in [L]$, there holds
> $$
> \|h^l(x_i)-h^l_0(x_i)\|_2\lesssim L^2R\sqrt{\frac{\log m}{m}},
> \|\Sigma^l(x_i)-\Sigma^l_0(x_i)\|_0\lesssim L^{4/3}(\log m)^{1/3}(mR)^{2/3} \quad \text{(2)}.
> $$
> - **Applying Lemma 15' to the covering set $D$**
>
>  Let $D= \{x^1,\cdots,x^{|D|}\}$ to be a $1/(C^L\sqrt{m})$-cover of the unit sphere ($C$ is a universal constant). Then we have:
> Suppose $m\gtrsim L^{10}\log(|D|L/\delta)(\log m)^4R^2$, then with probability at least $1-\delta$ over the randomness of initialization, for any $W \in \mathcal{B}_R(W(0)),j \in [|D|],l \in [L]$, there holds
> $$
> \|h^l(x^j)-h^l_0(x^j)\|_2\lesssim L^2R\sqrt{\frac{\log m}{m}},
> \|\Sigma^l(x^j)-\Sigma^l_0(x^j)\|_0\lesssim L^{4/3}(\log m)^{1/3}(mR)^{2/3}.\quad \text{(3)}
> $$
> - **Uniform control via covering**
>
>  For any $x\in \mathbb{S}^{d-1}$, there exists $x^j\in D$ such that $\|x-x^j\|_2\le 1/(C^L\sqrt{m})$. It then follows that $\|h^L(x)-h^L(x^j)\|_2\le\frac{1}{\sqrt{m}}, \|h^L_0(x)-h^L_0(x^j)\|_2\le\frac{1}{\sqrt{m}}$. Hence by triangle inequality we have $\|h^L(x)-h^L_0(x)\|_2\lesssim \frac{1}{\sqrt{m}}+L^2R\sqrt{\frac{\log m}{m}}$.
> - **On training set dependence**
>
> 1. We use (2) to analyze the optimization and estimate the Rademacher complexity. This **depends on the training set**.
> 2. We use (3) to prove Lemma 16, which **depends on the covering set $D$ and is independent of the training set.** Lemma 16 demonstrates the $\widetilde{O}(L^2)$-Lipschitzness of deep ReLU networks near initailization, which is **independent of the optimization trajectory**. The weights here are not updated via gradient descent; they remain within a neighborhood of the initialization. Lemma 16 is mainly used to derive a bound for $G'$ in Lemma 1. It is crucial to obtain a generalization bound that is only polynomial in $L$ (see Appendix C).
> 3. As $m\gtrsim L^{10}(\log m)^4R^2\max(\log(nL/\delta),\log(|D|L/\delta))$, (2) and (3) hold simutaneously with probability at least $1-\delta$. Both of them are important and they contribute to our theory **from different aspects**.
>
> We will revise Lemma 15 accoardingly and put (2) as a corollary in the next version.
> >**Q.** Why is it necessary to assume that  $R\ge1$?
>
> The assumption $R \ge 1$ is introduced for **notational simplicity** and does **not affect the generality** of our results. In particular, the bound $\frac{1}{\sqrt{m}}+L^2R\sqrt{\frac{\log m}{m}}$ can be written as  $L^2\sqrt{\frac{\log m}{m}}\max(R,1)$. Indeed, all of $R$ in our results can be replaced by $\max (R,1)$ by removing the assumption $R\ge1$.
>
>
>
> > **W2.** Additionally, I am uncertain about the correctness of the proof of Lemma 15. In the inductive argument showing that $A_l\le O(B_l)$, it is crucial that the constant hidden in the big-$O$ notation remains uniform across all induction steps. Otherwise, for example, if the constant grows multiplicatively at each step, the resulting bound would be exponential in $l$. Could you clarify why this issue does not arise in the proof of Lemma 15?
>
> In our analysis, **the constants hidden in the big-$O$ notation are uniform across all layers**, i.e., they do **not depend on the layer index $l$**. This is explicitly ensured by Lemma 12, which provides uniform bounds under overparameterization assumptions for all layers.
>
> - Specifically, in eq.(32) of our paper, the bound **holds uniformly** for all $l \in [L]$. In Lemma 15, we **do not apply a direct recursive argument** where each layer builds on the previous one with new constants. Instead, we **rely on Lemma 12** to establish uniform control at each layer to finish the induction, assuming the preconditions hold (which are guaranteed by a sufficiently large width $m$).
> - More concretely, we assume $\|h^l(x_i)-h^l_0(x_i)\|_2\le c_1L^2R\sqrt{\frac{\log m}{m}}$,$\|\Sigma^l(x_i)-\Sigma^l_0(x_i)\|_0\le c_2L^{4/3}(\log m)^{1/3}(mR)^{2/3},m\ge c_3L^{10}\log(nL/\delta)(\log m)^4R^2$ in Lemma 15. Let the universal constant in eq.(32) be $c_4$. Within Lemma 12, let $\|\widehat\Sigma^l(x_i)\|_0\le \frac{c_5m}{L^2\log m},\|\widehat G_b^a(x_i)\|_2\le c_6L\sqrt{\frac{\log m}{m}}$. We only need to choose $c_1,c_2,c_3$ to satisfy the following inequalities (which are independent of $l$): $c_1\ge 2, c_4+9c_1^2\le c_2, c_3\ge(2c_2c_5)^3, 2c_6\le c_1$. Hence, **all constants are fixed prior to the induction**, and do not accumulate as $l$ increases. This ensures that the final bounds in Lemma 15 remain polynomial in $L$, as claimed.
> - Similar analyses also appeared in other works [1,2],  which adopt uniform constants to control inductive arguments across layers.
>
> > **W3.** A minor concern: The authors obtain a bound only in the case where $m\ge d$.
> - We study the generalization performance of deep ReLU networks in the **overparameterized regime**, which aligns with common practical settings. In this case, the landscape is non-smooth and nonconvex, though the models can generalize well. Our work improves the understanding of gradient descent methods in deep neural networks. The assumption $m\ge d$ has been widely adopted in the existing literature [1,2].
> - Moreover, the assumption that the network has sufficiently many neurons is **often necessary** to ensure expressivity in high dimensions. [3] shows that **$\Omega(d^p)$ neurons** are required for NTK to learn a degree-$p$ polynomial. In the feature learning regime, even for simple data distribution (e.g., Guassian or boolean), [4,5] demonstrate that neural networks need $\Omega(d^s)$ neurons to learn single-index or multi-index models, where $s$ denotes the information exponent or leap complexity.
> - We agree that it is an interesting question to study the performance of deep networks in the $m<d$ regime. However, doing so might require **additional structural assumptions**, such as sparsity in the data distribution, low-dimensional manifold structure, or specific architectural biases. We view this as a promising direction for future research.
>
> [1] Zou et al. Stochastic gradient descent optimizes parameterized deep relu networks.
>
> [2] Chen et al. How much over-parameterization is sufficient to learn deep relu networks?
>
> [3] Ghorbani et al. When Do Neural Networks Outperform Kernel Methods?
>
> [4] Bietti et al. Learning single-index models with shallow neural networks.
>
> [5] Abbe et al. SGD learning on neural networks: leap complexity and saddle-to-saddle dynamics.

---

> > ### Comment · Reviewer_iHRn · 2025-08-03
> >
> > Thank you for the clarification. The concerns I raised in my review have been addressed.
> >
> > Following the rebuttal, I do have one additional question. The authors mention that “the landscape is non-smooth and non-convex.” However, the paper relies on Lemma 1 from Srebro (2010), which was originally proved under the assumption that the predictors are smooth. Could you please clarify why it is valid to apply this lemma in the non-smooth setting considered in the paper?

---

> ### Author Response · Authors · 2025-08-03
> **Clarification on smoothness**
>
> Thank you for your reply. We are glad to hear that the concerns raised in your previous review have been addressed. We also appreciate your additional comments regarding the smoothness.
>
> To clarify, Lemma 1 from Srebro (2010) only requires the loss function to be smooth; it does not impose any smoothness assumptions on the model itself. The key idea in Srebro (2010) is that the Lipschitz constant of the loss can be bounded by the loss value itself—known as the self-bounding property of smooth losses. This Lipschitz constant plays a crucial role in relating the covering numbers of the loss function class to those of the model class.
>
> By leveraging this self-bounding property, we can bound the Lipschitz constant in terms of the training errors, which then leads to the formulation of Lemma 1 from Srebro (2010). This explains why the training error appears in the bound presented in Srebro (2010). Importantly, this approach depends solely on the smoothness of the loss function, regardless of whether the model itself (e.g., a neural network) is smooth or not.
>
> In our work, we consider a smooth logistic loss. Therefore, Lemma 1 from Srebro (2010) remains applicable in our case. We will clarify this point in the revised version of the paper to avoid this confusion.

---

> > ### Comment · Reviewer_iHRn · 2025-08-04
> >
> > Thanks for the clarification regarding the smoothness assumption.
> >
> > After carefully reading the authors’ responses to all reviewers, I find the improvements in the dependencies on $n$ and $L$ to be important contributions. I have therefore raised my score to support the acceptance of this work.

---

> > > ### Author Response · Authors · 2025-08-04
> > > **thank you**
> > >
> > > Dear Reviewer iHRn,
> > >
> > > Thank you for your constructive suggestions and for recognising our contribution. We truly appreciate your efforts.
> > >
> > > best regards
> > > Authors

---

### Official Review · Reviewer_8eNE · 2025-07-03

**Clarity:** 3
**Significance:** 3
**Originality:** 2
**Rating:** 4
**Confidence:** 4

**Summary:**

This paper investigates the generalization error of $L$-th layer ReLU neural networks trained in the NTK regime. On the data that is $\gamma$-separated by the neural tangent kernel, the authors provide the excess risk of order $O(L^4(1+\gamma L^2)/\gamma^2 n)$, which is faster than the well-known suboptimal one $O(1/\sqrt{n})$ and only depends polynomially on the network depth $L$. They also provide a global convergence guarantee for the network using gradient descent and the NTK framework. In the analysis, they carefully evaluate the dependence on $L$ for the convergence analysis and Rademacher complexity bound, which enables them to obtain a sharper bound.

**Questions:**

- How is the initialization scheme defined in Assumption 1 crucial for obtaining the sharper bound? For example, while [Chen et al., 2021] considers the standard Gaussian initialization for the weights, can the same bound be derived for such settings?
- As the authors also mentioned as a future work, extending the results to stochastic optimization, which is covered by [Chent et al., 2021], will be a significant topic. What is the difficulty of such an extension?
- Another question is about the extension to other models. For example, how is the ReLU activation essential for obtaining the results, and is it possible to extend the results to other activations? Moreover, I am curious about the applicability of this approach to different models, such as ResNet and CNN.

**Ethical Concerns:**

["NO or VERY MINOR ethics concerns only"]

**Final Justification:**

I would like to maintain my original score.

**Limitations:**

The authors properly address the limitations.

**Quality:**

3

**Strengths And Weaknesses:**

**Strength**
- As the authors mentioned in the paper, much of the existing literature on the generalization error bound of deep neural networks exponentially depends on the network depth. This work mitigates this dependence to the polynomial one, which is a significant refinement.
- This paper is easy to follow, with detailed explanations of theorems and concise proof overviews.

**Weakness**
- The primary concern is that this paper mainly focuses on NTK-separable data, which imposes a substantial restriction on the data structure. It appears that several refinements, such as the polynomial dependence on the network depth, do not require the separability assumption. Therefore, it would be more plausible to compare the results obtained in this work with other NTK-based results that do not focus on the separability condition.

---

> ### Author Rebuttal · Authors · 2025-07-31
>
> Thanks for acknowledging the strengths of our work. Our point-to-point response to your comments follows next.
>
> > **W1.** This paper mainly focuses on NTK-separable data, which imposes a substantial restriction on the data structure.
>
> Our main contribution is that we are **the first to prove that deep ReLU networks trained by gradient descent can achieve optimal rate** ($1/n$). The existing works either obtain a bound of suboptimal rate $1/\sqrt{n}$ or focus on smooth activations. Our results hold in **general settings**, not limited to NTK-separable data.
> - Specifically, in Theorem 2, we provide generalization bounds in general cases, i.e., $\widetilde{O}\left(\frac{L^4F(\overline{W})}{n}\right)$. Hence, as long as there exists $F(\overline{W})$ with small error, the performance of the deep network trained with gradient descent is guaranteed.
> - The NTK-separability assumption **is only used to analyze $F(\overline{W})$ in a concrete case** and to better illustrate our main result. This assumption allows us to derive explicit estimates of $F(\overline{W})$, rather than being a necessary condition for our general theory. We believe our general generalization bound can be applied to various settings and demonstrate the power of deep learning.
> - The NTK separability assumption has been widely used in the literature [1,2,3,4]. It has been **theoretically justified in a range of settings**. For instance, [2] analyzes the margin through its dual form in different settings, including linearly separable case and the noisy $2$-XOR distribution.
>
> >**W2.** It would be more plausible to compare the results obtained in this work with other NTK-based results that do not focus on the separability condition.
>
> Many NTK-based results assume positive eigenvalues of NTK Gram matrix. [5] points out that NTK separable data is a milder assumption. In general, we need to estimate $F(\overline{W})$, or construct a predictor $f_{\overline{W}}(x)$ that achieves low training error under the structural  data assumption. For example, this is closely related to the Neural Tangent  Random Feature (NTRF) model, which has been well studied in the appealing work [3].
>
> > **Q1.** How is the initialization scheme defined in Assumption 1 crucial for obtaining the sharper bound? For example, while [Chen et al., 2021] considers the standard Gaussian initialization for the weights, can the same bound be derived for such settings?
>
> The symmetric initialization assumed in Assumption 1 is **a commonly used technique** in the theoretical deep learning literature [6], primarily to simplify the analysis.
> - Specifically, this ensures at the beginning, $f_{W(0)}(x_i)=0, \frac{\partial f_{W(0)}(x_i)}{\partial W^l(0)}=0,l\in[L-1]$, as illustrated in Lemma 7. This significantly reduces the complexity of bounding the error dynamics.
> - While our current proof uses this symmetric setup, it is not essential. Under standard Gaussian initialization, these quantities can still be shown to be **concentrated near zero** using standard concentration tools. For example, [3] shows that these quantities can be bounded by $\mathrm{poly}(\log(n/\delta))$ terms, which are **negligible in our final bounds**.
>
> > **Q2.** As the authors also mentioned as a future work, extending the results to stochastic optimization, which is covered by [Chen et al., 2021], will be a significant topic. What is the difficulty of such an extension?
>
> - While [3] has studied the online SGD setting, we believe that our theoretical framework can be adapted to **multi-pass SGD**. However, a major difficulty lies in carefully controlling the **trade-off between optimization error and generalization gap**. For example, how the number of updates $T$ grows with the dataset size $n$.
> - Moreover, analyzing **variants of SGD** such as noisy SGD, decentralized SGD, or SGD with momentum introduces additional challenges. These algorithms involve extra sources of randomness, communication constraints, or memory terms, which make it harder to track the evolution of the parameters.
>
> > **Q3.** How is the ReLU activation essential for obtaining the results, and is it possible to extend the results to other activations?
> - ReLU is widely used in practice for its **efficienct signal propagation and sparsity**. Theoretically, it enjoys a **variance-preserving property** at initialization: $\mathbb{E}\|h^l_0(x_i)\|_2^2=\|h^{l-1}_0(x_i)\|_2^2$. This property makes it possible to control the hidden layers and activation patterns rigorously (Appendix A).
> - For other activations, the core techniques in our paper **can still be applied**, but **additional assumptions may be needed**. For example, smoothness, convexity, Lipschitzness or other conditions to ensure signal propagation remains well-behaved.
>
> > **Q4.** The applicability of this approach to different models, such as ResNet and CNN.
>
> Thanks for raising this interesting point. We agree that the generalization analysis in different models is interesting for understanding deep ReLU networks. **The core techniques such as controlling the Lipschitz constant and Rademacher complexity can be extended to other architectures**. For example, [7] studies the NTK Gram matrix of ResNet and CNN. Building on their insights, it is interesting to explore how our framework can be used in these models, deriving architecture-specific generalization bounds.
>
>
> [1] Ji et al. Polylogarithmic width suffices for gradient descent to achieve arbitrarily small test error with shallow relu networks.
>
> [2] Taheri et al. Generalization and stability of interpolating neural networks with minimal width.
>
> [3] Chen et al. How much over-parameterization is sufficient to learn deep relu networks?
>
> [4] Taheri et al. Sharper guarantees for learning neural network classifiers with gradient methods.
>
> [5] Nitanda et al. Gradient descent can learn less over-parameterized two-layer neural networks on classification problems.
>
> [6] Xu et al. Overparametrized multi-layer neural networks: Uniform concentration of neural tangent kernel and convergence of stochastic gradient descent.
>
> [7] Du et al. Gradient descent finds global minima of deep neural networks.

---

> > ### Comment · Reviewer_8eNE · 2025-08-04
> >
> > I sincerely appreciate the authors' response.
> > The responses adequately address my concern, and then I would like to maintain my current score.

---

### Decision · Program_Chairs · 2025-09-17

**Decision:**

Accept (poster)

**Comment:**

This paper proves optimal generalization rates for deep networks in the NTK regime trained on a classification task. A new method for handling the regularity of the outputs w.r.t. to small changes of the weights around initialization, thus replacing an exponential dependence on depth with a polynomial one. The optimal rates of convergence are then proven assuming that the classes are separable with a certain margin w.r.t. the NTK geometry. The paper mixes approaches from NTK analysis and uniform generalization bounds.

We are happy to accept this paper as a poster at NeurIPS, in line with the recommendations of the reviewers.